# PHYSICS-INFORMED LEARNING UNDER MIXING: HOW PHYSICAL KNOWLEDGE SPEEDS UP LEARNING

**Anna Scampicchio*** *
Department of Electrical Engineering
Chalmers University of Technology, Sweden
`anna.scampicchio@chalmers.se`

**Leonardo F. Toso***
Department of Electrical Engineering
Columbia University, USA
`leonardo.toso@columbia.edu`

**Rahel Rickenbach**
Institute for Dynamic Systems and Control
ETH Zürich, Switzerland
`rrahel@ethz.ch`

**James Anderson**
Department of Electrical Engineering
Columbia University , USA
`james.anderson@columbia.edu`

**Melanie N. Zeilinger**
Institute for Dynamic Systems and Control
ETH Zürich, Switzerland
`mzeilinger@ethz.ch`

## ABSTRACT

A major challenge in physics-informed machine learning is to understand how the incorporation of prior domain knowledge affects learning rates when training data is not independent and identically distributed. Focusing on empirical risk minimization with physics-informed regularization, we derive complexity-dependent bounds on the excess risk in probability and in expectation. We prove that, when the physical prior information is aligned, the learning rate improves from the (slow) Sobolev minimax rate to the (fast) optimal i.i.d. one without sample-size deflation due to data dependence.

## 1 INTRODUCTION

Physics-informed machine learning encompasses a wide taxonomy of approaches that combine physical knowledge and learning algorithms to address two main tasks: (i) leveraging data-driven methods to enhance numerical solvability and accuracy of physical models given, for instance, by systems of partial differential equations; (ii) improve learning algorithm performance by including physical information in the loss function or as an additional constraint (Karniadakis et al., 2021; Meng et al., 2025). Focusing on the second class of methods, surveyed in Rai & Sahu (2020); von Rueden et al. (2023b), the resulting approaches turn out to be practically effective in terms of data efficiency, generalization capability and interpretability, which play an important role especially in view of downstream tasks such as safe learning-based control (Nghiem et al., 2023; Drgona et al., 2025). However, despite the empirical success of physics-informed machine learning algorithms (Raissi et al., 2019; Rai & Sahu, 2020; Cuomo et al., 2022; Hao et al., 2023), theoretically quantifying the beneficial impact of incorporating physical information into learning algorithms is technically challenging and still an active area of research (see von Rueden et al. (2023a) and references therein).

We tackle this problem by considering a statistical learning set-up and focusing on *regularized empirical risk minimization* problems of the following form:

$$\hat{f} = \underset{\substack{f \in \text{ ball in} \\ \text{Sobolev space}}}{\arg\min} \quad \boxed{\begin{array}{c} \text{data-fit} \\ \text{squared loss}(f) \end{array}} + \lambda_T \boxed{\begin{array}{c} \text{physics-informed} \\ \text{regularizer}(f) \end{array}}, \tag{1.1}$$

---

*Corresponding author. A. Scampicchio and L.F. Toso share first-authorship of this paper.

where data entering the fit term are *dependent*, derived from observations of a ground-truth nonlinear dynamical system $X_{t+1} = f_\star(X_t) + W_t$, with $W_t$ being a sub-Gaussian noise martingale difference sequence. The regularizer in (1.1) encodes the information that the true function to be estimated, $f_\star$, approximately satisfies a known partial differential equation induced by a linear operator $\mathscr{D}$ — i.e., we have that the regularizer takes the form $\|\mathscr{D}(f)\|_{\mathscr{L}^2}^2$, and we say that *knowledge alignment* occurs if it holds that $\|\mathscr{D}(f_\star)\|_{\mathscr{L}^2}^2 \simeq 0$.

The main results of this paper are *complexity-dependent* bounds — i.e., bounds that depend on $\|\mathscr{D}(f_\star)\|_{\mathscr{L}^2}$ (Lecué & Mendelson, 2017) — for the *excess risk* $\|\hat{f} - f_\star\|_{\mathscr{L}^2}^2$. Informally, our results (both in high probability and in expectation) take the form given below:

**Theorem (Informal).** For a suitable choice of the regularization parameter $\lambda_T$, for a sufficiently large number of samples $T$, and letting $d < 1$ be the *Sobolev minimax rate* (Ibragimov & Has'minskii, 1981; Nussbaum, 2006), it holds that

$$\text{(Excess risk)} \quad \|\hat{f} - f_\star\|_{\mathscr{L}^2}^2 \le C_{\texttt{slow}} \frac{\|\mathscr{D}(f_\star)\|_{\mathscr{L}^2}^{\text{some power}}}{T^d} + C_{\texttt{fast}} \frac{\text{noise level}}{T}.$$

The error bound established above reveals for the first time that, under knowledge alignment, the regularized estimate $\hat{f}$ converges to the true, unknown function $f_\star$ at the i.i.d. rate of $\mathcal{O}(1/T)$: in other words, it behaves like classic optimal rates for i.i.d. learning *even with dependent data* after a suitable burn-in time.

The remainder of the paper unfolds as follows: Section 2 provides the set-up of the learning problem, introducing the *weighted, vector-valued* function spaces that will be used throughout the paper. Next, the learning problem is stated in Section 3, and in Section 4 we provide the general statement for the excess risk bounds, both in probability and in expectation. Our analysis culminates in Section 5, where we prove how knowledge alignment leads to optimal i.i.d. rates even if data are dependent. We discuss our results in juxtaposition with related works in Section 6, and present some concluding remarks in Section 7.

## 2 PROBLEM SET-UP

This section collects preliminary concepts, defining the probability set-up of the data-generation mechanism (Section 2.1) and the involved weighted, vector-valued function spaces (Section 2.2).

### 2.1 INPUT DOMAIN AND TRAJECTORY DISTRIBUTION

Let $\Omega \subseteq [-L, L]^{d_X} \subset \mathbb{R}^{d_X}$ be the input domain whose boundary is locally Lipschitz (Adams & Fournier, 2003, Definition 4.9). For a horizon length $T$, the input trajectories denoted by $X \doteq (X_0, X_1, \cdots, X_{T-1})$ belong to the metric space $(\Omega^T, \{\mathcal{X}_t\}_{t=0}^{T-1}, \mathbb{P}_X)$, where $\Omega^T \doteq \bigtimes_{t=0}^{T-1} \Omega$ is the Cartesian product of the single-component input domains $\Omega$; $\{\mathcal{X}_t\}_{t=0}^{T-1}$ is the *filtration* given by a sequence of increasing $\sigma$-algebras $\mathcal{X}_{t+1} \subset \mathcal{X}_t$ with respect to which $X$ is *adapted* (Rogers & Williams, 2000, Chapter II.45); and $\mathbb{P}_X$ is the joint probability distribution of the input trajectory. As detailed in Section A.1, there exists a probability distribution associated with every component of $X$ — we denote it by $\mu_t$ for each $t = 0, \cdots, T-1$, and we mostly work with a *known* initial distribution $\mu_0$ for $X_0$ (typically, a Dirac measure centered at the observed initial state $X_0$). Overall, we make use of the following:

*Assumption* 1. Let $\mu_\lambda$ be the Lebesgue measure defined on $\Omega \subset \mathbb{R}^{d_X}$. For all $t = 0, \cdots, T-1$, each measure $\mu_t \colon \mathcal{X}_t \to \mathbb{R}_{\ge 0}$ is assumed to admit a density with respect to $\mu_\lambda$. We denote such density by $p_t(\cdot)$, and we assume that there exist $0 < \underline{\kappa} < \overline{\kappa} < \infty$ such that, for all $t = 0, \cdots, T-1$, $\underline{\kappa} \le p_t(\cdot) \le \overline{\kappa}$.

Note that Assumption 1 accounts for many cases of practical relevance, such as the uniform, truncated Gaussian, and beta distributions (Krishnamoorthy, 2016).

## 2.2 SPACES OF FUNCTIONS

**Space of square-integrable functions $\mathscr{L}^2$.** We will focus on the Hilbert space $\mathscr{L}^2(\Omega^T, \mathbb{P}_X; \mathbb{R}^{d_Y})$ of vector-valued, square-integrable functions that consist of multiple evaluations of a function $f \colon \Omega \to \mathbb{R}^{d_Y}$ along the input trajectory $X$. Such a space allows us to consider the trajectory $X$ and is endowed with the inner product defined as follows: given $f, g \colon \Omega \to \mathbb{R}^{d_Y}$, we have

$$\langle f, g \rangle_{\mathscr{L}^2(\Omega^T, \mathbb{P}_X; \mathbb{R}^{d_Y})} \doteq \frac{1}{T} \sum_{t=0}^{T-1} \mathbb{E}_{\mathbb{P}_X} \left[ \langle f(X_t), g(X_t) \rangle_2 \right] = \frac{1}{T} \sum_{t=0}^{T-1} \int_{\Omega^T} \langle f(X_t), g(X_t) \rangle_2 \, d\mathbb{P}_X$$

$$= \frac{1}{T} \sum_{t=0}^{T-1} \int_{\Omega} \langle f(X_t), g(X_t) \rangle_2 \, \mu_t(dX_t), \tag{2.1}$$

where $\langle \cdot, \cdot \rangle_2$ is the standard inner product defined in the Euclidean space $\mathbb{R}^{d_Y}$, and $\mu_t$ is the probability measure of the $t$-th component of $X$ introduced in Section 2.1. The inner product (2.1) induces the trajectory norm $\|f\|_{\mathscr{L}^2(\Omega^T, \mathbb{P}_X; \mathbb{R}^{d_Y})}$ such that $\|f\|_{\mathscr{L}^2(\Omega^T, \mathbb{P}_X; \mathbb{R}^{d_Y})}^2 := \langle f, f \rangle_{\mathscr{L}^2(\Omega^T, \mathbb{P}_X; \mathbb{R}^{d_Y})}$. Moreover, it follows by construction that one can write $\|f\|_{\mathscr{L}^2(\Omega^T, \mathbb{P}_X; \mathbb{R}^{d_Y})}^2 = \frac{1}{T} \sum_{t=0}^{T-1} \mathbb{E}_{\mathbb{P}_X}[\|f(X_t)\|_2^2]$. Note in addition that, thanks to the separability of $\mathbb{R}^{d_Y}$, the vector-valued space $\mathscr{L}^2(\Omega^T, \mathbb{P}_X; \mathbb{R}^{d_Y}) = \{f \colon \Omega \to \mathbb{R}^{d_Y} \mid \|f\|_{\mathscr{L}^2(\Omega^T, \mathbb{P}_X; \mathbb{R}^{d_Y})} < \infty\}$ can be written as the direct sum $\bigoplus_{i=1}^{d_Y} \mathscr{L}^2(\Omega^T, \mathbb{P}_X; \mathbb{R})$ (Conway, 2007, Chapter I.6): indeed, following (2.1), we can write

$$\|f\|_{\mathscr{L}^2(\Omega^T, \mathbb{P}_X; \mathbb{R}^{d_Y})}^2 = \sum_{i=1}^{d_Y} \frac{1}{T} \sum_{t=0}^{T-1} \mathbb{E}_{\mathbb{P}_X} \left[ f_i(X_t)^2 \right] = \sum_{i=1}^{d_Y} \|f_i\|_{\mathscr{L}^2(\Omega^T, \mathbb{P}_X; \mathbb{R})}^2.$$

**General $\mathscr{L}^p$ spaces.** In general, one can define the space $\mathscr{L}^p(\Omega^T, \mathbb{P}_X; \mathbb{R}^{d_Y})$ for any $p \in \mathbb{Z}_{\geq 0}$ endowed with the norm $\|f\|_{\mathscr{L}^p(\Omega^T, \mathbb{P}_X; \mathbb{R}^{d_Y})}^p = \frac{1}{T} \sum_{t=0}^{T-1} \mathbb{E}_{\mathbb{P}_X}[\|f(X_t)\|_2^p]$. Of particular interest will be the Banach space of bounded functions $\mathscr{L}^\infty(\Omega^T; \mathbb{R}^{d_Y})$ equipped with the norm $\|f\|_{\mathscr{L}^\infty(\Omega^T; \mathbb{R}^{d_Y})} \doteq \sup_{x \in \Omega} \|f(x)\|_2$.

**Sobolev space $\mathscr{H}^s$.** Another fundamental function space derived from $\mathscr{L}^2(\Omega^T, \mathbb{P}_X; \mathbb{R}^{d_Y})$ is the multi-output, weighted *Sobolev space* of order $s \in \mathbb{Z}_{\geq 0}$, which is defined as follows:

$$\mathscr{H}^s(\Omega^T, \mathbb{P}_X; \mathbb{R}^{d_Y}) \doteq \left\{ f \in \mathscr{L}^2(\Omega^T, \mathbb{P}_X; \mathbb{R}^{d_Y}) \mid \|f\|_{\mathscr{H}^s(\Omega^T, \mathbb{P}_X; \mathbb{R}^{d_Y})} < \infty \right\},$$

where the norm is induced by the inner product

$$\langle f, g \rangle_{\mathscr{H}^s(\Omega, \mathbb{P}_X; \mathbb{R}^{d_Y})} \doteq \sum_{|\alpha| \leq s} \langle D^\alpha f, D^\alpha g \rangle_{\mathscr{L}^2(\Omega^T, \mathbb{P}_X; \mathbb{R}^{d_Y})},$$

with $D^\alpha f$ being the differential given by the multi-index $\alpha \doteq (\alpha_1, \cdots, \alpha_{d_X})$ of non-negative integers with order $|\alpha| \doteq \sum_{i=1}^{d_X} \alpha_i$, i.e., $D^\alpha \doteq \partial^{|\alpha|} f / \partial x_1^{\alpha_1} \cdots \partial x_{d_X}^{\alpha_{d_X}}$. Regarding the order of the Sobolev spaces we will consider, we will rely on the following:

*Assumption* 2. The order $s$ of $\mathscr{H}^s(\Omega^T, \mathbb{P}_X; \mathbb{R}^{d_Y})$ is a non-negative integer that satisfies $s \geq 2d_X$.

Finally, note that also the space $\mathscr{H}^s(\Omega^T, \mathbb{P}_X; \mathbb{R}^{d_Y})$ admits the representation as the direct sum $\bigoplus_{i=1}^{d_Y} \mathscr{H}^s(\Omega^T, \mathbb{P}_X; \mathbb{R})$ thanks to the separability of $\mathbb{R}^{d_Y}$. This allows us to extend key results of scalar Sobolev spaces to our vector-valued ones, as detailed in Section B. In particular, we show that the Sobolev Imbedding Theorem (Adams & Fournier, 2003, Theorem 4.12) holds in our set-up, which will provide the necessary structure for the hypothesis space involved in the learning problem.

## 3 PROBLEM STATEMENT

**Measurement model.** Our data consists of trajectories of length $T$, denoted , $\mathcal{D} \doteq \{X_t, Y_t\}_{t=0}^{T-1}$, generated according to the measurement model

$$Y_t \doteq X_{t+1} = f_\star(X_t) + W_t, \tag{3.1}$$

where the noise sequence satisfies the following:

*Assumption* 3. The additive noise $\{W_t\}_{t\in\mathbb{Z}_{\geq 0}}$ is a martingale difference sequence with respect to the filtration $\{\mathcal{X}_t\}_{t\in\mathbb{Z}_{\geq 0}}$: thus, $\mathbb{E}_{W_t}[W_t|\mathcal{X}_{t-1}] = 0$ for all $t = 0, \cdots, T-1$. Moreover, each $W_t$ is also assumed to be $\sigma_W^2$-conditionally sub-Gaussian given $\mathcal{X}_{t-1}$: i.e., it holds that, for every $\xi \in \mathbb{R}$ and every $u$ in the unit sphere in $(\mathbb{R}^{d_Y}, \|\cdot\|_2)$,

$$\mathbb{E}\left[\exp\left\{\xi \langle W_t, u\rangle_2\right\} \mid \mathcal{X}_{t-1}\right] \leq \exp\left\{\frac{\xi^2 \sigma_W^2}{2}\right\}. \tag{3.2}$$

**The learning problem.** In general, the learning problem can be stated as that of minimizing the *excess risk* $||\hat{f} - f_\star||^2_{\mathscr{L}^2(\Omega^T, \mathbb{P}_X; \mathbb{R}^{d_Y})}$, searching for the estimate $\hat{f}$ within a chosen *hypothesis space* $\mathscr{F}$ (which we specify later). However, since the underlying probability measures are unknown, the amount of data in $\mathcal{D}$ is finite and the hypothesis space $\mathscr{F}$ might be large, the estimate $\hat{f}$ is typically computed through *(regularized) empirical risk minimization*:

$$\hat{f} \doteq \underset{f \in \mathscr{F}}{\arg\min} \frac{1}{T} \sum_{t=0}^{T-1} \|Y_t - f(X_t)\|_2^2 + \lambda_T \Psi(f). \tag{3.3}$$

**Physics-informed regularization.** The regularizer $\Psi(\cdot)\colon \mathscr{F} \to \mathbb{R}_{\geq 0}$ encodes available prior physical information about the "true" function $f_\star$ — in other words, $\overline{\Psi}(f)$ is designed to penalize the physical inconsistency between $f$ and the prior on $f_\star$. Such physical information is conveyed by the fact that $f_\star$ is assumed to approximately satisfy a known partial differential equation given by the linear operator $\mathscr{D}\colon \mathscr{H}^s(\Omega, \mu_\lambda; \mathbb{R}^{d_Y}) \to \mathscr{L}^2(\Omega, \mu_\lambda; \mathbb{R}^{d_Y})$. Such an operator is defined component-wise as

$$[\mathscr{D}(f)]_i \doteq \sum_{|\alpha| \leq s} p_{i,\alpha} D^\alpha f_i \text{ for all } i = 1, \cdots, d_Y, \tag{3.4}$$

where each $p_{i,\alpha}\colon \Omega \to \mathbb{R}$ is a bounded function — therefore, if we denote by $p$ the collection of all $p_{i,\alpha}$, then we have that $\|p\|_\infty$ is finite. To describe the regularity of the differential operator in (3.4), we make the following assumption:

*Assumption* 4. The differential operator $\mathscr{D}(f)$ is *elliptic* — that is, for all $i = 1, \cdots, d_Y$ and any $\xi \in \mathbb{R}^{d_X} \setminus \{0\}$, it holds that $\sum_{|\alpha|=s} p_{i,\alpha} \xi_1^{\alpha_1} \cdots \xi_{d_X}^{\alpha_{d_X}} \neq 0$.

Elliptic partial differential equations abound in practical applications, as they can be seen as generalizations of the Laplace and Poisson operators (Evans, 2010, Chapter 6). The differential operator $\mathscr{D}$ enters the definition of the regularizer in (3.3), where we have

$$\Psi(f) \doteq \|\mathscr{D}(f)\|^2_{\mathscr{L}^2(\Omega^T, \mathbb{P}_X; \mathbb{R}^{d_Y})}, \tag{3.5}$$

which is a 2-proper regularizer (Lecué & Mendelson, 2017, Assumption 1.1) — see Section E for the definition and further insights.

**Hypothesis space.** Let us now focus on the hypothesis space $\mathscr{F}$. We consider it as the ball of radius $\rho_f$ in the Sobolev space, i.e.,

$$\mathscr{F} \doteq \left\{ f \in \mathscr{H}^s(\Omega^T, \mathbb{P}_X; \mathbb{R}^{d_Y}) \mid \|f\|_{\mathscr{H}^s(\Omega^T, \mathbb{P}_X; \mathbb{R}^{d_Y})} \leq \rho_f \right\}. \tag{3.6}$$

Alternatively, as pointed out in (Cucker & Zhou, 2007, Theorem 8.21), one could write the cost in (3.3) as $\frac{1}{T} \sum_{t=0}^{T-1} (Y_t - f(X_t))^2 + \tilde{\lambda}_T \|f\|^2_{\mathscr{H}^s(\Omega^T, \mathbb{P}_X; \mathbb{R}^{d_Y})} + \lambda_T \Psi(f)$, and the minimization would be performed for $f \in \mathscr{H}^s(\Omega^T, \mathbb{P}_X; \mathbb{R}^{d_Y})$, thanks to the equivalence yielding $\rho_f = \rho_f(\tilde{\lambda}_T)$. In this paper, we make the following standing assumption:

*Assumption* 5. The hypothesis space $\mathscr{F}$ contains the unknown function to be estimated, $f_\star$.

In case such an assumption is violated, the excess risk bound would feature an additional term quantifying the error introduced by the mis-specification of the hypothesis space $\mathscr{F}$. Such a bias error

is entirely data-independent and deterministic — therefore, it is decoupled from the analysis performed in this paper. In general, tools to bound the bias error rely on *interpolation theory* (Lunardi, 2018), and some results can be found, e.g., in Cucker & Smale (2002); Cucker & Zhou (2007).

Additionally, we will also consider the *effective hypothesis space* induced by the regularizer, namely

$$\mathscr{F}^\rho = \left\{ f \in \mathscr{F} \,\middle|\, \Psi(f - f_\star) \le \rho \right\}. \tag{3.7}$$

For a visualization of these hypothesis spaces, please refer to Figure 1. Finally, we will sometimes simplify notation by considering the shifted hypothesis space $\mathcal{H}_\star \doteq \mathcal{H} - f_\star = \{f - f_\star \mid f \in \mathcal{H}\}$, with $\mathcal{H}$ being for instance $\mathscr{F}$ or $\mathscr{F}^\rho$.

**Modelling sample dependence in trajectories.** Finally, we assume regularity in the trajectory $X$ given by the following one-sided exponential inequality (Samson, 2000):

*Assumption* 6. The trajectory $X$ governed by the law $\mathbb{P}_X$ in the hypothesis class $\mathscr{F}$ is $S$-persistent for some $S \in [1, \infty)$. Specifically, for every $\xi \ge 0$ and every $f \in \mathscr{F}$, we have that

$$\mathbb{E}\left[\exp\left(-\xi \sum_{t=0}^{T-1} \|f(X_t)\|_2^2\right)\right] \le \exp\left(-\xi \sum_{t=0}^{T-1} \mathbb{E}\left[\|f(X_t)\|_2^2\right] + \frac{\xi^2 S}{2} \sum_{t=0}^{T-1} \mathbb{E}\left[\|f(X_t)\|_2^4\right]\right).$$

Typically, $S$ is expressed in terms of the *dependence matrix* of $X$ (see Section A.2 for its definition), and such a parameter attains higher values the more dependent $X_t$ is on its past. In general, $S$ might depend on $T$; however, in this paper we will focus on the case in which $S$ is a constant: as pointed out in (Samson, 2000, Section 2), this is a rather weak condition satisfied by a large class of Markov chains and of $\phi$-mixing processes — see Section A.2 for more details.

**Contribution.** Our results demonstrate that the physics-informed regularization in the empirical risk minimization problem (3.3) can speed-up the learning even in presence of dependent data. In particular, we derive complexity-dependent bounds for the excess risk $\|\hat{f} - f_\star\|_{\mathscr{L}^2(\Omega^T, \mathbb{P}_X; \mathbb{R}^{d_Y})}^2$, both in probability and in expectation, for learning under mixing, and prove that the rate of the excess risk matches the one from i.i.d learning in presence of knowledge alignment. Therefore, our results theoretically quantify the beneficial impact of physical knowledge in learning algorithms, even in the challenging scenario of learning with dependent data.

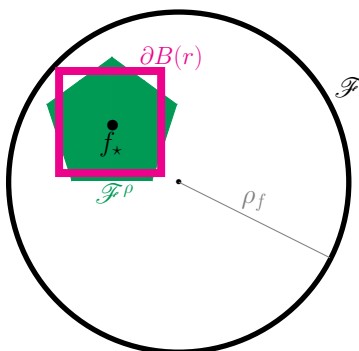

Figure 1: Visualization of the involved hypothesis spaces. Note that the set $\partial B(r) = \{f \in \mathscr{F}_\star \mid \|f\|_{\mathscr{L}^2(\Omega^T, \mathbb{P}_X; \mathbb{R}^{d_Y})}^2 = r^2\}$ introduced in Section 4.1 is represented as a square to highlight the fact that the norm therein involved is different to the one defining $\mathscr{F}$ (3.6). Similarly, we represented $\mathscr{F}^\rho$ (3.7) as a convex set that is not necessarily a ball in the Sobolev norm.

## 4 ERROR BOUNDS

We now present the bounds for the excess risk, both in probability and in expectation. We start in Section 4.1 by conveying the underlying ideas that lead to those results, and then provide the bound in probability (Section 4.2) and in expectation (Section 4.3). These results will be further analyzed in Section 5 to obtain our main claims on the convergence rate of learning with physics-informed regularization. Before proceeding, we emphasize that the excess risk is a random quantity depending on the distribution of the input sequence $X$ and of the noises $\{W_t\}_{t=0}^{T-1}$: therefore, often we will simply write $\mathbb{P}$ and $\mathbb{E}$ instead of $\mathbb{P}_{\mathbb{P}_X, W}$ and $\mathbb{E}_{\mathbb{P}_X, W}$ to streamline notation.

### 4.1 THE IDEA

The main idea consists of identifying an event according to which, with high probability and for some parameter $\theta$,

$$\|f - f_\star\|^2_{\mathscr{L}^2(\Omega^T, \mathbb{P}_X; \mathbb{R}^{d_Y})} \leq \frac{\theta}{T} \sum_{t=0}^{T-1} \|f(X_t) - f_\star(X_t)\|^2_2. \tag{4.1}$$

This kind of one-sided concentration inequality was studied for the i.i.d. setting in Mendelson (2014), to which we defer for a thorough discussion. The proof that (4.1) holds with high probability in the i.i.d. case is given in Mendelson (2014) thanks to the *small-ball condition*, which is a rather weak assumption from a statistical point of view: see the discussion after Assumption 1.2 in Lecué & Mendelson (2017), together with its interpretation in terms of identifiability. In our data-dependent setting, the small-ball condition will be imposed by $(C, \alpha)$-hypercontractivity with $\alpha = 2$ (see Section D.2), and we show that it holds in the set $\partial B(r) \doteq \{f \in \mathscr{F} \mid \|f - f_\star\|^2_{\mathscr{L}^2(\Omega^T, \mathbb{P}_X; \mathbb{R}^{d_Y})} = r^2\}$ for any fixed $r > 0$. Therefore, the probability level of the event in (4.1) will be controlled by the radius $r$. We present a visualization of $\partial B(r)$, together with all the hypothesis spaces, in Figure 1. Crucially, inequality (4.1) allows us to shift the analysis of the excess risk to that of its empirical version. The next step consists then in upper-bounding the latter (i.e., the right-hand side in (4.1)) by the *martingale offset complexity* of the effective hypothesis space, $\mathbf{M}_T[\mathscr{F}^\rho_\star]$. In particular, for every $f \in \mathscr{F}^\rho_\star$ (i.e., $f = f' - f_\star$ for some $f' \in \mathscr{F}^\rho$), one has that

$$\frac{1}{T} \sum_{t=0}^{T-1} \|f(X_t)\|^2_2 \leq \sup_{f \in \mathscr{F}^\rho_\star} \frac{1}{T} \sum_{t=0}^{T-1} 4 \langle W_t, f(X_t) \rangle_2 - \|f(X_t)\|^2_2 \doteq \mathbf{M}_T[\mathscr{F}^\rho_\star]. \tag{4.2}$$

We defer to Theorem G.1 for a derivation of such an inequality. Along the lines of Liang et al. (2015), we would like to stress that the term $\|f(X_t)\|^2_2$ in the right-hand side introduces a self-normalizing effect that compensates the fluctuations of the term $\langle W_t, f(X_t) \rangle_2$. This fact is key in making the martingale offset complexity *not depend on mixing*, as discussed in Section 5. One can provide bounds in probability and in expectation for the martingale offset complexity (see Section G), and these will play a key role in the excess risk bounds that we present in the remainder of the section and further discuss in Section 5.

Before presenting the aforementioned bounds, let us formally introduce the *lower isometry event*, which is the complement of (4.1), whose probability we bound in Section F:

$$\mathcal{A}_r \doteq \sup_{f \in \mathscr{F}^\rho_\star \setminus B(r)} \left\{ \frac{1}{T} \sum_{t=0}^{T-1} \|f(X_t)\|^2_2 - \frac{1}{\theta} \|f\|^2_{\mathscr{L}^2(\Omega^T, \mathbb{P}_X; \mathbb{R}^{d_Y})} \leq 0 \right\}.$$

We are now in place to present the bounds for the excess risk, both in probability and in expectation.

### 4.2 RESULT IN PROBABILITY

**Theorem 4.1.** *Let Assumptions 1 to 3, 5 and 6 hold. Consider a parameter $\theta > 8$, and let $\hat{f}$ be the solution of the estimation problem (3.3) with $\lambda_T > 0$, and let the radius $\rho$ defining the effective hypothesis class $\mathscr{F}^\rho$ be such that $\rho \geq 10\Psi(f_\star)$. Then, on the event*

$$\mathcal{A}^{\complement}_r \cap \left\{ \lambda_T \geq \frac{40}{3\rho} \mathbf{M}_T[\mathscr{F}^\rho] \right\}$$

*we have that*

$$\left\| \hat{f} - f_\star \right\|^2_{\mathscr{L}^2(\Omega^T, \mathbb{P}_X; \mathbb{R}^{d_Y})} \leq \theta \mathbf{M}_T[\mathscr{F}^\rho] + 2\lambda_T \Psi(f_\star) + r^2. \tag{4.3}$$

*Proof.* (Sketch). The proof follows Lecué & Mendelson (2017); Ziemann & Tu (2022) and it consists in characterizing the scenarios that lead to the event $\mathcal{A}^{\complement}_r$, showing that the case for which $\hat{f} \in \mathscr{F} \setminus \mathscr{F}^\rho$ cannot occur for $\lambda_T$ sufficiently large. The detailed proof is given in Section H.1. $\quad\square$

### 4.3 RESULT IN EXPECTATION

**Theorem 4.2.** *Let Assumptions 1 to 3, 5 and 6 hold. Having the set $\partial B(r)$ that is $(C(r), 2)$-hypercontractive, let $\mathscr{F}_r$ be a $r/\sqrt{\theta}$-cover in the infinity norm of $\partial B(r)$, and let $\mathcal{N}_\infty \left( \partial B(r), \frac{r}{\sqrt{\theta}} \right)$ be the associated covering number. Consider the regularized empirical risk minimization problem in (3.3) with regularization parameter satisfying $\lambda_T \geq \frac{40}{3\rho} \mathbb{E}_W \left[ \mathbf{M}_T \left[ \mathscr{F}^\rho \right] \right]$, where $\rho \geq 10\Psi(f_\star)$. Then, letting $B$ be the positive constant such that $\|f\|_{\mathscr{L}^\infty(\Omega^T; \mathbb{R}^{d_Y})} \leq B$ for all $f \in \mathscr{F}$, we have*

$$\mathbb{E}\left[ \left\| \hat{f} - f_\star \right\|^2_{\mathscr{L}^2(\Omega^T, \mathbb{P}_X; \mathbb{R}^{d_Y})} \right] \leq 4B^2 \mathcal{N}_\infty \left( \partial B(r), \frac{r}{\sqrt{\theta}} \right) \exp\left\{ -\frac{8T}{\theta^2 C_r S} \right\}$$
$$+ \theta \mathbb{E}\left[ \mathbf{M}_T \left[ \mathscr{F}^\rho \right] \right] + \lambda_T \Psi(f_\star) + r^2.$$

*Proof.* (Sketch). The idea consists in decomposing the expected value according to the lower-isometry event $\mathcal{A}_r$ and its complement: informally, we would write $\mathbb{E} \left[ \text{excess risk} \right] = \mathbb{E} \left[ \text{excess risk} \cap \mathcal{A}_r \right] + \mathbb{E} \left[ \text{excess risk} \cap \mathcal{A}_r^\complement \right]$. The first term would then be bounded thanks to $S$-persistence, $(C, 2)$-hypercontractivity and $B$-boundedness, which allow us to quantify the probability of the lower-isometry event $\mathcal{A}_r$ (see Section F). The bound for the second term is derived along the lines of the proof of Theorem 4.1. The full details are presented in Section H.2. □

Overall, our analysis deploys the concepts of $S$-persistence and $(C, \alpha)$-hypercontractivity to adapt the small-ball argument of Mendelson (2014) to the data-dependent case. Thanks to this construction, we can identify the lower-isometry event, which enables the derivation of our bounds depending on the martingale offset complexity, the ground-truth regularizer $\Psi(f_\star)$ and the critical radius $r$. In the next section, we will characterize the behavior of these terms to obtain the desired convergence rates for physics-informed learning.

## 5 CONVERGENCE RATES

We finally provide our main results in terms of convergence rates for the excess risk $\|\hat{f} - f_\star\|^2_{\mathscr{L}^2((\Omega^T, \mathbb{P}_X; \mathbb{R}^{d_Y}))}$, whose detailed proofs are deferred to Section I. Building upon Theorems 4.1 and 4.2, we also obtain sufficient conditions on the parameter $\lambda_T$ and on the minimal sample size $T$ for the bounds to hold. Throughout this section, we will denote by $d = {2s}/{2s+d_X}$ the Sobolev minimax rate, and $d' = {2d_X}/{2s+d_X}$.

### 5.1 BOUND IN PROBABILITY

**Theorem 5.1.** *Let Assumptions 1 to 6 hold, and let $\hat{f}$ be the solution of (3.3). Fix a probability of failure $\delta \in (0, 1)$, and assume the regularization parameter $\lambda_T$ satisfies*

$$\lambda_T \geq \frac{4}{3T^d} \left[ \frac{C_I \sigma_W^{1+d}}{\Psi(f_\star)^{1-\frac{d'}{4}}} + \frac{(C_{II} + C_{IV})\sigma_W^{2d}}{\Psi(f_\star)^{1-\frac{d'}{2}}} + \frac{C_{III}\sigma_W^2 \log(1/\delta)}{\Psi(f_\star)} \right],$$

*where $C_I, C_{II}, C_{III}$ and $C_{IV}$ are constants depending only on $s, d_X, d_Y$ and $\sqrt{\log(1/\delta)}$. If the number of samples $T$ satisfies*

$$T \geq \frac{\theta^2 C_h S}{8} \left[ C_M \left( \frac{1}{r} \right)^{\frac{6d_X}{2s-d_X}} \log\left( 1 + C_L \left( \frac{1}{r} \right)^{\frac{4s-d_X}{2s-d_X}} \right) + \left( \frac{1}{r} \right)^{\frac{4d_X}{2s-d_X}} \log(1/\delta) \right]$$

*for $r^2 = \lambda_T \Psi(f_\star) + \sigma_W^2/T$ and $C_h, C_M, C_L$ being uniform constants depending on $\rho_f, \overline{\kappa}, \theta, s, d_X$ and $\Omega$, then, with probability at least $1 - 6\delta$, the excess risk enjoys the following convergence rate:*

$$\left\| \hat{f} - f_\star \right\|^2_{\mathscr{L}^2(\Omega^T, \mathbb{P}_X; \mathbb{R}^{d_Y})} \leq C_{slow} \frac{\max\left\{ \Psi(f_\star)^{d'/4}, \Psi(f_\star)^{d'/2} \right\}}{T^d} + C_{fast} \frac{\sigma_W^2 \log(1/\delta)}{T},$$

*where $C_{slow}$ is a constant that depends on $s, d_X, d_Y, \sigma_W^2, \sqrt{\log(1/\delta)}$, and $C_{fast}$ is a constant that depends on $s, d_X, d_Y$.*

*Proof.* (Sketch). The result builds upon the bound in probability on the excess risk of Theorem 4.1, and its crux consists in conveniently setting the values for the critical radius $r$, the radius $\rho$ of the effective hypothesis class $\mathscr{F}^\rho$, and the regularization parameter $\lambda_T$. This allows us to rewrite the excess risk bound (4.3) in terms of the martingale offset complexity, which can in turn be bounded according to (Ziemann, 2022, Theorem 4.2.2) (see Theorem G.2 for its detailed proof). Finally, the characterization of the burn-in follows from the probability of the lower-isometry event. The full proof is reported in Section I.1, where the values of all of the involved constants are given. □

## 5.2 BOUND IN EXPECTATION

**Theorem 5.2.** *Let Assumptions 1 to 6 hold, and let $\hat{f}$ be the solution of* (3.3) *with regularization parameter $\lambda_T$ satisfying*

$$\lambda_T \geq \frac{4(C_I + C_{II})(\sigma_W^2)^d}{3T\Psi(f_\star)^{1-\frac{d'}{2}}},$$

*where $C_I$ and $C_{II}$ are constants depending only on $s, d_X$ and $d_Y$. If $T$ satisfies*

$$T \geq \frac{\theta^2 C_h S}{8}\left(\frac{1}{r}\right)^{\frac{4d_X}{2s-d_X}}\left[C_M\left(\frac{1}{r}\right)^{\frac{2d_X}{2s-d_X}}\log\left(4B^2\left(1+C_L\left(\frac{1}{r}\right)^{\frac{4s-d_X}{2s-d_X}}\right)\right)+\log\left(\frac{\sigma_W^2}{T}\right)\right],$$

*where $B$ is such that $\|f\|_{\mathscr{L}^\infty(\Omega^T;\mathbb{R}^{d_Y})} \leq B$ for all $f \in \mathscr{F}$ and $C_M, C_h, C_L$ are constants depending on $\rho_f, \overline{\kappa}, \theta, s, d_X$ and $\Omega$, then the excess risk enjoys the following convergence rate:*

$$\mathbb{E}\left[\left\|\hat{f}-f_\star\right\|^2_{\mathscr{L}^2(\Omega^T,\mathbb{P}_X;\mathbb{R}^{d_Y})}\right] \leq C_{slow}\frac{\Psi(f_\star)^{d'/2}}{T^d} + C_{fast}\frac{\sigma_W^2}{T},$$

*where $C_{slow}$ and $C_{fast}$ are constants that depend on $s, d_X, d_Y$ and $\sigma_W^2$.*

*Proof.* (Sketch). Similarly to Theorem 5.1, one starts from Theorem 4.2 to set the values for $\rho$ and $\lambda_T$, and then deploys the bound on the expected martingale offset complexity of (Ziemann, 2022, Theorem 3.2.1) (see Theorem G.3 for its detailed proof). Ultimately, the claim is obtained by suitably choosing the critical radius $r$ and accordingly characterizing the lower-isometry event probability, leading to the expression for the burn-in. The detailed proof can be found in Section I.2. □

Notably, our analysis allows us to transfer the contribution of data dependence from the excess risk bound to the burn-in time condition. Moreover, our bounds feature a fast, i.i.d.-like term ($\mathcal{O}(T^{-1})$) and a slower Sobolev rate term ($\mathcal{O}(T^{-d})$) that becomes annihilated when $\Psi(f_\star) \simeq 0$: this proves that, under knowledge alignment, the learning rate speeds up to $\mathcal{O}(T^{-1})$ even if data are dependent.

## 6 RELATED WORK AND DISCUSSION

**General statistical learning framework.** Statistical learning theory (Vapnik, 1998; Cucker & Smale, 2002) offers powerful tools to analyze the theoretical performance of nonparametric learning algorithms. Within such a framework, two main streams to derive learning rates have been developed, as identified by Fischer & Steinwart (2020). The first relies on the spectral analysis of integral operators in reproducing kernel Hilbert spaces (Smale & Zhou, 2007; Caponnetto & De Vito, 2007; Steinwart et al., 2009), while the second builds on empirical process techniques and the small-ball method (Mendelson, 2014; 2018; Lecué & Mendelson, 2017). Our work belongs to the latter stream, adapting the small-ball method to the *dependent-data* case along the lines of the localization analysis of Ziemann & Tu (2022).

**Learning rates for dependent data.** A common approach to handle dependence is through *blocking* techniques (Yu, 1994; Sancetta, 2021), where the trajectory is divided into blocks of length $k$ so that consecutive blocks can be treated as independent. However, this deflates the effective sample size, leading to suboptimal rates. Similar rates appear also in Steinwart & Christmann (2009); Zou et al. (2009); Agarwal & Duchi (2012); Kuznetsov & Mohri (2017), and Nagaraj et al. (2020) shows that such a deflation in a worst-case agnostic model set-up is unavoidable. To contrast this

phenomenon, a significant line of work has studied learning under dependent data *without regularization*. In the linear setting, Simchowitz et al. (2018) and Nagaraj et al. (2020) established sample complexity bounds for system identification and stochastic gradient descent. Moreover, Roy et al. (2021) extended the small-ball method to dependent processes, but without using one-sided concentration, leading to slower rates. Similar slower-rate phenomena also appear in Ziemann et al. (2022). More recently, Ziemann & Tu (2022) proposed an adaptation of the small-ball method and offset complexity technique of Liang et al. (2015) to obtain optimal rates for nonlinear settings. Our work builds upon this line of thought, extending the analysis to *physics-informed regularization*. However, the results in this paper are not a mere adaptation: the physics-informed regularizer introduces additional challenges, such as characterizing the entropy numbers of the effective hypothesis class (e.g., under ellipticity, non-trivial nullspaces of the operator, and boundary conditions), determining trajectory hypercontractivity and working with weighted, vector-valued Sobolev spaces.

**Theoretical analysis of physics-informed machine learning.** Our work belongs to the branch of physics-informed machine learning that aims at enhancing learning algorithms with available physical knowledge — a class of models also known as *hybrid modeling* (Rai & Sahu, 2020; Cuomo et al., 2022; von Rueden et al., 2023b; Hao et al., 2023). To the best of the authors' knowledge, results aimed at quantifying the beneficial impact of physical priors in learning algorithms are von Rueden et al. (2023a) and Doumèche et al. (2024). The present paper is very similar in spirit to the latter work in the way complexity-dependent rates are derived, but crucially deals with non-i.i.d. data and presents bounds for the excess risk not only just in expectation, but also in probability. We further summarize related work in Table 1.

Table 1: Comparison of convergence rates for non-parametric regression with and without regularization. The rate from Ziemann & Tu (2022) follows from its Corollary 4.1 with $q = d_X/s$ under the metric entropy bound $\log \mathcal{N}_\infty(\mathscr{F}, \varepsilon) \sim (1/\varepsilon)^q$. The rate from Lecué & Mendelson (2017) follows from its Lemma 2.1 assuming $r^2(\rho) \sim \sigma_W^2 T^{-1}$, with $\lambda_T \sim T^{-d}$.

| Work | Hypothesis class | Data | Regularization | Assumption | Rate |
|---|---|---|---|---|---|
| Nussbaum (2006) | $\mathscr{L}^2$ Sobolev space | i.i.d. | ✗ | $\sigma_W^2$-Gaussian, $d_X = 1$ | $\sigma_W^2 T^{-2s/(2s+1)}$ |
| Farahmand & Szepesvári (2012) | General Sobolev space | non-i.i.d. | ✗ | Exponential mixing, $d_Y = 1$ | $T^{-d} \log(T)$ |
| Lecué & Mendelson (2017) | General | i.i.d. | Proper regularizer | $\sigma_W^2$-sub-Gaussian, $d_Y = 1$ | $\Psi(f_\star)T^{-d} + \sigma_W^2 T^{-1}$ |
| Ziemann & Tu (2022) | General (not too large) | non-i.i.d. | ✗ | $\sigma_W^2$-sub-Gaussian | $\sigma_W^2 T^{-d}$ |
| Doumèche et al. (2024) | Periodic Sobolev space | i.i.d. | Physics-informed | $\sigma_W^2$-sub-Gamma, $d_Y = 1$ | $\Psi(f_\star)T^{-d} + \sigma_W^2 T^{-1}$ |
| **Our work** | $\mathscr{L}^2$ Sobolev space | non-i.i.d. | Physics-informed | $\sigma_W^2$-sub-Gaussian, $s \geq 2d_X$ | $\Psi(f_\star)^{d/2}T^{-d} + \sigma_W^2 T^{-1}$ |

**Quantifying the impact of knowledge alignment.** We now showcase the impact of knowledge alignment $\Psi(f_\star) \simeq 0$ in contrast with the rates of empirical risk minimization *without regularization*, i.e., considering $\hat{f}'$ as the solution of (3.3) when $\lambda_T = 0$. As shown in detail in Section J, the excess risk for $\hat{f}'$ behaves, both in probability and in expectation, in the following way (informally):

$$\text{(Excess risk)} \quad ||\hat{f}' - f_\star||^2_{\mathscr{L}^2(\Omega^T, \mathbb{P}_X; \mathbb{R}^{d_Y})} \leq \frac{C'_{\text{slow}}}{T^d} + C'_{\text{fast}} \frac{\sigma_W^2}{T}.$$

We can notice how, for the result without regularization, the term decaying according to the Sobolev rate is not modulated by any design parameter (as happened with $\Psi(f_\star)$ in Theorems 5.1 and 5.2), and is thus the dominant term dictating the slow Sobolev convergence rate of the excess risk.

**On the behavior of $\lambda_T$.** It is worth emphasizing that, in both the expectation and probability analyses, the condition on the regularization parameter depends on $1/\Psi(f_\star)^\beta$ for some $\beta > 1$. This condition reflects the well-known regularization-complexity trade-off: as the hypothesis class is restricted (i.e., as $\rho$ becomes small), one must increase $\lambda_T$ to compensate for the reduced richness of the class and the potentially higher sensitivity to noise or variance, as discussed in (Lecué & Mendelson, 2017, Section 2) and also displayed in (Doumèche et al., 2024, Theorem 5.3). Even if such a phenomenon prevents us from considering the case $\Psi(f_\star) = 0$, our bounds still capture the (practical) annihilation of the Sobolev rate term in presence of knowledge alignment. Finally, as pointed out in Doumèche et al. (2024), even if $\lambda_T$ depends on the unknown $\Psi(f_\star)$, it can still be estimated in practice via, e.g., cross-validation (Wahba, 1990).

**On the burn-in condition and the Sobolev order $s$.** In Theorems 5.1 and 5.2, the burn-in time scales as $(1/r)^{6d_X/2s-d_X}$, and $r$ in turn scales as $T^{-1/2}$. Therefore, to ensure well-posedness of the

burn-in time condition, we have to impose that $3d_X/2s-d_X \leq 1$, which yields Assumption 2. Thus, our results come at the price of a stronger requirement on $s$ with respect to the standard $s \geq d_X/2$ needed, e.g., for the Sobolev imbedding theorem (Section B).

**Numerical illustration.** We complement our theoretical analysis with an example showcasing the benefit of prior domain knowledge in learning a nonlinear dynamical system. In this experiment, whose full details can be found in Section K, we consider the dynamics of a unicycle robot described by the differential equations

$$
\begin{cases}
\dot{x}_1(t) = \nu(t) \cos \vartheta(t), \\
\dot{x}_2(t) = \nu(t) \sin \vartheta(t), \\
\dot{\vartheta}(t) = \omega(t),
\end{cases}
$$

where $(x_1, x_2) \in \mathbb{R}^2$ is the position of the robot on the plane, $\vartheta \in [0, \pi/2]$ is the orientation angle, and $(\nu, \omega)$ are the translational and angular velocities, respectively. The physical information we want to incorporate is that the velocity has no lateral component, enforcing the non-slip behavior of the unicycle kinematics. Such a constraint is embedded in the learning problem (3.3) as a (discretized) $\mathcal{L}^2$-regularization term, and we perform estimation by deploying a multilayer perceptron with two hidden layers featuring 64 nodes and ReLU activation functions.

The experiment, whose results are displayed in Figure 2, compares the empirical rates obtained with and without physics-informed regularization. We can notice that both estimators eventually return an accurate model for the ground-truth dynamics. However, without physics knowledge the rate of decay of the estimation error is relatively slow, with an empirical slope of approximately $\mathcal{O}(T^{-0.646})$. In contrast, incorporating physics-informed regularization yields a markedly faster decay, with an empirical slope of approximately $\mathcal{O}(T^{-0.993})$, as the model is explicitly constrained by the domain knowledge that unicycle dynamics do not admit lateral velocity. This experiment demonstrates how embedding physics-based operators into the training objective leads to provable improvements in sample efficiency, consistent with the theory in Section 5 – especially the result in expectation presented in Theorem 5.2.

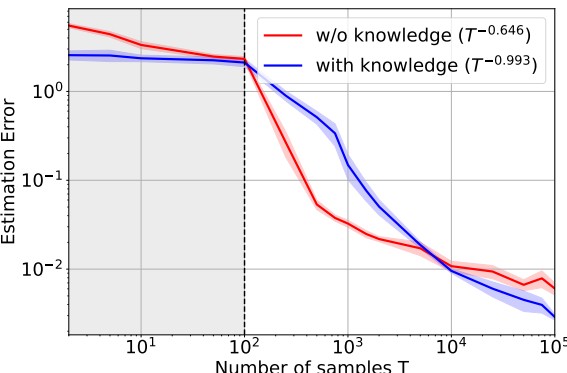

Figure 2: Log-log plot of the empirical excess risk (estimation error) with respect to the number of samples $T$ for the unicycle dynamics. Each curve is obtained by averaging over 20 independent random realizations of the training data, with solid lines indicating the mean estimation error and shaded regions denoting 95% confidence intervals. The gray-shaded area displays the estimated burn-in time, after which the predicted learning rates become observable.

## 7   CONCLUSIONS

This work focused on vector-valued function estimation from dependent data, and studied the excess risk of the estimate $\hat{f}$ obtained through regularized empirical risk minimization, where regularization is induced by physical knowledge (namely, that the unknown function approximately satisfies a partial differential equation). The analysis is set in the general framework of statistical learning, and applies to a wide range of approaches such as kernel-based methods and (deep) neural networks with physics-informed loss functions. The main message of this work is that knowledge alignment (i.e., the regularizer is approximately zero when evaluated at the ground-truth function $f_\star$) allows to speed up the learning rate from the slow, Sobolev rate $\mathcal{O}(T^{-d})$, with $d = {}^{2s}/_{2s+d_X} < 1$, to the fast, optimal i.i.d. one $\mathcal{O}(T^{-1})$. Taken together, our results provide the first convergence rates for physics-informed learning under dependent data that avoid the sample-size deflation inherent to blocking techniques, and reveal a transition from Sobolev minimax rates to fast i.i.d.-optimal rates through knowledge alignment. This bridges classical statistical learning theory, physics-informed regularization, and learning with dependent data.

## ACKNOWLEDGMENTS

Anna Scampicchio acknowledges that this work was partially supported by the Wallenberg AI, Autonomous Systems and Software Program (WASP) funded by the Knut and Alice Wallenberg Foundation. Leonardo F. Toso thanks Ingvar Ziemann for an instructive discussion during the early stage of this work. Leonardo F. Toso is funded by the Center for AI and Responsible Financial Innovation (CAIRFI) Fellowship and by the Columbia Presidential Fellowship. James Anderson is partially funded by NSF grants ECCS 2144634 and 2231350.

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
