# PHYSICS-INFORMED LEARNING UNDER MIXING: HOW PHYSICAL KNOWLEDGE SPEEDS UP LEARNING

**Anna Scampicchio*** [*]
Department of Electrical Engineering
Chalmers University of Technology, Sweden
`anna.scampicchio@chalmers.se`

**Leonardo F. Toso***
Department of Electrical Engineering
Columbia University, USA
`leonardo.toso@columbia.edu`

**Rahel Rickenbach**
Institute for Dynamic Systems and Control
ETH Zürich, Switzerland
`rrahel@ethz.ch`

**James Anderson**
Department of Electrical Engineering
Columbia University , USA
`james.anderson@columbia.edu`

**Melanie N. Zeilinger**
Institute for Dynamic Systems and Control
ETH Zürich, Switzerland
`mzeilinger@ethz.ch`

## ABSTRACT

A major challenge in physics-informed machine learning is to understand how the incorporation of prior domain knowledge affects learning rates when training data is not independent and identically distributed. Focusing on empirical risk minimization with physics-informed regularization, we derive complexity-dependent bounds on the excess risk in probability and in expectation. We prove that, when the physical prior information is aligned, the learning rate improves from the (slow) Sobolev minimax rate to the (fast) optimal i.i.d. one without sample-size deflation due to data dependence.

## 1 INTRODUCTION

Physics-informed machine learning encompasses a wide taxonomy of approaches that combine physical knowledge and learning algorithms to address two main tasks: (i) leveraging data-driven methods to enhance numerical solvability and accuracy of physical models given, for instance, by systems of partial differential equations; (ii) improve learning algorithm performance by including physical information in the loss function or as an additional constraint (Karniadakis et al., 2021; Meng et al., 2025). Focusing on the second class of methods, surveyed in Rai & Sahu (2020); von Rueden et al. (2023b), the resulting approaches turn out to be practically effective in terms of data efficiency, generalization capability and interpretability, which play an important role especially in view of downstream tasks such as safe learning-based control (Nghiem et al., 2023; Drgona et al., 2025). However, despite the empirical success of physics-informed machine learning algorithms (Raissi et al., 2019; Rai & Sahu, 2020; Cuomo et al., 2022; Hao et al., 2023), theoretically quantifying the beneficial impact of incorporating physical information into learning algorithms is technically challenging and still an active area of research (see von Rueden et al. (2023a) and references therein).

We tackle this problem by considering a statistical learning set-up and focusing on *regularized empirical risk minimization* problems of the following form:

$$\hat{f} = \underset{\substack{f \in \text{ball in} \\ \text{Sobolev space}}}{\arg\min} \quad \boxed{\substack{\text{data-fit} \\ \text{squared loss}(f)}} + \lambda_T \boxed{\substack{\text{physics-informed} \\ \text{regularizer}(f)}}, \

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

# • Technical appendix •

In the following sections we provide the derivations of all of the results stated in the paper. This technical appendix is structured as follows:

**Section A** provides the necessary results for the probability set-up of the estimation problem. Specifically, we construct the marginal probability measures stated in Assumption 1 (Section A.1); we discuss the meaning of the $S$-persistence given in Assumption 6 and its relation to data dependence (Section A.2); and derive some useful properties of sub-Gaussian random vectors that will be useful in the martingale offset complexity bounds (Section A.3).

**Section B** constructs the auxiliary set-up of weighted and vector-valued Sobolev spaces needed to define the hypothesis spaces for the empirical risk minimization problem (3.3). First, we extend the Sobolev imbedding theorem to the weighted, vector-valued case (Section B.1). Next, we present definitions and key results of periodic Sobolev spaces (Section B.2), and show that the Sobolev space of interest, $\mathscr{H}^s(\Omega^T, \mathbb{P}_X; \mathbb{R}^{d_Y})$ is imbedded in one of them. Such a construction will be leveraged in the proof of $(C(r), 2)$-hyper-contractivity of $\partial B(r)$ in Section D.2.

**Section C** focuses on bounds for covering numbers of convex sets of vector-valued Sobolev spaces. By deploying the direct-sum structure of the Sobolev space $\mathscr{H}^s(\Omega^T, \mathbb{P}_X; \mathbb{R}^{d_Y})$ elucidated in Section 2.2, we first show how the covering of the multi-dimensional set can be obtained from the covering of the one-dimensional counterpart (Section C.1). We then use such a result to extend classic results on covering numbers, culminating in the bound for the covering number of the effective hypothesis space $\mathscr{F}^\rho$ (Section C.2).

**Section D** shows key properties of the hypothesis spaces $\mathscr{F}$ and $\mathscr{F}^\rho$, such as convexity and $B$-boundedness (Section D.1), and $(C(r), 2)$-hypercontractivity of $\partial B(r)$ (Section D.2). We also show how the trajectory hypercontractivity condition enforces the small-ball property.

**Section E** focuses on the physics-informed regularizer and its properties. In particular, we show that the physics-informed regularizer is 2-proper (Section E.1), and prove an inequality on the difference $\Psi(\hat{f}) - \Psi(f_\star)$ that will be useful in the proofs of Theorems 4.1 and 4.2 (Section E.2).

**Section F** provides the bound for the probability of the lower-isometry event $\mathcal{A}_r$ introduced in Section 4.1. We first show an ancillary inequality linking hypercontractivity and $S$-persistence (Section F.1) and then derive the main lower-isometry bound result (Section F.2), also presenting its corollary that will be useful in characterizing the burn-ins in Section 5.

**Section G** provides the full derivation of the bounds for the martingale offset complexity, which play a key role in the results of Sections 4 and 5. We first show the inequality that underpins the definition of martingale offset complexity (Section G.1), and then prove its bounds, both in probability (Section G.2) and in expectation (Section G.3).

**Section H** contains the proofs of the excess bound rates, namely of Theorem 4.1 (Section H.1) and of Theorem 4.2 (Section H.2).

**Section I** collects the proof of the convergence rates results of Section 5, specifically of Theorem 5.1 (Section I.1) and of Theorem 5.2 (Section I.2).

**Section J** provides the corollaries of the results given in Section 5 dealing with empirical risk minimization without regularization, which we use in the comparison performed in Section 6. Specifically, we derive the result in probability (Section J.1) and in expectation (Section J.2).

**Section K** presents the full details of the numerical experiment set-up outlined in the discussion reported in Section 6 (Section K.1), together with an additional experiment involving the Poisson equation (Section K.2).

## A PROBABILITY MEASURE AND STOCHASTICS SET-UP

This section collects all the ancillary results concerning the probability space $(\Omega^T, \{\mathcal{X}_t\}_{t=0}^{T-1}, \mathbb{P}_X)$, the inter-sample dependence in trajectories $X$ belonging to such a space, and the noise sequence we are considering. In particular, in Section A.1 we specify the marginal distributions presented in Assumption 1; then, in Section A.2 we discuss the $S$-persistence condition in Assumption 6, showing how $S$ quantifies the degree of dependence between data samples separated in time; finally, in Section A.3 we present some useful ancillary results on the second statistical moment of the sub-Gaussian random vectors given in Assumption 3.

### A.1 ON THE CONSTRUCTION OF PROBABILITY MEASURES

We now characterize the probability measures $\mu_t$, defined for each $t = 0, \dots, T-1$, associated with each term of the input trajectory $X$.

The classic set-up involves *independent* samples. In this situation, the $\sigma$-algebra on $\Omega^T$ is given by the tensor product of the single $\sigma$-algebras $\mathcal{X}_t$. Moreover, by construction each component $X_t$ of the trajectory $X$ has a distribution $\mu_t$, and the resulting probability space is $(\Omega^T, \otimes_{t=0}^{T-1} \mathcal{X}_t, \prod_{t=0}^{T-1} \mu_t)$ — see, e.g., (Halmos, 1950, Chapter VII) and (Billingsley, 2012, Section 18).

We now detail the case with *dependent data* building upon the results in (Çinlar, 2011, Chapter I.6). We are in the situation in which the transition between $X_{t-1}$ and $X_t$ for all $t = 1, \dots, T$ is described by a map from $(\Omega, \mathcal{X}_{t-1})$ to $(\Omega, \mathcal{X}_t)$. Such a map is called *transition kernel* $\mathcal{K}_t(\cdot, \cdot) : \Omega \times \mathcal{X}_t \to \mathbb{R}_{\geq 0}$ and is such that $x_{t-1} \mapsto \mathcal{K}_t(x_t, A)$ is $\mathcal{X}_{t-1}$-measurable for every set $A \in \mathcal{X}_t$, and $A \mapsto \mathcal{K}_t(x_{t-1}, A)$ is a measure on $(\Omega, \mathcal{X}_t)$ for every $x_{t-1} \in \Omega$. Before proving the main result in Theorem A.3, we recall two key results:

**Lemma A.1.** *Let $(E, \mathcal{E})$ be a measurable space, and let $L$ be a functional mapping the space of non-negative measurable functions defined on $\mathcal{E}$ to $\mathbb{R}_{\geq 0}$. Then there exists a unique measure $\nu$ on $(E, \mathcal{E})$ such that $L(g) = \nu g$ for any function $g$ in the domain of $L$ if and only if*

1. *$g = 0$ implies $L(g) = 0$;*

2. *for any $\mathfrak{a}, \mathfrak{b} \in \mathbb{R}_{\geq 0}$ and any $g, g'$ in the domain of $L$, $L(\mathfrak{a}g + \mathfrak{b}g') = \mathfrak{a}L(g) + \mathfrak{b}L(g')$;*

3. *for any increasing sequence $\{g_n\}_n \nearrow g$ we have that $L(g_n) \nearrow L(g)$.*

*Proof.* We defer the interested reader to (Çinlar, 2011, Theorem 4.21). $\square$

**Lemma A.2.** *Let $\mathcal{K}_\tau(\cdot, \cdot)$ be a transition kernel from $(\Omega, \mathcal{X}_{\tau-1})$ to $(\Omega, \mathcal{X}_\tau)$, and $\mathcal{K}_{\tau+1}(\cdot, \cdot)$ be a transition kernel from $(\Omega, \mathcal{X}_\tau)$ to $(\Omega, \mathcal{X}_{\tau+1})$. Then, their product is the transition kernel $\mathcal{K}_\tau \mathcal{K}_{\tau+1}$ from $(\Omega, \mathcal{X}_{\tau-1})$ to $(\Omega, \mathcal{X}_{\tau+1})$ such that*

$$\mathcal{K}_\tau \mathcal{K}_{\tau+1}(x_{\tau-1}, A) = \int_\Omega \mathcal{K}_\tau(x_{\tau-1}, dx_\tau) \mathcal{K}_{\tau+1}(x_\tau, A) \quad \text{for } x_{\tau-1} \in \Omega, A \in \mathcal{X}_{\tau+1}.$$

*Proof.* It follows directly from Theorem A.1. $\square$

We are now ready to state the existence and uniqueness of the probability measures associated with each component $X_t$ of the input trajectory $X$.

**Theorem A.3.** *Let $g \colon \Omega \to \mathbb{R}_{\geq 0}$, and assume that there exists a probability measure $\mu_0$ associated with the first component of the input trajectory $X$. Then, for each $t = 1, \dots, T-1$ there exists a unique probability measure such that*

$$\int_{\Omega^T} g(X_t) d\mathbb{P}_X = \int_\Omega g(X_t) d\mu_t.$$

*Proof.* We are considering

$$\int_{\Omega^T} g(X_t) d\mathbb{P}_X = \int_\Omega \mu_0(dX_0) \cdots \int_\Omega \mathcal{K}_t(X_{t-1}, dX_t) g(X_t) \cdots \int_\Omega \mathcal{K}_T(X_{T-1}, dX_T). \quad \text{(A.1)}$$

By an iterative application of Theorem A.2 to Equation (A.1), the contribution of the transition kernels $\mathcal{K}_{t+1}(\cdot,\cdot), \cdots \mathcal{K}_T(\cdot,\cdot)$ integrates to 1. Furthermore, we can apply again Theorem A.2 to the kernels $\mathcal{K}_1(\cdot,\cdot), \cdots, \mathcal{K}_t(\cdot,\cdot)$ and obtain the composed kernel $\bar{\mathcal{K}}(\cdot,\cdot)$ such that (A.1) can be re-written as

$$\int_{\Omega^T} g(X_t) d\mathbb{P}_X = \int_{\Omega} \mu_0(dX_0) \int_X \bar{\mathcal{K}}(X_0, dX_t) g(X_t).$$

It can be shown, along the lines of (Çinlar, 2011, Theorem 6.3), that the right-hand side of the equation above satisfies (A.1), thus proving the claim. □

Note that this theorem holds also for the independent-measures case, where each $\mu_t$ is the $t$-th marginal of $\mathbb{P}_X$ and can be computed relying on Fubini's Theorem.

## A.2 ON $S$-PERSISTENCE AND DATA DEPENDENCE

We now focus on the concept of $S$-persistence (see Assumption 6) and on how it relates to the dependence of the samples in the trajectory $X$.
We first start by recalling the definition of $S$-persistence. The tuple $(\mathscr{F}, \mathbb{P}_X)$ is $S$-persistent if, for every $\xi \geq 0$ and $f \in \mathscr{F}$, we have that

$$\mathbb{E}\left[\exp\left(-\xi \sum_{t=0}^{T-1} \|f(X_t)\|_2^2\right)\right] \leq \exp\left(-\xi \sum_{t=0}^{T-1} \mathbb{E}\left[\|f(X_t)\|_2^2\right] + \frac{\xi^2 S}{2} \sum_{t=0}^{T-1} \mathbb{E}\left[\|f(X_t)\|_2^4\right]\right).$$

The parameter $S$ is related to the "degree of dependence" of the samples within the trajectory $X$: this is typically quantified in terms of the *dependence matrix* (also defined *mixing matrix* in Paulin (2018)), which we now define.

Let $\mathcal{X}_{i:j}$ be the $\sigma$-field generated by the truncated input sequence $\{X_t\}_{t=i}^j$, which we represent as $X_{i:j}$. Additionally, denote with $\|\cdot\|_{TV}$ the total variation norm between two probability measures on $\mathcal{X}_{i:j}$: specifically, such a metric is defined as $\|\nu_1 - \nu_2\|_{TV} \doteq \sup_{A \in \mathcal{X}_{i:j}} |\nu_1(A) - \nu_2(A)|$ for any couple of measures $\nu_1$ and $\nu_2$ defined on the $\sigma$-algebra $\mathcal{X}_{i:j}$. The dependence matrix $\Gamma(\mathbb{P}_X)$ is a lower-triangular matrix whose $(i,j)$ element, for $i, j = 1, \cdots, T$, is given by

$$[\Gamma(\mathbb{P}_X)]_{i,j} \doteq \begin{cases} \sqrt{2 \sup_{A \in \mathcal{X}_{0:T-i}} \left\|\mathbb{P}_{X_{j:T-1}}(\cdot|A) - \mathbb{P}_{X_{j:T-1}}\right\|_{TV}} & (i < j) \\ 1 & (i = j) \\ 0 & (i > j) \end{cases} \quad (A.2)$$

Such a matrix provides a measure of dependence through its induced 2-norm, which is $\|\Gamma(\mathbb{P}_X)\|_2 = \arg\inf_{a>0}\{\|\Gamma(\mathbb{P}_X)v\|_2 \leq a \|v\|_2, v \in \mathbb{R}^T\}$: thus, we have that $S$-persistence is ruled by $S = \|\Gamma(\mathbb{P}_X)v\|_2$. If the stochastic process has independent samples, then it holds that $\|\Gamma(\mathbb{P}_X)\|_2 = 1$; on the other hand, if the process is fully dependent, then we have that $\|\Gamma(\mathbb{P}_X)\|_2$ grows linearly in $T$. We will focus on scenarios in which $\|\Gamma(\mathbb{P}_X)\|_2$ is a constant: as elucidated in (Samson, 2000, Section II), these are the following.

(a) *uniformly ergodic Markov chains*. In these Markov chains, the transition kernels $\mathcal{K}_t(\cdot,\cdot) = \mathcal{K}(\cdot,\cdot)$ are time-homogeneous, and there exists an invariant distribution $\widetilde{\pi}$ such that, for every initial condition $x$, there exists a rate $\mathfrak{r} < 1$ and a constant $\mathfrak{A} > 0$ such that $\left\|\mathcal{K}(x,\cdot)^t - \widetilde{\pi}(\cdot)\right\|_{TV} \leq \mathfrak{A}\mathfrak{r}^t$ (Meyn & Tweedie, 2009, Section 16.2.1). Another characterization of uniformly ergodic Markov chains is given by the Doeblin condition, and in (Doukhan, 1994, Section 2.4, Theorem 1) uniform ergodicity is proven also for non-homogeneous kernels. In general, Markov chains satisfying uniform ergodicity are given, for instance, by linear and stable auto-regressive models $X_{t+1} = FX_t + W_t$: indeed, these are T-chains (Meyn & Tweedie, 2009, Proposition 6.3.5), and the latter that are uniformly ergodic (Meyn & Tweedie, 2009, Theorem 16.2.5). Another notable example is given by nonlinear state-space models $X_{t+1} = F_t(X_t, U_0, \cdots, U_t, W_k)$ of the form presented in (Meyn & Tweedie, 2009, Section 2.2.2) with control model $F_t(U_0, \cdots, U_t)$ that is stable in the sense of Lagrange (Meyn & Tweedie, 2009, Section 16.2.3).

(b) *contracting Markov chains*. A weaker condition imposed on the behavior of the transition kernels is given by Marton (1996), where a weaker form of Doeblin's condition states that $\sup_{x_1,x_2\in\Omega}\|\mathcal{K}_t(x_1,)-\mathcal{K}_t(x_2,)\|_{TV} < 1$ for all $t$. Markov chains satisfying such a condition do not have to be time-homogeneous and are called *contracting* in Marton (1996). As argued in (Samson, 2000, Equation (2.8)), this kind of Markov chain also leads to a $\|\Gamma(\mathbb{P}_X)\|_2$ that does not depend on $T$.

(c) *$\phi$-mixing processes*. A more general way of characterizing ergodicity without assuming the Markov property of the process is given by means of *mixing processes* reviewed, e.g., in Bradley (1986; 2005) and (Doukhan, 1994, Chapter 1). Here the focus is on $\phi$-mixing processes, thoroughly studied in Ibragimov (1962), which can lead to $\|\Gamma(\mathbb{P}_X)\|_2 = \mathcal{O}(1)$. These kind of processes characterize mixing through the measure $\Phi(\mathcal{X}_{0:i},\mathcal{X}_{j:T-1}) \doteq \sup_{A\in\mathcal{X}_{0:i}}\sup_{B\in\mathcal{X}_{j:T-1}}|\mathbb{P}(B|A)-\mathbb{P}(B)|$; additionally, let $\Phi_k(i,j) \doteq \sup_{i,j=0,\dots T-1}\{\Phi(\mathcal{X}_{0:i},\mathcal{X}_{j:T-1}),\ j-i \geq k\}$. A process is $\phi$-mixing if $\Phi_k(i,j) \to 0$ as $k \to \infty$ for all $i,j$: in other words, the measure $\Phi(\cdot,\cdot)$ quantifies how "independent" two non-overlapping blocks of variables in the trajectory $X$ become as their distance increases. Examples of $\phi$-mixing processes are uniformly ergodic Markov processes (Diananda & Bartlett, 1953) and chains of infinite order, which are non-Markovian processes where the transition probability is influenced only slightly by the remote past, that have been deployed to model psychology experiments (Harris, 1955; Bush & Mosteller, 1953; Lamperti & Suppes, 1959).

Noting how $\Phi$ enters the definition of the dependence matrix (A.2), $\|\Gamma(\mathbb{P}_X)\|_2$ can be characterized by $\Phi_k \doteq \sup_{i,j}\Phi_k(i,j)$: if $\Phi_k$ exhibits an exponential decay, or it holds that $\sum_{k=1}^{\infty}\sqrt{\Phi_k} < \infty$, then we have that $\|\Gamma(\mathbb{P}_X)\|_2$ is a constant (Samson, 2000, p.425).

## A.3 USEFUL PROPERTIES OF SUB-GAUSSIAN VECTORS

We now present two lemmas that will be useful in the proofs of the martingale offset complexity bounds in Section G. First, we recall that, according to Assumption 3, the noise sequence is a martingale difference sequence with sub-Gaussian noise. Recalling the definition, we then have, for every $\xi \in \mathbb{R}$ and every $u$ in the unit sphere in $(\mathbb{R}^{d_Y},\|\cdot\|_2)$,

$$\mathbb{E}\left[\exp\left\{\xi\langle W_t,u\rangle_2|\mathcal{X}_{t-1}\right\}\right] \leq \exp\left\{\frac{\xi^2\sigma_W^2}{2}\right\}. \tag{3.2}$$

We now present the lemma on the second moment of a sub-Gaussian random vector.

**Lemma A.4.** *Let $W$ be a sub-Gaussian random vector according to Assumption 3 and let $\mathfrak{A}$ be a positive constant. Then*

$$\mathbb{E}_W\left[\exp\left\{\xi^2\|W\|_2^2\right\}\right] \leq \exp\left\{\mathfrak{A}^2\xi^2\right\} \quad \text{for } |\xi| < \frac{1}{\mathfrak{A}}.$$

*Proof.* The proof for the scalar case can be found in (Vershynin, 2024, Proposition 2.5.2): i.e., when $d_Y = 1$, we obtain that

$$\mathbb{E}_W\left[\exp\left\{\xi^2 W^2\right\}\right] \leq \exp\left\{\mathfrak{A}^2\xi^2\right\} \quad \text{for } |\xi| < \frac{1}{\mathfrak{A}}.$$

To extend the claim to the case $d_Y \geq 1$, we follow the argument in (Ziemann, 2022, Lemma 6.3.4) and consider an auxiliary random vector $V$ that is drawn uniformly over the canonical Euclidean basis of $\mathbb{R}^{d_Y}$, which we denote by $\{e_1,\cdots,e_{d_Y}\}$. With such a construction, we have then that $\mathbb{E}_V[VV^\top] = \frac{1}{d_Y}\mathbb{I}_{d_Y}$, where $\mathbb{I}_{d_Y}$ is the identity matrix in $\mathbb{R}^{d_Y}$. Then, we can consider

$$\begin{aligned}
\mathbb{E}_W\left[\exp\left\{\xi^2\|W\|_2^2\right\}\right] &= \mathbb{E}_W\left[\exp\left\{\xi^2 W^\top\mathbb{I}_{d_Y}W\right\}\right] \\
&= \mathbb{E}_W\left[\exp\left\{\xi^2 d_Y W^\top\mathbb{E}_V[VV^\top]W\right\}\right] \\
&\leq \mathbb{E}_{W,V}\left[\exp\left\{\xi^2 d_Y(V^\top W)^2\right\}\right] \quad \text{by Jensen's inequality,} \\
&\leq \exp\left\{\mathfrak{A}^2 d_Y^2\xi^2\right\} \quad \text{for } |\xi| < \frac{1}{\mathfrak{A}d_Y},
\end{aligned}$$

by applying the result of (Vershynin, 2024, Proposition 2.5.2) on the scalar sub-Gaussian random variable $V^\top W$ conditioned on $V$. $\square$

Next, we derive a similar result for the 2-norm of a sub-Gaussian random vector.

**Lemma A.5.** *Let $W$ be a sub-Gaussian random vector according to Assumption 3. Then*

$$\mathbb{E}_W\left[\|W\|_2\right] \leq 3\sqrt{d_Y}\sigma_W.$$

*Proof.* Again, the claim for the scalar case $d_Y = 1$ can be found in (Vershynin, 2024, Proposition 2.5.2(ii)), where we have that $\mathbb{E}_W\left[\|W\|\right] \leq 3\sigma_W$: the value for the exact constants can be retrieved from the proof using $K_5^2 = \sigma_W^2/2$, $K_1^2 = 2\sigma_W^2$ and $K_2 = (3p)^{1/p}\sqrt{pK_1^2/2}$.

To obtain the result for the general case $d_Y \geq 1$ we proceed similarly to Theorem A.4 and work with the auxiliary random vector $V$ that is uniformly drawn from the canonical basis of $\mathbb{R}^{d_Y}$, resulting in

$$\mathbb{E}_W\left[\sqrt{W^\top W}\right] = \mathbb{E}_W\left[\sqrt{d_Y W^\top \mathbb{E}_V\left[VV^\top\right]W}\right] = \sqrt{d_Y}\mathbb{E}_{W,V}\left[|V^\top W|\right] \leq 3\sqrt{d_Y}\sigma_W.$$

$\square$

# B SOBOLEV SPACES

We now provide the essential information on Sobolev spaces needed in this paper. For a deeper treatment on the subject, we defer to the monograph Adams & Fournier (2003), as well as (Evans, 2010, Chapter 5), Brezis (2011), (Grisvard, 2011, Chapter 1), (Renardy & Rogers, 2004, Chapter 7), (Taylor, 2023, Chapter 4). References focused on weighted Sobolev spaces are Kufner (1985); Kilpelainen (1994); Chua (1996); Gol'dshtein & Ukhlov (2009). Finally, results of vector-valued Banach spaces given by direct sums of scalar spaces can be found in (Conway, 2007, Chapter I.6), (Farenick, 2016, Proposition 5.81), (Bowers & Kalton, 2014, Section 3.7).

## B.1 IMBEDDING PROPERTIES

We start with the well-known Sobolev imbedding theorem.

**Preliminary definitions.** A normed space $(\mathcal{H}_a, \|\cdot\|_{\mathcal{H}_a})$ is *imbedded* in another normed space $(\mathcal{H}_b, \|\cdot\|_{\mathcal{H}_b})$ if $\mathcal{H}_a \subseteq \mathcal{H}_b$, and for any $u \in \mathcal{H}_a$ there exists a constant $\mathfrak{C}_b$ such that $\|u\|_{\mathcal{H}_b} \leq \mathfrak{C}_b\|u\|_{\mathcal{H}_a}$. We denote such an imbedding by $\mathcal{H}_a \hookrightarrow \mathcal{H}_b$.

We now define the Banach space of differentiable functions that have continuous partial derivatives up to some order $j \in \mathbb{Z}_{\geq 0}$ for our vector-valued set-up. We will consider the space $\mathscr{C}^j\left(\Omega^T; \mathbb{R}^{d_Y}\right)$ describing functions $f\colon \Omega \to \mathbb{R}^{d_Y}$ evaluated along each component of the input trajectory $X$. We endow such a space with the norm

$$\|f\|_{\mathscr{C}^j\left(\Omega^T; \mathbb{R}^{d_Y}\right)} \doteq \sum_{i=1}^{d_Y}\sum_{|\alpha|\leq j}\frac{1}{T}\sum_{t=0}^{T-1}\sup_{X_t\in\Omega}|D^\alpha f_i(X_t)| = \sum_{i=1}^{d_Y}\sum_{|\alpha|\leq j}\sup_{X_t\in\Omega}|D^\alpha f_i(X_t)|.$$

**Main result.** We are now ready to prove the Sobolev imbedding theorem for the vector-valued Sobolev spaces introduced in Section 2.2.

**Theorem B.1** (Sobolev imbedding)**.** *Let Assumption 1 and Assumption 2 hold, and let $j$ be a non-negative integer such that Assumption 2 can be written as $s = \lceil\frac{d_X}{2}\rceil + j$. Then*

*(a)* $\mathscr{H}^s(\Omega^T, \mathbb{P}_X; \mathbb{R}^{d_Y}) \hookrightarrow \mathscr{C}^j(\Omega^T; \mathbb{R}^{d_Y})$;

*(b)* $\mathscr{H}^s(\Omega^T, \mathbb{P}_X; \mathbb{R}^{d_Y}) \hookrightarrow \mathscr{L}^q(\Omega^T, \mathbb{P}_X; \mathbb{R}^{d_Y})$ *for $q \geq 2$.*

*Proof.* Let us first define the order $j$. We can write Assumption 2 as $s = 2d_X + j'$ for some $j' \in \mathbb{Z}_{\geq 0}$. Then it has to hold that $j = 2d_X - \lceil\frac{d_X}{2}\rceil + j' \geq 2d_X - \lceil\frac{d_X}{2}\rceil$.

After noting that (i) with our definition of $\mathscr{L}^2$, we can consider each component of the input trajectory separately; (ii) Assumption 1 ensures that the weighted Sobolev spaces are equivalent to the standard ones (Kufner, 1985); and (iii) the structure of the multi-output function spaces is given in terms of direct sums of scalar ones, we can then consider the claim for $\mathscr{H}^s(\Omega, \mu_t; \mathbb{R}^{d_Y})$ and apply (Adams & Fournier, 2003, Theorem 4.12), to which we defer for a complete proof.

The imbedding theorem in Adams & Fournier (2003) is stated under the assumption that the input domain satisfies the "cone condition" (Adams & Fournier, 2003, Definition 4.6). We now argue that our set-up satisfies it. Since our input domain $\Omega$ is bounded with locally Lipschitz boundary, we have that it satisfies the *strong local Lipschitz condition* (Adams & Fournier, 2003, Definition 4.9). Then, from (Adams & Fournier, 2003, Paragraph 4.11) we have that such a property implies the *uniform cone condition* (Adams & Fournier, 2003, Definition 4.8) (see also (Grisvard, 2011, Theorem 1.2.2.2)), which in turn implies that the cone condition holds. □

As a corollary of the imbedding theorem, one can obtain the following result:

**Corollary B.2** (Berlinet & Thomas-Agnan (2004), Theorem 121). *Under Assumption 1 and Assumption 2, $\mathscr{H}^s(\Omega^T, \mathbb{P}_X; \mathbb{R}^{d_Y})$ is a Reproducing Kernel Hilbert space.*

## B.2 PERIODIC SOBOLEV SPACE $\mathscr{H}_{\text{PER}}^s$

We will now characterize another kind of Sobolev space that will be useful for further analysis. Let us denote $Q_L \doteq [-L, L]^{d_X}$ the hyper-cube in $\mathbb{R}^{d_X}$ that contains the input domain $\Omega$; similarly, we will use $Q_{2L} \doteq [-2L, 2L]^{d_X}$ and $Q_{4L} \doteq [-4L, 4L]^{d_X}$. In the following, we will define and present the main properties of the periodic Sobolev space $\mathscr{H}_{\text{per}}^s(Q_{2L}^T, \mu_\lambda^T; \mathbb{R}^{d_Y})$, i.e., considering the Lebesgue measure on $\Omega$ instead of the marginals $\mu_t$ for each $t = 0, \cdots, T-1$, to simplify the presentation. After that, we will leverage Assumption 1 to handle the general probability measure $\mathbb{P}_X$ and derive the results for $\mathscr{H}_{\text{per}}^s(Q_{2L}^T, \mathbb{P}_X; \mathbb{R}^{d_Y})$.

The material in this subsection presents a generalization of the material of (Doumèche et al., 2024, Appendix A). Additional references on periodic Sobolev are Temam (1995); Iorio & Iorio (2001); Berlinet & Thomas-Agnan (2004); Penteker (2015); Gwinner & Stephan (2018); Bogachev & Smolyanov (2020).

**Preliminary definitions.** Given a point $x = (x_1, \cdots, x_{d_X})$ and a function $f: Q_{2L} \to \mathbb{R}^{d_Y}$, its *periodic extension* $E_{\text{per}}(f)(x)$ is the operator mapping $\mathscr{L}^2(Q_{2L}, \mu_\lambda^T; \mathbb{R}^{d_Y})$ to $\mathscr{L}^2(Q_{4L}, \mu_\lambda^T; \mathbb{R}^{d_Y})$; considering $f = (f_1, \cdots, f_{d_Y})$, the periodic extension acts component-wise on $f$ as $[E_{\text{per}}(f)(x)]_i \doteq f_i(x_1 - 4L\lfloor \frac{x_1}{4L} \rfloor, \cdots, x_{d_X}) - 4L\lfloor \frac{x_{d_X}}{4L} \rfloor)$. Such an operator allows us to define the *periodic Sobolev space* $\mathscr{H}_{\text{per}}^s(Q_{2L}^T, \mu_\lambda^T; \mathbb{R}^{d_Y})$ as the space of functions such that $E_{\text{per}}(f)(\cdot)$ belongs to $\mathscr{H}^s(Q_{4L}^T, \mu_\lambda^T; \mathbb{R}^{d_Y})$. Therefore, $\mathscr{H}_{\text{per}}^s(Q_{2L}^T, \mu_\lambda^T; \mathbb{R}^{d_Y})$ is a subspace of $\mathscr{H}^s(Q_{2L}^T, \mu_\lambda^T; \mathbb{R}^{d_Y})$ consisting of functions whose $4L$-periodic extension is still $s$-times differentiable.

**Fourier characterization of** $\mathscr{H}_{\text{per}}^s(Q_{2L}^T, \mu_\lambda^T; \mathbb{R}^{d_Y})$**.** A more practical representation of $\mathscr{H}_{\text{per}}^s(Q_{2L}^T, \mu_\lambda^T; \mathbb{R}^{d_Y})$ is given by means of Fourier basis: indeed, for each component $f_i(\cdot)$ in $f(\cdot) = [f_1(\cdot), \cdots, f_{d_Y}(\cdot)]^\top$ there exists a unique (infinite-dimensional) vector $z_i$ indexed by $\mathbb{Z}^{d_X}$ with components in $\mathbb{C}$ (thus, $z \in \mathbb{C}^{\mathbb{Z}^{d_X}}$) such that we have

$$f_i(x) = \sum_{k \in \mathbb{Z}^{d_X}} z_{i,k} \exp\left\{ \iota \frac{\pi}{2L} \langle k, x \rangle_2 \right\}, \text{ and} \tag{B.1a}$$

$$D^\alpha f_i(x) = \left( \iota \frac{\pi}{2L} \right)^{|\alpha|} \sum_{k \in \mathbb{Z}^{d_X}} z_{i,k} \exp\left\{ \iota \frac{\pi}{2L} \langle k, x \rangle_2 \right\} \prod_{j=1}^{d_X} k_j^{\alpha_j} \tag{B.1b}$$

for any multi-index $\alpha = (\alpha_1, \cdots, \alpha_{d_X}) \in \mathbb{Z}_{\geq 0}^{d_X}$ such that $|\alpha| \leq s$ (Doumèche et al., 2024, Proposition A.5)). Therefore, we obtain the following result.

**Proposition B.3.** *The periodic Sobolev space $\mathscr{H}^s\left(Q_{2L}^T, \mu_\lambda^T; \mathbb{R}^{d_Y}\right)$ can be characterized as*

$$\mathscr{H}_{per}^s\left(Q_{2L}^T, \mu_\lambda^T; \mathbb{R}^{d_Y}\right) = \left\{z \in \bigoplus_{i=1}^{d_Y} \mathbb{C}^{\mathbb{Z}^{d_X}} \ \Big| \ \sum_{i=1}^{d_Y} \sum_{k \in \mathbb{Z}^{d_X}} |z_{i,k}|^2 \, \|k\|_2^{2s} < \infty, \ z_{i,-k} = z_{i,k}^*\right\}.$$

*Proof.* We specify the expression for the Sobolev norm: the rest of the claim follows directly by (Doumèche et al., 2024, Proposition A.5).
By simply considering the norm in $\mathscr{H}^s(Q_{2L}, \mu_\lambda^T; \mathbb{R}^{d_Y})$, we obtain by using (B.1) that

$$\|f\|_{\mathscr{H}^s(Q_{2L}, \mu_\lambda^T; \mathbb{R}^{d_Y})}^2 = \sum_{i=1}^{d_Y} \sum_{k \in \mathbb{Z}^{d_X}} |z_{i,k}|^2 \sum_{|\alpha| \leq s} \left(\frac{\pi}{2L}\right)^{2|\alpha|} \prod_{j=1}^{d_X} k_j^{2\alpha_j}.$$

We now show that such a norm is equivalent to $\sum_{i=1}^{d_Y} \sum_{k \in \mathbb{Z}^{d_X}} |z_{i,k}|^2 \, \|k\|_2^{2s}$. First, neglecting the sums over $i$ and $k$, we can find a pair of constants $0 < \underline{\mathfrak{a}} < \overline{\mathfrak{a}}$ such that $\underline{\mathfrak{a}} \sum_{|\alpha| \leq s} \prod_{j=1}^{d_X} k_j^{2\alpha_j} \leq \sum_{|\alpha| \leq s} \left(\frac{\pi}{2L}\right)^{2|\alpha|} \prod_{j=1}^{d_X} k_j^{2\alpha_j} \leq \overline{\mathfrak{a}} \sum_{|\alpha| \leq s} \prod_{j=1}^{d_X} k_j^{2\alpha_j}$, so we can focus on the term $\sum_{|\alpha| \leq s} \prod_{j=1}^{d_X} k_j^{2\alpha_j}$. By an application of the multinomial theorem (see also Royer (2020)), we can find constants $0 < \underline{\mathfrak{b}} < \overline{\mathfrak{b}}$ such that $\underline{\mathfrak{b}} \sum_{|\alpha| \leq s} \prod_{j=1}^{d_X} k_j^{2\alpha_j} \leq (1 + \|k\|_2^2)^s \leq \overline{\mathfrak{b}} \sum_{|\alpha| \leq s} \prod_{j=1}^{d_X} k_j^{2\alpha_j}$. Finally, we can find another pair of non-negative constants $\underline{\mathfrak{c}} < \overline{\mathfrak{c}}$, again bounded away from zero, such that $\underline{\mathfrak{c}} \|k\|_2^{2s} \leq (1 + \|k\|_2^2)^s \leq \overline{\mathfrak{c}} \|k\|_2^{2s}$ (for instance, $\underline{\mathfrak{c}} = 1$ will do, and $\overline{\mathfrak{c}}$ can be found by upper-bounding the formula of the binomial theorem) – see also (Temam, 1995, Chapter 2.1). $\square$

Finally, we also point out that the characterization of Theorem B.3 can be also more conveniently rewritten using a re-indexing of $z$: indeed, by (Doumèche et al., 2024, Proposition A.7), there exists a one-to-one mapping $\ell \in \mathbb{N} \mapsto k(\ell) \in \mathbb{Z}^{d_X}$ such that we can write $\phi_\ell(x) \doteq \exp\{\iota\pi \langle k(\ell), x\rangle_2 /(2L)\}$. With this, we can express each component $f_i(x)$, for $i = 1, \cdots, d_Y$, as $f_i(x) = \sum_{\ell \in \mathbb{N}} z_{i,\ell} \phi_\ell(x)$. Thus, the space is characterized as follows:

$$\mathscr{H}_{per}^s\left(Q_{2L}^T, \mu_\lambda^T; \mathbb{R}^{d_Y}\right) = \left\{z \in \bigoplus_{i=1}^{d_Y} \mathbb{C}^{\mathbb{Z}^{d_X}} \ \Big| \ \sum_{i=1}^{d_Y} \sum_{\ell \in \mathbb{N}} |z_{i,\ell}|^2 \ell^{2s/d_X} < \infty\right\}. \tag{B.2}$$

**Extension results for $\mathscr{H}^s(\Omega^T, \mu_\lambda^T; \mathbb{R}^{d_Y})$.** Along the lines of the analysis carried out in (Doumèche et al., 2024, Proposition A.6), it is possible to show that the characterization given in Theorem B.3 holds also for $\mathscr{H}^s(\Omega^T, \mu_\lambda^T; \mathbb{R}^{d_Y})$. Indeed, one can use the Sobolev extension Theorem (Stein, 1970, Chapter VI) to linearly extend every function $f$ in $\mathscr{H}^s(\Omega^T, \mu_\lambda^T; \mathbb{R}^{d_Y})$ to the function $\tilde{E}(f)(x)$, whose $i$-th component is given by $[\tilde{E}(f)(x)]_i = \sum_{k \in \mathbb{Z}^{d_X}} z_{i,k} \exp\{\iota\frac{\pi}{2L} \langle k, x\rangle_2\}$. Thus, $\tilde{E}(f)(x)$ belongs to $\mathscr{H}_{per}^s(Q_{2L}^T, \mu_\lambda^T; \mathbb{R}^{d_Y})$ and is such that $\|\tilde{E}(f)(\cdot)\|_{\mathscr{H}_{per}^s(Q_{2L}^T, \mu_\lambda^T; R^{d_Y})}^2 \leq \mathfrak{C}_{s,\Omega} \|f\|_{\mathscr{H}^s(\Omega^T, \mu_\lambda^T; R^{d_Y})}^2$ for some constant $\mathfrak{C}_{s,\Omega}$, yielding the imbedding $\mathscr{H}^s(\Omega^T, \mu_\lambda^T; R^{d_Y}) \hookrightarrow \mathscr{H}_{per}^s(Q_{2L}^T, \mu_\lambda^T; R^{d_Y})$. This allows us to leverage the structure of the periodic Sobolev space to derive a key property of $\mathscr{H}^s(\Omega^T, \mu_\lambda^T; R^{d_Y})$, namely the trajectory $(C, 2)$-hypercontractivity (see Section D.2).

**From Lebesgue measure to $\mathbb{P}_X$.** In the preceding paragraphs we focused on the Lebesgue measure defined on $\Omega$. In such a set-up, it holds that $\|f\|_{\mathscr{L}^2(\Omega^T, \mu_\lambda^T; \mathbb{R}^{d_Y})}^2 = \|f\|_{\mathscr{L}^2(\Omega, \mu_\lambda^T; \mathbb{R}^{d_Y})}^2$, and in such a space the functions $\exp\{\iota\pi \langle k, x\rangle_2 /(2L)\}$ are orthonormal: this enables the application of Parseval's Theorem, leading to the characterization in Theorem B.3. Such a property gets lost as soon we consider the distribution $\mathbb{P}_X$; nevertheless, thanks to Assumption 1, we obtain that $\mathscr{H}^s(\Omega^T, \mathbb{P}_X; \mathbb{R}^{d_Y}) \hookrightarrow \mathscr{H}^s(\Omega^T, \mu_\lambda^T; \mathbb{R}^{d_Y})$; therefore, when needed, analysis can be carried out in $\mathscr{H}^s(\Omega^T, \mu_\lambda^T; \mathbb{R}^{d_Y})$ and the results carry over to the space of interest $\mathscr{H}^s(\Omega^T, \mathbb{P}_X; \mathbb{R}^{d_Y})$.

## C   On covering numbers

This section focuses on the complexity measure for the hypothesis space, which plays a crucial role in the excess risk bounds derived in this paper. After recalling the definition of covering number and metric entropy, we present how classical results, stated for scalar function spaces, extend to our vector-valued set-up. This section culminates with the derivation of the covering number of the effective hypothesis space $\mathscr{F}^\rho$.

### C.1   From scalar to vector-valued hypothesis spaces

To quantify the complexity of a certain hypothesis space $\mathcal{H}$, we will resort to its *covering number*, which is defined as follows:

*Definition* C.1. Let $\mathcal{S}$ be a subset of a metric space $\mathcal{H}_\mathfrak{a}$ with distance function induced by its norm $\|\cdot\|_{\mathcal{H}_\mathfrak{a}}$. The $\varepsilon$-*cover* of $\mathcal{S}$ is a set $\{f^1, \cdots, f^N\} \subset \mathcal{S}$ such that, for each $f \in \mathcal{S}$, there exists some $\ell = 1, \ldots, N$ such that $\|f - f^\ell\|_{\mathcal{H}_\mathfrak{a}} \leq \varepsilon$. The $\varepsilon$-*covering number*, denoted by $\mathcal{N}_\mathfrak{a}(\mathcal{S}, \varepsilon)$, is the cardinality of the smallest $\varepsilon$-cover.

Additionally, $\log \mathcal{N}_\mathfrak{a}(\mathcal{S}, \varepsilon)$ is called *metric entropy* of $\mathcal{S}$ at resolution $\varepsilon$.

There is a vast literature on bounds for covering numbers: see, e.g., (Cucker & Zhou, 2007, Chapter 5), (Wainwright, 2019, Chapter 5), as well as Zhou (2002); Guo et al. (2002); Zhou (2003); Wang et al. (2009). However, results are typically presented for *scalar* function spaces. In this section, we will adapt covering number estimates to our set-up involving multi-output function spaces.

**Proposition C.1.** *Let us consider the metric* $\|\cdot\|_{\mathscr{L}^\infty(\Omega^T; \mathbb{R}^{d_Y})}$, *and let* $\overline{\mathcal{H}} = \bigoplus_{i=1}^{d_Y} \mathcal{H}$ *be a subset of the Sobolev space* $\mathscr{H}^s(\Omega^T, \mathbb{P}_X; \mathbb{R}^{d_Y})$. *Assume there exists a* $\varepsilon/d_Y$-*cover of* $\mathcal{H}$ *in the direct sum with the metric of* $\mathscr{L}^\infty(\Omega^T; \mathbb{R})$, *and let* $\mathcal{N}_\infty(\mathcal{H}, \varepsilon/d_Y)$ *be its covering number: then it holds that*

$$\mathcal{N}_\infty\left(\overline{\mathcal{H}}, \varepsilon\right) \leq \left(\mathcal{N}_\infty\left(\mathcal{H}, \varepsilon/d_Y\right)\right)^{d_Y}.$$

*Proof.* For it to be a $\varepsilon$-cover of $\overline{\mathcal{H}}$, it has to hold that, for every $f \in \overline{\mathcal{H}}$ there exists an element $f'$ in the cover such that $\|f - f'\|_{\mathscr{L}^\infty(\Omega^T; \mathbb{R}^{d_Y})} \leq \varepsilon$. Similarly, assume there is a $\breve{\varepsilon}$-cover of $\mathcal{H}$ such that, for any arbitrary element $h \in \mathcal{H}$, we have a function $h'$ in the cover such that $\|h - h'\|_{\mathscr{L}^\infty(\Omega^T; \mathbb{R})} \leq \breve{\varepsilon}$. Now, by the definition of the vector-valued version of the infinity norm (see Section 2.2), we have

$$\|f - f'\|_{\mathscr{L}^\infty(\Omega^T; \mathbb{R}^{d_Y})} = \sup_{x \in \Omega} \|f(x) - f'(x)\|_2 \leq \sum_{i=1}^{d_Y} \sup_{x \in \Omega} |f_i(x) - f_i'(x)| \leq d_Y \breve{\varepsilon}.$$

Thus, letting $\breve{\varepsilon} = \varepsilon/d_Y$, we obtain an $\varepsilon$-approximation for the covering of $\overline{\mathcal{H}}$. The claim follows by observing that, by construction of the direct sum of scalar hypothesis spaces, the cover of $\overline{\mathcal{H}}$ is given by the Cartesian product of the covers of $\mathcal{H}$. $\quad\square$

### C.2   Covering numbers for vector-valued hypothesis spaces

We conclude this section by adapting standard covering number bounds for scalar function spaces to vector-valued ones. We start from (Cucker & Zhou, 2007, Theorem 5.3).

**Proposition C.2.** *Let* $\overline{\mathcal{H}} = \bigoplus_{i=1}^{d_Y} \mathcal{H}$ *be a finite-dimensional Banach space of dimension $E$, and let* $\mathcal{B}_R$ *be the set such that* $\mathcal{B}_R = \{f \in \overline{\mathcal{H}} \mid \|f\|_{\overline{\mathcal{H}}} \leq R\}$. *Then it holds that*

$$\mathcal{N}_{\overline{\mathcal{H}}}(\mathcal{B}_R, \varepsilon) \leq \left(\frac{2Rd_Y}{\varepsilon} + 1\right)^{E \cdot d_Y}.$$

*Proof.* This result follows by taking (Cucker & Zhou, 2007, Theorem 5.3) and extending it according to the construction in Theorem C.1. $\quad\square$

We now proceed by characterizing the covering number for a ball in the Sobolev space $\mathscr{H}^s(\Omega^T, \mathbb{P}_X; \mathbb{R}^{d_Y})$.

**Lemma C.3.** *Let $\mathcal{B}_R$ be a ball of radius $R$ in the Sobolev space $\mathscr{H}^s(\Omega^T, \mathbb{P}_X; \mathbb{R}^{d_Y})$ satisfying Assumption 2, where $\Omega$ is a domain with locally-Lipschitz boundary. Then the metric entropy of $\mathcal{B}_R$ satisfies*

$$\log \mathcal{N}_\infty (\mathcal{B}_R, \varepsilon) \leq C_c' d_Y^{\frac{2s+d_X}{2s}} \left( \frac{R}{\varepsilon} \right)^{\frac{d_X}{s}}.$$

*Proof.* Let us start from the *scalar-valued* Sobolev space $\mathscr{H}^s(\Omega^T, \mathbb{P}_X; \mathbb{R})$ and let $\varsigma_R$ be its ball of radius $R$. If the input domain $\Omega$ is smooth, then (Cucker & Smale, 2002, Chapter I, Section 6, Proposition 6) claims that, for some positive constant $c$, we have

$$\log \mathcal{N}_\infty (\varsigma_R, \varepsilon) \leq \left( \frac{cR}{\varepsilon} \right)^{\frac{d_X}{s}} + 1 \leq C_c' \left( \frac{R}{\varepsilon} \right)^{\frac{d_X}{s}} \tag{C.1}$$

for some other constant $C_c'$ that is big enough to absorb also the contribution of the "+1" in (C.1).

Such a result relies on a bound on the entropy number of the embedding $\mathscr{H}^s(\Omega^T, \mathbb{P}_X; \mathbb{R}) \hookrightarrow \mathscr{L}^\infty(\Omega^T; \mathbb{R})$ given by (Edmunds & Triebel, 1996, Section 3.3) (using their notation, we are looking at $F_{2,q_1}^s \hookrightarrow F_{\infty,q_2}^0$). However, in (Edmunds & Triebel, 1996, Section 3.5), such a result is extended to non-smooth domains — specifically, to the *minimally regular* ones (Edmunds & Triebel, 1996, Section 2.5, Definition 2), which include domains with locally-Lipschitz boundary as special cases. By the way, the theorem in (Edmunds & Triebel, 1996, Section 3.5) is stated for the function spaces $B_{pq}^s$, but the results holds also for the spaces $F_{pq}^s$: see the argument presented in the proof of the Theorem in (Edmunds & Triebel, 1996, Section 3.3.2).

Thus, overall, (C.1) holds also for our choice of $\Omega$. The proof is concluded by invoking Theorem C.1 to extend (C.1) to the vector-valued case. Specifically, we have that $\mathcal{N}_\infty (\mathcal{B}_R, \varepsilon) \leq (\mathcal{N}_\infty (\varsigma_R, \varepsilon/d_Y))^{d_Y}$, and taking logarithms we obtain $\log \mathcal{N}_\infty (\mathcal{B}_R, \varepsilon) \leq d_Y \log \mathcal{N}_\infty (\varsigma_R, \varepsilon/d_Y)$, and substituting (C.1) leads to the final claim. $\square$

We conclude this section by deriving a bound for the metric entropy of the effective hypothesis space $\mathscr{F}^\rho$ presented in (3.7). The idea is to leverage the structure of the differential operator (Assumption 4) and find a ball in the Sobolev space that approximates $\mathscr{F}^\rho$, and then invoke Theorem C.3 to bound its metric entropy.

**Proposition C.4.** *Let Assumptions 2 and 4 hold and consider the effective hypothesis space*

$$\mathscr{F}^\rho = \left\{ f \in \mathscr{F}_\star \,|\, \|\mathscr{D}(f)\|_{\mathscr{L}^2(\Omega^T, \mathbb{P}_X; \mathbb{R}^{d_Y})}^2 \leq \rho \right\}. \tag{3.7}$$

*Then, for some positive constant $C_c$ not depending on $\varepsilon$ and $\rho$, we have that*

$$\log \mathcal{N}_\infty (\mathscr{F}^\rho, \varepsilon) \leq C_c d_Y^{\frac{2s+d_X}{2s}} \left( \frac{\sqrt{\rho}}{\varepsilon} \right)^{\frac{d_X}{s}}.$$

*Proof.* Similarly to Theorem C.3, we prove the result for scalar-valued function spaces and then invoke Theorem C.1 to obtain the claim for the vector-valued case.

We start by recalling an important property of elliptic operators derived from (Evans, 2010, Chapter 6.3, Theorem 5) that will allow us to find the Sobolev ball centered at $f_\star$ containing $\mathscr{F}^\rho$. Specifically, for some positive constant $C_e$ and $f \in \mathscr{F}_\star$, we have that

$$\|f\|_{\mathscr{H}^s(\Omega^T, \mathbb{P}_X; \mathbb{R}^{d_Y})} \leq C_e \left( \|\mathscr{D}(f)\|_{\mathscr{L}^2(\Omega^T, \mathbb{P}_X; \mathbb{R}^{d_Y})} + \|f\|_{\mathscr{L}^2(\Omega^T, \mathbb{P}_X; \mathbb{R}^{d_Y})} \right). \tag{C.2}$$

Let $\mathcal{H}$ be the scalar-valued version of $\mathscr{F}^\rho$. We decompose $\mathcal{H}$ into the null-space of $\mathscr{D}$ and its orthogonal complement, obtaining $\mathcal{H} = \ker(\mathscr{D}) \oplus \ker(\mathscr{D})^\perp$. Accordingly, any $f \in \mathcal{H}$ can be written as $f = g + h$, with $g \in \ker(\mathscr{D})$ and $h \in \ker(\mathscr{D})^\perp$, and the constraint $\|\mathscr{D}(f)\|_{\mathscr{L}^2(\Omega^T, \mathbb{P}_X; \mathbb{R})}^2 \leq \rho$ reduces to $\|\mathscr{D}(h)\|_{\mathscr{L}^2(\Omega^T, \mathbb{P}_X; \mathbb{R})}^2 \leq \rho$.

We first focus on the subspace $\ker(\mathscr{D})^\perp$ and prove a preliminary result that allows us to rewrite (C.2). Specifically, we want to show that, for some positive constant $C_l$ and any $h \in \ker(\mathscr{D}(f))^\perp$,

$$\|h\|_{\mathscr{L}^2(\Omega^T, \mathbb{P}_X; \mathbb{R})} \leq C_l \|\mathscr{D}(h)\|_{\mathscr{L}^2(\Omega^T, \mathbb{P}_X; \mathbb{R})}. \tag{C.3}$$

We show this by contradiction. If the claim were false, then we could find a sequence $\{h_k\}_k$ in $\ker(\mathscr{D}(f))^{\perp}$ such that $\|h_k\|_{\mathscr{L}^2(\Omega^T, \mathbb{P}_X; \mathbb{R})} = 1$ and $\|\mathscr{D}(h_k)\|_{\mathscr{L}^2(\Omega^T, \mathbb{P}_X; \mathbb{R})} = 1/k$. By (C.2), the sequence $\{h_k\}_k$ is uniformly bounded, which implies that there exists a subsequence $\{h_{k_\ell}\}_\ell \subset \{h_k\}_k$ that converges weakly to some $h \in \ker(\mathscr{D})^{\perp}$ (Evans, 2010, Appendix D, Theorem 3) (see also (Adams & Fournier, 2003, Theorem 3.6)). However, this implies that $\mathscr{D}(h_{k_\ell}) \to \mathscr{D}(h) = 0$, and $h \neq 0$ because $\|h_k\|_{\mathscr{L}^2(\Omega^T, \mathbb{P}_X; \mathbb{R})} = 1$ by construction. Hence, we would obtain that $h \in \ker(\mathscr{D})$, which is absurd.

Thanks to (C.3), we can re-write (C.2) for $h \in \ker(\mathscr{D})^{\perp}$ as $\|h\|_{\mathscr{H}^s(\Omega^T, \mathbb{P}_X; \mathbb{R})} \leq C_e(C_l + 1)\sqrt{\rho}$, showing that any $h \in \ker(\mathscr{D})^{\perp}$ is contained in a ball of the Sobolev space of radius proportional to $\sqrt{\rho}$. Finally, we can determine the covering number for $\ker(\mathscr{D})^{\perp}$ by invoking (C.1).

We now proceed by focusing on $\ker(\mathscr{D})$. Ellipticity stated in Assumption 4 implies that such a subspace is finite-dimensional. Additionally, by Theorem B.1 (Sobolev imbedding), there exists a uniform positive constant $\tilde{B}$ such that $\|f\|_{\mathscr{L}^\infty(\Omega^T; \mathbb{R})} \leq \tilde{B}$ for every function $f \in \mathcal{H}$, which is a closed and convex subset of the Sobolev space $\mathscr{H}^s(\Omega^T, \mathbb{P}_X; \mathbb{R})$ — thus, also $\|g\|_{\mathscr{L}^\infty(\Omega^T; \mathbb{R})} \leq \tilde{B}$. In light of these considerations, $\ker(\mathscr{D})$ belongs to a ball of radius $\tilde{B}$ of a finite-dimensional Euclidean space with dimension $\dim \ker(\mathscr{D})$, and its covering number can be calculated according to (Cucker & Zhou, 2007, Theorem 5.3).

At this point, since the decomposition of $\mathcal{H}$ into $\ker(\mathscr{D})$ and $\ker(\mathscr{D})^{\perp}$ is orthogonal, the covering number of $\mathcal{H}$ is given by the product of the covering numbers of the two subspaces. Taking logarithms, we obtain, for sufficiently large constants $\mathfrak{c}$ and $C_c$,

$$\log \mathcal{N}_\infty (\mathcal{H}, \varepsilon) \leq \mathfrak{c} \left( \left( \frac{\sqrt{\rho}}{\varepsilon} \right)^{\frac{d_X}{s}} + \dim \ker(\mathscr{D}) \log \left( \frac{\tilde{B}}{\varepsilon} \right) \right) \leq C_c \left( \frac{\sqrt{\rho}}{\varepsilon} \right)^{\frac{d_X}{s}}, \tag{C.4}$$

where the contribution of $\ker(\mathscr{D})$ is incorporated in $C_c$ as the logarithmic term is negligible for small $\varepsilon$.

To conclude the proof, we proceed along the lines of the proof for Theorem C.3 and obtain the final claim by invoking Theorem C.1. $\qquad\square$

# D PROPERTIES OF THE HYPOTHESIS SPACES

We now demonstrate some useful properties of our hypothesis spaces $\mathscr{F}$ and $\mathscr{F}^\rho$. We will start by focusing on convexity of the spaces and on the boundedness of the functions that belong to them; next, we proceed by showing that our effective hypothesis space $\mathscr{F}^\rho$ satisfies the small-ball condition introduced in Mendelson (2014) at least on a subset of interest for the following proofs.

## D.1 CONVEXITY AND $B$-BOUNDEDNESS

**Lemma D.1.** *The hypothesis space $\mathscr{F}$ presented in (3.6) is convex – i.e., for any $0 \leq \xi \leq 1$ and any $f, h \in \mathscr{F}$, we have that $\xi f + (1 - \xi)h$ still belongs to $\mathscr{F}$. Additionally, there exists a constant $B > 0$ such that every $f \in \mathscr{F}$ is $B$-bounded, i.e., $\|f\|_\infty \leq B$ for all $f \in \mathscr{F}$.*

*Proof.* Given an arbitrary Hilbert space $\mathcal{H}$ with norm induced by the inner product $\langle \cdot, \cdot \rangle_{\mathcal{H}}$, the parallelogram law yields

$$\|x - y\|_{\mathcal{H}} + \|x + y\|_{\mathcal{H}} = 2\|x\|_{\mathcal{H}} + 2\|y\|_{\mathcal{H}} \Rightarrow \|x - y\|_{\mathcal{H}} \leq 2\|x\|_{\mathcal{H}} + 2\|y\|_{\mathcal{H}} \tag{D.1}$$

This leads to showing that Hilbert spaces (then, also the Sobolev space $\mathscr{H}^s(\Omega^T, \mathbb{P}_X; \mathbb{R}^{d_Y})$) are uniformly convex (Adams & Fournier, 2003, Definition 1.20 and Theorem 3.5): therefore, the first claim follows by noting that $\mathscr{F}$ is a convex subset of the Sobolev space. The second claim is a consequence of the imbedding presented in Theorem B.1(b): indeed, there exist a constant $\mathfrak{C}_\infty$ such that $\|f\|_{\mathscr{L}^\infty(\Omega^T, \mathbb{P}_X; \mathbb{R}^{d_Y})} \leq \mathfrak{C}_\infty \|f\|_{\mathscr{L}^2(\Omega^T, \mathbb{P}_X; \mathbb{R}^{d_Y})} \leq \mathfrak{C}_\infty \rho_f \doteq B$ by definition of $\mathscr{F}$ in (3.6). $\qquad\square$

## D.2 $(C, 2)$-HYPERCONTRACTIVITY

We start by providing the general definition.

*Definition* D.1. For a given hypothesis space $\mathcal{H}$ of vector-valued functions and uniform constants $C > 0$ and $\alpha \in [1, 2]$, the tuple $(\mathcal{H}, \mathbb{P}_\Omega)$ is $(C, \alpha)$-*hypercontractive* if it holds that, for every $f \in \mathcal{H}$,

$$\mathbb{E}_{\mathbb{P}_X} \left[ \frac{1}{T} \sum_{t=0}^{T-1} \|f(X_t)\|_2^4 \right] \leq C \left( \mathbb{E}_{\mathbb{P}_X} \left[ \frac{1}{T} \sum_{t=0}^{T-1} \|f(X_t)\|_2^2 \right] \right)^\alpha. \tag{D.2}$$

In this paper, we will be focusing on the corner case $\alpha = 2$, which implies that the *small-ball condition* (Mendelson, 2014) holds:

**Lemma D.2.** *Let $(\mathcal{H}, \mathbb{P}_X)$ be $(C, 2)$-hypercontractive for a suitable positive constant $C$. Then it holds that, for any $f, h$ in $\mathcal{H}$, there exist $\varepsilon$ and $\xi$ such that*

$$\mathbb{P}_X \left( \frac{1}{T} \sum_{t=0}^{T-1} \|f(X_t) - h(X_t)\|_2 \geq \xi \|f - h\|_{\mathscr{L}^2(\Omega^T, \mathbb{P}_X; \mathbb{R}^{d_Y})} \right) \geq \varepsilon.$$

*Proof.* First, as pointed out in the discussion in Section 2.2, note that Definition D.1 can be written as $\|f\|_{\mathscr{L}^4(\Omega^T, \mathbb{P}_X; \mathbb{R}^{d_Y})}^4 \leq C \|f\|_{\mathscr{L}^2(\Omega^T, \mathbb{P}_X; \mathbb{R}^{d_Y})}^{2 \cdot 2}$. Next, let $c_{2,4} \doteq \|f - h\|_{\mathscr{L}^2(\Omega^T, \mathbb{P}_X; \mathbb{R}^{d_Y})} / \|f - h\|_{\mathscr{L}^4(\Omega^T, \mathbb{P}_X; \mathbb{R}^{d_Y})}$. By the Paley-Zygmund inequality (De La Peña & Giné, 1999, Corollary 3.3.2), we have that

$$\mathbb{P}_X \left( \frac{1}{T} \sum_{t=0}^{T-1} \|f(X_t) - h(X_t)\|_2 \geq u \|f - h\|_{\mathscr{L}^2(\Omega^T, \mathbb{P}_X; \mathbb{R}^{d_Y})} \right) \geq [(1 - u^2) c_{2,4}^2]^2 \geq (1 - u^2)^2 / C,$$

where the last step follows by Definition D.1. Conclusion follows as soon as we let $u \leftrightarrow \xi$ and $(1 - u^2)^2 / C \leftrightarrow \varepsilon$. $\qquad \square$

Drawing inspiration from (Ziemann, 2022, Proposition 3.4.4), we now prove that the hypothesis space $\mathscr{F}$ satisfies this particular kind of hypercontractivity on a subset of interest for Theorem F.2, which will allow us to quantify the probability of the lower isometry event.

**Theorem D.3.** *Let Assumptions 1, 2, 5 and 6 hold. Given some $r > 0$, consider the set $\partial B(r) = \{f \in \mathscr{F} \mid \|f - f_\star\|_{\mathscr{L}^2(\Omega^T, \mathbb{P}_X; \mathbb{R}^{d_Y})}^2 = r^2\}$ and let $\mathscr{F}_\epsilon$ be its cover with balls of radius $\epsilon$. Furthermore, let $\widetilde{\rho_f} \propto \rho_f / \overline{\kappa}$, where $\overline{\kappa}$ is as per Assumption 1. Then, we have that the covering number of $\partial B(r)$ (in other words, the cardinality of $\mathscr{F}_\epsilon$) satisfies*

$$\mathcal{N}_\infty (\partial B(r), \epsilon) \leq \left( \frac{8 \widetilde{\rho_f} m_\epsilon^{s/d_X} d_Y}{\epsilon} + 1 \right)^{m_\epsilon d_Y} \tag{D.3}$$

*with $m_\epsilon$ being the smallest integer solution of*

$$m \geq \left( \frac{16 \widetilde{\rho_f} d_X}{(2s - d_X) \epsilon^2} \right)^{d_X / (2s - d_X)}.$$

*Additionally, as long as $\epsilon \leq \inf_{f \in \partial B(r / \sqrt{(\overline{\kappa})})} \|f\|_{\mathscr{L}^2(\Omega^T, \mu_\lambda^T; \mathbb{R}^{d_Y})}$, the tuple $(\mathscr{F}_\epsilon, \mathbb{P}_X)$ is $(C(\epsilon), 2)$-hypercontractive, with*

$$C(\epsilon) \propto \left( \frac{\mu_\lambda(\Omega)}{32} + 8 \mu_\lambda(\Omega) m_\epsilon^2 \left( \frac{\mu_\lambda(\Omega)}{8} + 2 \right)^2 \right). \tag{D.4}$$

*Proof.* We will make use of the construction presented in Section B.2 and focus on the Fourier characterization of $\mathscr{H}^s(\Omega^T, \mu_\lambda^T; \mathbb{R}^{d_Y})$: the result carries over to $\mathscr{H}^s(\Omega^T, \mathbb{P}_X; \mathbb{R}^{d_Y})$ by the imbedding $\mathscr{H}^s(\Omega^T, \mathbb{P}_X; \mathbb{R}^{d_Y}) \hookrightarrow \mathscr{H}^s(\Omega^T, \mu_\lambda^T; \mathbb{R}^{d_Y})$. As such, we will consider the space $\mathscr{F}_{\mu_\lambda} \doteq \left\{ f \in \mathscr{H}^s(\Omega^T, \mu_\lambda^T; \mathbb{R}^{d_Y}) \mid \|f\|_{\mathscr{H}^s(\Omega^T, \mu_\lambda^T; \mathbb{R}^{d_Y})}^2 \leq \rho_f^2 / \overline{\kappa} \right\}$ such that $\mathscr{F} \subseteq \mathscr{F}_{\mu_\lambda}$: indeed, the condition on the norm reads as $\overline{\kappa} \|f\|_{\mathscr{H}^s(\Omega^T, \mu_\lambda^T; \mathbb{R}^{d_Y})}^2 \geq \|f\|_{\mathscr{H}^s(\Omega^T, \mathbb{P}_X; \mathbb{R}^{d_Y})}^2$ by Assumption 1. According to the construction in Section B.2 and the Fourier decomposition in terms of the basis functions

$\phi_\ell(x) = \exp\{\iota\pi \langle k(\ell), x\rangle_2 /(2L)\}$, we can leverage (B.2) to write

$$\mathscr{F}_{\mu_\lambda} = \left\{ f(x) = \sum_{\ell\in\mathbb{N}} z_\ell\phi_\ell(x), \ z_\ell \in \mathbb{R}^{d_Y} \ \forall \ell \in \mathbb{N} \ \middle| \ \sum_{\ell\in\mathbb{N}} \frac{\|z_\ell\|_2^2}{\ell^{-2s/d_X}} \leq \widetilde{\rho_f}^2, \right\}, \qquad \text{(D.5)}$$

where $\widetilde{\rho_f}^2$ is equal to $\rho_f^2/\overline{\kappa}$ times a multiplicative constant that can be retrieved along the lines of the proof of Theorem B.3, but we do not specify it because it does not affect the main message of our results. Additionally, we will consider $\widetilde{r}^2 \doteq r^2/\overline{\kappa}$ such that $B(\widetilde{r}) \supset B(r)$. Note that, due to such an inclusion, $\mathcal{N}_\infty \left(\partial B(r), \epsilon\right) \leq \mathcal{N}_\infty \left(\partial B(\widetilde{r}), \epsilon\right)$.

**Covering.** We start by proving the result on the covering of $\partial B(\widetilde{r})$. The idea is to find a suitable finite-dimensional approximation of the space $\mathscr{F}_{\mu_\lambda}$, compute its covering number using Theorem C.2, and then express $\mathcal{N}_\infty \left(\partial B(\widetilde{r}), \epsilon\right)$ in terms of such a cover.
We first seek an approximation of $\mathscr{F}_{\mu_\lambda}$ at resolution $\epsilon/4$ by considering, for some $m \in \mathbb{Z}_{\geq 0}$, the finite-dimensional space

$$\mathscr{F}_{\mu_\lambda}^m = \left\{ f(x) = \sum_{\ell=1}^m z_\ell\phi_\ell(x), \ z_\ell \in \mathbb{R}^{d_Y} \ \middle| \ \sum_{\ell\in\mathbb{N}} \frac{\|z_\ell\|_2^2}{\ell^{-2s/d}} \leq \widetilde{\rho_f}^2 \right\}.$$

Now, fix $f \in \mathscr{F}_{\mu_\lambda}$ with coordinates $\{z_\ell\}_{\ell\in\mathbb{N}}$ and let $f'$ be its projection onto $\mathscr{F}_{\mu_\lambda}^m$. Then the following holds:

$$\|f - f'\|_{\mathscr{L}^\infty(\Omega^T;\mathbb{R}^{d_Y})} = \left\| \sum_{\ell=m+1}^\infty z_\ell\phi_\ell \right\|_{\mathscr{L}^\infty(\Omega^T;\mathbb{R}^{d_Y})}$$

$$\leq \left\| \sqrt{\sum_{\ell=m+1}^\infty \frac{\|z_\ell\|_2^2}{\ell^{-2s/d_X}}} \sqrt{\sum_{\ell=m+1}^\infty \ell^{-2s/d_X}|\phi_\ell|^2} \right\|_{\mathscr{L}^\infty(\Omega^T;\mathbb{R}^{d_Y})}$$

$$\overset{\text{(D.5)}}{\leq} \widetilde{\rho_f} \left\| \sqrt{\sum_{\ell=m+1}^\infty \ell^{-2s/d_X}|\phi_\ell|^2} \right\|_{\mathscr{L}^\infty(\Omega^T;\mathbb{R}^{d_Y})}$$

$$\leq \widetilde{\rho_f} \sqrt{\sum_{\ell=m+1}^\infty \ell^{-2s/d_X} \left\| |\phi_\ell|^2 \right\|_{\mathscr{L}^\infty(\Omega;\mathbb{R})}} \leq \widetilde{\rho_f} \sqrt{\sum_{\ell=m+1}^\infty \ell^{-2s/d_X}}, \qquad \text{(D.6)}$$

where the first inequality is given by the Cauchy-Schwarz one, and the last one follows by definition of the basis $\phi_\ell(\cdot)$. We now use the integral test (Rudin, 1976, Chapter 6, Exercise 8) to upper-bound the last expression. Let $\mathfrak{p} \doteq 2s/d_X$ to simplify notation, and note that $\mathfrak{p} > 1$ by Assumption 2. Then we have that

$$\sum_{\ell=m+1}^\infty \left(\frac{1}{\ell}\right)^{\mathfrak{p}} \leq \sum_{\ell=m+1}^\infty \int_{\ell-1}^\ell \left(\frac{1}{x}\right)^{\mathfrak{p}} dx = \int_m^\infty \left(\frac{1}{x}\right)^{\mathfrak{p}} dx = \frac{1}{\mathfrak{p}-1}\frac{1}{m^{\mathfrak{p}-1}}.$$

Plugging such a bound in (D.6), to ensure that $\|f - f'\|_{\mathscr{L}^\infty(\Omega^T;\mathbb{R}^{d_Y})} \leq \epsilon/4$ for any $f \in \mathscr{F}_{\mu_\lambda}$ we then require that

$$\widetilde{\rho_f}\sqrt{\frac{1}{\mathfrak{p}-1}\frac{1}{m^{\mathfrak{p}-1}}} \leq \epsilon/4 \iff m \geq \left(\frac{16\widetilde{\rho_f}^2 d_X}{(2s-d_X)\epsilon^2}\right)^{\frac{d_X}{2s-d_X}}. \qquad \text{(D.7)}$$

Thus, we take $m_\epsilon$ the smallest integer $m$ satisfying (D.7) to have the approximation of $\mathscr{F}_{\mu_\lambda}$ at resolution $\epsilon/4$.
We can now proceed by constructing the covering for $\mathscr{F}_{\mu_\lambda}^{m_\epsilon}$ at resolution $\epsilon/4$. We start by characterizing such a space in terms of the coefficients $\{z_\ell\}_{\ell=1}^{m_\epsilon}$ by considering

$$\mathcal{Z}^{m_\epsilon} \doteq \left\{ z_\ell \in \mathbb{R}^{d_Y}, \ \ell = 1, \cdots, m_\epsilon \ \middle| \ \sum_{\ell=1}^{m_\epsilon} \frac{\|z_\ell\|_2^2}{\ell^{-2s/d_X}} \leq \widetilde{\rho_f}^2. \right\}.$$

This is a ball of radius $\widetilde{\rho_f}$ in a finite-dimensional Banach space with a weighted 2-norm, which we denote by $\|\cdot\|_w$. With such a norm, by Theorem C.2, its covering number with balls of radius $\bar{\epsilon}$ satisfies

$$\mathcal{N}_w\left(\mathcal{Z}^{m_\epsilon}, \bar{\epsilon}\right) \leq (2\widetilde{\rho_f} d_Y/\bar{\epsilon} + 1)^{m_\epsilon d_Y} \doteq N. \tag{D.8}$$

Now, denote the elements of the optimal covering of $\mathcal{Z}^{m_\epsilon}$ as $\{z_\ell^1, \cdots, z_\ell^N\}_{\ell=1,\cdots,m_\epsilon}$. These identify a further approximation $\mathscr{F}_{\mu_\lambda}^{m_\epsilon, N} \subset \mathscr{F}_{\mu_\lambda}^{m_\epsilon}$ defined as

$$\mathscr{F}_{\mu_\lambda}^{m_\epsilon, N} \doteq \left\{ \sum_{\ell=1}^{m_\epsilon} z_\ell^1 \phi_\ell, \cdots, \sum_{\ell=1}^{m_\epsilon} z_\ell^N \phi_\ell \right\}.$$

We can use this construction to characterize the radius $\bar{\epsilon}$. Let $f'(\cdot) = \sum_{\ell=1}^{m_\epsilon} z_\ell' \phi_\ell(\cdot)$ be an arbitrary function in $\mathscr{F}_{\mu_\lambda}^{m_\epsilon}$. Then we have that

$$\min_{n=1,\cdots,N} \left\| f' - \sum_{\ell=1}^{m_\epsilon} z_\ell^n \phi_\ell \right\|_{\mathscr{L}^\infty(\Omega^T; \mathbb{R}^{d_Y})} = \min_{n=1,\cdots,N} \left\| \sum_{\ell=1}^{m_\epsilon} (z_\ell' - z_\ell^n)\phi_\ell \right\|_{\mathscr{L}^\infty(\Omega^T; \mathbb{R}^{d_Y})}$$

$$\leq \min_{n=1,\cdots,N} \left\| \sqrt{\sum_{\ell=1}^{m_\epsilon} \frac{\|z_\ell' - z_\ell^n\|_2^2}{\ell^{-2s/d_X}}} \sqrt{\sum_{\ell=1}^{m_\epsilon} \ell^{-2s/d_X} |\phi_\ell|^2} \right\|_{\mathscr{L}^\infty(\Omega^T; \mathbb{R}^{d_Y})}$$

$$\overset{(D.8)}{\leq} \bar{\epsilon}\sqrt{\sum_{\ell=1}^{m_\epsilon} \left(\frac{1}{\ell}\right)^{\mathfrak{p}}} \leq \bar{\epsilon} m_\epsilon^{\mathfrak{p}/2}.$$

Therefore, if we take $\bar{\epsilon} \leq \epsilon/(4m_\epsilon^{\mathfrak{p}/2})$ we obtain the $\epsilon/4$-cover of $\mathscr{F}_{\mu_\lambda}^{m_\epsilon}$ we seek. This leads to the claim that the covering number of $\mathscr{F}_{\mu_\lambda}^{m_\epsilon}$ satisfies

$$\mathcal{N}_\infty\left(\mathscr{F}_{\mu_\lambda}^{m_\epsilon}, \frac{\epsilon}{4}\right) \leq \left(\frac{8\widetilde{\rho_f} m_\epsilon^{\mathfrak{p}/2} d_Y}{\epsilon} + 1\right)^{d_Y m_\epsilon}. \tag{D.9}$$

This part of the proof is concluded by converting the covering $\mathscr{F}_{\mu_\lambda}^{m_\epsilon, N}$ into an *exterior cover* of the set $\partial B(\widetilde{r}) \subset \mathscr{F}$ — that is, its elements cover $\partial B(\widetilde{r})$ but they are required to belong to $\mathscr{F}$ and not necessarily to $\partial B(\widetilde{r})$. Indeed, with the construction carried out so far, for each $f \in \partial B(\widetilde{r})$ we can identify $f' \in \mathscr{F}_{\mu_\lambda}^{m_\epsilon}$ such that $\|f - f'\|_{\mathscr{L}^\infty(\Omega^T; \mathbb{R}^{d_Y})} \leq \epsilon/4$, and a $f'' \in \mathscr{F}_{\mu_\lambda}^{m_\epsilon, N}$ such that $\|f' - f''\|_{\mathscr{L}^\infty(\Omega^T; \mathbb{R}^{d_Y})} \leq \epsilon/4$: thus, by the triangle inequality, $\|f - f''\|_{\mathscr{L}^\infty(\Omega^T; \mathbb{R}^{d_Y})} \leq \epsilon/2$, implying that $\mathscr{F}_{\mu_\lambda}^{m_\epsilon, N}$ is an exterior cover of $\partial B(\widetilde{r})$ of resolution $\epsilon/2$. The final claim follows by applying (Vershynin, 2024, Exercise 4.2.9).

**Hypercontractivity.** We now prove $(C(\epsilon), 2)$-hypercontractivity of the tuple $(\mathscr{F}_\epsilon, \mu_\lambda^T)$ — the same claim will hold also for $(\mathscr{F}_\epsilon, \mathbb{P}_X)$ by multiplying $C_\epsilon$ by a constant not depending on $\epsilon$ and is thus not relevant to our analysis.

We start by showing that $\mathscr{F}_{\mu_\lambda}^{m_\epsilon}$ satisfies the hypercontractivity condition (Definition D.1) with $\alpha = 2$. Letting $f = \sum_{\ell=1}^{m_\epsilon} z_\ell \phi_\ell \in \mathscr{F}_{\mu_\lambda}^{m_\epsilon}$, we first observe that

$$\left\| \sum_{\ell=1}^{m_\epsilon} z_\ell \phi_\ell \right\|_{\mathscr{L}^2(\Omega^T, \mu_\lambda^T; \mathbb{R}^{d_Y})}^2 = \left\| \sum_{\ell=1}^{m_\epsilon} z_\ell \phi_\ell \right\|_{\mathscr{L}^2(\Omega, \mu_\lambda; \mathbb{R}^{d_Y})}^2 = \sum_{\ell=1}^{m_\epsilon} \|z_\ell\|_2^2. \tag{D.10}$$

Next, looking at the fourth moment,

$$\left\| \sum_{\ell=1}^{m_\epsilon} z_\ell \phi_\ell \right\|_{\mathscr{L}^4(\Omega^T, \mu_\lambda^T; \mathbb{R}^{d_Y})}^4 = \left\| \sum_{\ell=1}^{m_\epsilon} z_\ell \phi_\ell \right\|_{\mathscr{L}^4(\Omega, \mu_\lambda; \mathbb{R}^{d_Y})}^4$$

$$= \int_\Omega \left\| \sum_{\ell=1}^{m_\epsilon} z_\ell \phi_\ell(\mathrm{x}) \right\|_2^4 d\mathrm{x}$$

$$\leq \int_\Omega \left( \sum_{\ell=1}^{m_\epsilon} \|z_\ell\|_2 \, |\phi_\ell(x)| \right)^4 dx$$

$$\leq \mu_\lambda(\Omega) \left( \sum_{\ell=1}^{m_\epsilon} \|z_\ell\|_2 \right)^4$$

$$\leq \mu_\lambda(\Omega) \left( \sqrt{\sum_{\ell=1}^{m_\epsilon} \|z_\ell\|_2^2} \right)^4 (\sqrt{m_\epsilon})^4$$

$$\overset{\text{(D.10)}}{=} \mu_\lambda(\Omega) m_\epsilon^2 \left( \left\| \sum_{\ell=1}^{m_\epsilon} z_\ell \phi_\ell \right\|_{\mathscr{L}^2(\Omega^T, \mu_\lambda^T; \mathbb{R}^{d_Y})}^2 \right)^2, \tag{D.11}$$

which shows the $(\mu_\lambda(\Omega) m_\epsilon^2, 2)$-hypercontractivity of $\mathscr{F}_{\mu_\lambda}^{m_\epsilon}$.

Before showing hypercontractivity of $(\mathscr{F}_\epsilon, \mu_\lambda^T)$, we first state some additional useful relations. Recalling that $\epsilon \leq \inf_{f \in \partial B(\tilde{r})} \|f\|_{\mathscr{L}^2(\Omega^T, \mu_\lambda^T; \mathbb{R}^{d_Y})}$, letting $f$ be an arbitrary element in the cover $\mathscr{F}_\epsilon$ and $f'$ be a function in $\mathscr{F}_{\mu_\lambda}^{m_\epsilon}$ such that $\|f - f'\|_{\mathscr{L}^\infty(\Omega, \mathbb{R}^{d_Y})} \leq \epsilon/4$, we have:

$$\|f(x)\|_2^4 \leq 8 \left( \|f(x) - f'(x)\|_2^4 + \|f'(x)\|_2^4 \right) \leq \frac{\epsilon^4}{32} + 8 \|f'(x)\|_2^4 \tag{D.12a}$$

$$\|f'(x)\|_2^2 \leq 2 \left( \|f(x) - f'(x)\|_2^2 + \|f(x)\|_2^2 \right) \leq \frac{\epsilon^2}{8} + 2 \|f(x)\|_2^2 \tag{D.12b}$$

$$\epsilon^2 \leq \|f\|_{\mathscr{L}^2(\Omega^T, \mu_\lambda^T; \mathbb{R}^{d_Y})}^2, \qquad \epsilon^4 \leq \left( \|f\|_{\mathscr{L}^2(\Omega^T, \mu_\lambda^T; \mathbb{R}^{d_Y})}^2 \right)^2. \tag{D.12c}$$

We can now get to the final claim and show hypercontractivity for $\mathscr{F}_\epsilon$. Letting again $f$ be an arbitrary element in the cover $\mathscr{F}_\epsilon$, it holds that

$$\|f\|_{\mathscr{L}^4(\Omega^T, \mu_\lambda^T; \mathbb{R}^{d_Y})} = \int_\Omega \|f(x)\|_2^4 \, dx$$

$$\overset{\text{(D.12a)}}{\leq} \mu_\lambda(\Omega) \frac{\epsilon^4}{32} + 8 \int_\Omega \|f'(x)\|_2^4 \, dx$$

$$= \mu_\lambda(\Omega) \frac{\epsilon^4}{32} + 8 \int_\Omega \left\| \sum_{\ell=1}^{m_\epsilon} z_\ell' \phi_\ell(x) \right\|_2^4 dx$$

$$\overset{\text{(D.11)}}{\leq} \mu_\lambda(\Omega) \frac{\epsilon^4}{32} + 8\mu_\lambda(\Omega) m_\epsilon^2 \left( \int_\Omega \|f'(x)\|_2^2 \, dx \right)^2$$

$$\overset{\text{(D.12b)}}{\leq} \mu_\lambda(\Omega) \frac{\epsilon^4}{32} + 8\mu_\lambda(\Omega) m_\epsilon^2 \left( \int_\Omega \frac{\epsilon^2}{8} + 2 \|f(x)\|_2^2 \, dx \right)^2$$

$$\overset{\text{(D.12c)}}{\leq} \left( \frac{\mu_\lambda(\Omega)}{32} + 8\mu_\lambda(\Omega) m_\epsilon^2 \left( \frac{\mu_\lambda(\Omega)}{8} + 2 \right)^2 \right) \left( \|f\|_{\mathscr{L}^2(\Omega^T, \mu_\lambda; \mathbb{R}^{d_Y})} \right)^2,$$

which concludes the proof. $\qquad\qquad\qquad\qquad\qquad\qquad\qquad\qquad\qquad\qquad\qquad\qquad \square$

# E  ON THE REGULARIZER $\Psi(f)$

We now present the main properties of the regularizer $\Psi(f) \colon \mathscr{F} \to \mathbb{R}_{\geq 0}$ introduced in the learning problem (3.3), where we recall that $\Psi(f) = \|\mathscr{D}(f)\|_{\mathscr{L}^2(\Omega^T, \mathbb{P}_X; R^{d_Y})}^2$.

## E.1  PHYSICS-INFORMED REGULARIZATION IS 2-PROPER

We start from the definition of $\eta$-regularizer (Lecué & Mendelson, 2017):

*Definition* E.1. An $\eta$-proper regularizer defined for a hypothesis space $\mathcal{H}$ is a function $\Psi(\cdot) \colon \mathcal{H} \to \mathbb{R}$ satisfying the following properties:

(a) it is non-negative, even, convex, and such that $\Psi(0) = 0$;

(b) for $\eta \geq 1$, it holds for every $f, h$ in $\mathscr{F}$ that $\Psi(f + h) \leq \eta\left(\Psi(f) + \Psi(h)\right)$;

(c) for every $0 \leq a \leq 1$, $\Psi(af) \leq a\Psi(f)$. Additionally, if $\eta = 2$, it holds that $\Psi(af) \leq a^2 \Psi(f)$.

In particular, any square-norm-based regularizer is 2-proper. Therefore, by construction, the physics-informed regularizer considered in this paper (see (3.5)) is 2-proper.

## E.2 Useful inequality from Lecué & Mendelson (2017)

We now report an inequality that will be used in the proof of Theorem 4.1 proved in Section H.1.

**Lemma E.1** (Lecué & Mendelson (2017), Inequality 2.3). *Denote with $\hat{f}$ the solution of the regularized empirical risk minimization over $\mathscr{F}^\rho$. Write $\hat{f} = f_\star + R(h - f_\star)$, where $R \geq 1$ and $\Psi(h - f_\star) = \rho$, with $\rho \geq 5\eta\Psi(f_\star)$. Then it holds that*

$$\Psi(\hat{f}) - \Psi(f_\star) \geq \frac{R}{2\eta^2}(\Psi(h) - \Psi(f_\star)). \tag{E.1}$$

*Proof.* By the triangle inequality and the fact that the regularizer is an even function (both reported in Definition E.1), it holds that

$$\Psi(\hat{f}) = \Psi(f_\star + R(h - f_\star)) \geq \frac{1}{\eta}\Psi(R(h - f_\star)) - \Psi(f_\star) \geq \frac{R}{\eta}\Psi(h - f_\star) - \Psi(f_\star)$$

recalling that $R \geq 1$. Adding $\Psi(f_\star)$ on both sides,

$$\Psi(\hat{f}) - \Psi(f_\star) \geq \frac{R}{\eta}\Psi(h - f_\star) - 2\Psi(f_\star). \tag{E.2}$$

As an intermediate step, we find a lower bound on the term $\frac{R}{\eta}\Psi(h - f_\star)$ in (E.2): specifically, it holds that

$$
\begin{aligned}
\frac{R}{\eta}\Psi(h - f_\star) &= \frac{R}{2\eta}\Psi(h - f_\star) + \frac{R}{2\eta}\Psi(h - f_\star) \\
&\geq \frac{R}{2\eta}\Psi(h - f_\star) + \frac{5R}{2}\Psi(f_\star) \quad \text{because } \rho = \Psi(h - f_\star) \geq 5\eta\Psi(f_\star)) \\
&\geq \frac{R}{2\eta}\Psi(h - f_\star) + \frac{R}{2}\Psi(f_\star) + 2\Psi(f_\star) \quad \text{because } R \geq 1, \\
&\geq \frac{R}{2\eta}\left(\Psi(h - f_\star) + \Psi(f_\star)\right) + 2\Psi(f_\star) \quad \text{as } \eta \geq 1 \text{ and } R \geq 1, \\
&\geq \frac{R}{2\eta^2}\Psi(h) + 2\Psi(f_\star)
\end{aligned}
$$

again as a consequence of the triangle inequality in Definition E.1(b). Plugging such an inequality back in (E.2), we obtain

$$\Psi(\hat{f}) - \Psi(f_\star) \geq \frac{R}{2\eta^2}\Psi(h) \geq \frac{R}{2\eta^2}\Psi(h) - \frac{R}{2\eta^2}\Psi(f_\star),$$

which yields the claim. $\qquad\square$

## F Lower isometry bound

We start by presenting in Section F.1 an ancillary result combining $(C, \alpha)$-hypercontractivity (Definition D.1) and $S$-persistence (Section A.2). This will then play a key role in Section F.2, where we prove an upper bound for the probability of the lower isometry event, which will be crucial in our main results stated in Sections 4 and 5.

### F.1 COMBINING $(C, \alpha)$-HYPERCONTRACTIVITY AND $S$-PERSISTENCE

We now present an ancillary result obtained by generalizing (Ziemann, 2022, Lemma 3.1.1.).

**Lemma F.1.** *Consider $g \colon \Omega \to \mathbb{R}_{\geq 0}$ satisfying $(C, \alpha)$-hypercontractivity (see Definition D.1) and $S$-persistence (see Assumption 6). Then, for $\theta \geq 8$, it holds that*

$$\mathbb{P}_X \left( \sum_{t=0}^{T-1} g(X_t) \leq \frac{4}{\theta} \sum_{t=0}^{T-1} \mathbb{E}_{\mathbb{P}_X} [g(X_t)] \right) \leq \exp \left( -\frac{8T}{CS\theta^2} \left( \sum_{t=0}^{T-1} \mathbb{E}_{\mathbb{P}_X} [g(X_t)] \right)^{2-\alpha} \right) \quad \text{(F.1)}$$

*Proof.* We start by generalizing the $S$-persistence bound in Assumption 6. Specifically, introducing $\varepsilon > 0$, it holds that

$$\mathbb{E} \left[ \exp \left( -\xi \sum_{t=0}^{T-1} g(X_t) \right) \right] \leq \exp \left( -\frac{8\xi}{\theta} \sum_{t=0}^{T-1} \mathbb{E}[g(X_t)] + \frac{\xi^2 S \varepsilon}{\theta} \sum_{t=0}^{T-1} \mathbb{E}[g^2(X_t)] \right), \quad \text{(F.2)}$$

where it is required that $\theta \geq 8$ and $\varepsilon/\theta \geq 1/2$.
Now we consider the left-hand side of (F.1) and apply a Chernoff bound to obtain

$$\mathbb{P} \left( \sum_{t=0}^{T-1} g(X_t) \leq \frac{4}{\theta} \sum_{t=0}^{T-1} \mathbb{E}[g(X_t)] \right) \leq \inf_{\xi > 0} \mathbb{E} \left[ \exp \left( \frac{4\xi}{\theta} \sum_{t=0}^{T-1} \mathbb{E}[g(X_t)] - \xi \sum_{t=0}^{T-1} g(X_t) \right) \right]$$

$$\overset{\text{(F.2)}}{\leq} \inf_{\xi > 0} \mathbb{E} \left[ \exp \left( -\frac{4\xi}{\theta} \sum_{t=0}^{T-1} \mathbb{E}[g(X_t)] + \frac{\xi^2 S \varepsilon}{\theta} \sum_{t=0}^{T-1} \mathbb{E}[g^2(X_t)] \right) \right].$$

We find the optimal $\xi$, which reads as

$$\xi = \frac{2 \sum_{t=0}^{T-1} \mathbb{E}[g(X_t)]}{S \varepsilon \sum_{t=0}^{T-1} \mathbb{E}[g^2(X_t)]},$$

and plugging it in we obtain

$$\mathbb{P} \left( \sum_{t=0}^{T-1} g(X_t) \leq \frac{4}{\theta} \sum_{t=0}^{T-1} \mathbb{E}[g(X_t)] \right) \leq \exp \left( -\frac{4}{\theta S \varepsilon} \frac{\left( \sum_{t=0}^{T-1} \mathbb{E}[g(X_t)] \right)^2}{\sum_{t=0}^{T-1} \mathbb{E}[g^2(X_t)]} \right)$$

$$\leq \exp \left( -\frac{4T}{\theta C S \varepsilon} \left( \sum_{t=0}^{T-1} \mathbb{E}[g(X_t)] \right)^{2-\alpha} \right)$$

by $(C, \alpha)$-hypercontractivity given in Definition D.1. Conclusion follows by minimizing the bound over $\varepsilon \geq \theta/2$. $\qquad \square$

### F.2 THE MAIN BOUND ON LOWER ISOMETRY

We are now ready to prove the key bound for the lower isometry event by generalizing (Ziemann, 2022, Theorem 3.1.2.).

**Theorem F.2.** *Assume that the tuple $(\mathscr{F}^\rho, \mathbb{P}_X)$ is $S$-persistent (Assumption 6). For a given $r > 0$, define $B(r) \doteq \left\{ f \in \mathscr{F} \mid \|f\|_{\mathscr{L}^2(\Omega^T, \mathbb{P}_X; \mathbb{R}^{d_Y})} \leq r^2 \right\}$ and let $\partial B(r)$ be its boundary. Additionally, assume that the hypothesis space satisfies the $(C, \alpha)$-hypercontractivity condition (Definition D.1) on $\partial B(r)$. For a fixed $\theta > 8$, define $\mathscr{F}_r$ the $r/\sqrt{\theta}$-cover in the $\|\cdot\|_{\mathscr{L}^\infty(\Omega^T; \mathbb{R}^{d_Y})}$ of $\partial B(r)$, and denote by $\mathcal{N}_\infty \left( \partial B(r), \frac{r}{\sqrt{\theta}} \right)$ the corresponding covering number. Define the lower-isometry event*

$$\mathcal{A}_r \doteq \sup_{f \in \mathscr{F}_\star^\rho \setminus B(r)} \left\{ \frac{1}{T} \sum_{t=0}^{T-1} \|f(X_t)\|_2^2 - \frac{1}{\theta} \|f\|_{\mathscr{L}^2(\Omega^T, \mathbb{P}_X; \mathbb{R}^{d_Y})}^2 \leq 0 \right\}. \quad \text{(F.3)}$$

*Then the following lower-isometry estimate holds:*

$$\mathbb{P}_X(\mathcal{A}_r) \leq \mathcal{N}_\infty\left(\partial B(r), \frac{r}{\sqrt{\theta}}\right) \exp\left\{-\frac{8Tr^{4-2\alpha}}{\theta^2 CS}\right\}.$$

*Proof.* We first show a preliminary result that allows us to focus just on the boundary $\partial B(r)$ instead of the full $\mathscr{F}_\star \setminus B(r)$. Specifically, we show that, if $f \in \mathscr{F}_\star^\rho$ and $0 \leq \xi \leq 1$, then $\xi f$ still belongs to $\mathscr{F}_\star^\rho$: that is, we show that $\mathscr{F}_\star^\rho$ is *star-shaped* around 0 (see (Mendelson, 2014, Definition 5.1)). To prove that $\xi(f - f_\star)$ belongs to $\mathscr{F}_\star^\rho$ for $f, f_\star \in \mathscr{F}^\rho$ we deploy Theorem D.1: specifically, we have

$$\xi f - \xi f_\star \pm f_\star = \underbrace{\xi f + (1-\xi)f_\star}_{w} - f_\star,$$

and $w \in \mathscr{F}^\rho$ by convexity, thus proving the claim.

Thanks to the result above obtained, we can focus on $\partial B(r)$ and then obtain the final claim by rescaling: if $f' \in \mathscr{F}_\star^\rho \setminus B(r)$, then $\|f'\|_{\mathscr{L}^2(\Omega^T, \mathbb{P}_X; \mathbb{R}^{d_Y})} > r$ by construction, which implies $\frac{r}{\|f'\|_{\mathscr{L}^2(\Omega^T, \mathbb{P}_X; \mathbb{R}^{d_Y})}} < 1$; thus, if we consider $f = f' \frac{r}{\|f'\|_{\mathscr{L}^2(\Omega^T, \mathbb{P}_X; \mathbb{R}^{d_Y})}}$, we are on $\partial B(r)$, and $f \in \mathscr{F}_\star^\rho$ by it being star-shaped around 0.

Define the event

$$\mathcal{E} \doteq \bigcup_{f^i \in \mathscr{F}_r} \left\{\frac{1}{T}\sum_{t=0}^{T-1} \|f^i(X_t)\|_2^2 \leq \frac{4}{\theta}\|f^i\|_{\mathscr{L}^2(\Omega^T, \mathbb{P}_X; \mathbb{R}^{d_Y})}^2\right\}.$$

By Theorem F.1 with $g(x) = \|f^i(x)\|_2^2$, the union bound yields

$$\mathbb{P}_X(\mathcal{E}) \leq \mathcal{N}_\infty\left(\partial B(r), \frac{r}{\sqrt{\theta}}\right) \exp\left\{-\frac{8Tr^{4-2\alpha}}{\theta^2 CS}\right\}.$$

Now, fixing an arbitrary $f \in \partial B(r)$:

$$\frac{1}{T}\sum_{t=0}^{T-1} \|f(X_t)\|_2^2 \geq \frac{1}{2T}\sum_{t=0}^{T-1} \|f^i(X_t)\|_2^2 - \frac{r^2}{\theta} \qquad \text{Equation (D.1),}$$

$$\geq \frac{2}{\theta}\|f^i\|_{\mathscr{L}^2(\Omega^T, \mathbb{P}_X; \mathbb{R}^{d_Y})}^2 - \frac{r^2}{\theta} \qquad \text{on } \mathcal{E}^\complement,$$

$$= \frac{2r^2}{\theta} - \frac{r^2}{\theta} = \frac{r^2}{\theta} \qquad \text{by definition of } \mathscr{F}_r.$$

Since $f$ was arbitrary, we obtain

$$\mathbb{P}\left(\sup_{f \in \partial B(r)}\left\{\frac{1}{T}\sum_{t=0}^{T-1} \|f(X_t)\|_2^2 - \frac{r^2}{\theta} \leq 0\right\}\right) \leq \mathcal{N}_\infty\left(\partial B(r), \frac{r}{\sqrt{\theta}}\right)\exp\left\{-\frac{8Tr^{4-2\alpha}}{\theta^2 CS}\right\}.$$

The claim is finally obtained by rescaling. $\qquad \square$

We can now provide a special case of Theorem F.2 that will be useful in the derivations of the paper.

**Corollary F.3.** *Under the assumptions of Theorem F.2, assume that the hypothesis space satisfies the $(C(r), 2)$−hypercontractivity condition according to Theorem D.3. Then, the following lower-isometry estimate holds:*

$$\mathbb{P}_X(\mathcal{A}_r) \leq \left(C_L\left(\frac{1}{r}\right)^{\frac{4s-d_X}{2s-d_X}} + 1\right)^{d_Y C_m\left(\frac{1}{r}\right)^{\frac{2d_X}{2s-d_X}}} \exp\left\{-\frac{8Tr^{\frac{4d_X}{2s-d_X}}}{\theta^2 C_h S}\right\},$$

*where $C_L, C_m$ and $C_h$ are constants that depend on $\rho_f, \overline{\kappa}, d_Y, \theta, s, d_X$ and $\Omega$.*

*Proof.* The corollary is obtained by leveraging Theorem D.3, which provides expressions for both the covering number of $\mathscr{F}_r$ and for the hypercontractivity parameter $C(r)$, setting the covering of $\partial B(r)$ to have resolution equal to $r/\sqrt{\theta}$.

**Covering number.** Using the notation of Theorem D.3, we have that

$$\mathcal{N}_\infty\left(\partial B(r), \frac{r}{\sqrt{\theta}}\right) \overset{\text{(D.3)}}{\leq} \left(\frac{8\widetilde{\rho_f}d_Y\sqrt{\theta}m_{\frac{r}{\sqrt{\theta}}}^{\frac{s}{d_X}}}{r} + 1\right)^{d_Y m_{\frac{r}{\sqrt{\theta}}}},$$

$$\text{where } m_{\frac{r}{\sqrt{\theta}}} \geq \left(\frac{16\widetilde{\rho_f}d_X\theta}{2s-d_X}\cdot\left(\frac{1}{r^2}\right)\right)^{\frac{d_X}{2s-d_X}} \longrightarrow m_{\frac{r}{\sqrt{\theta}}} = C_m\left(\frac{1}{r}\right)^{\frac{2d_X}{2s-dx}}. \tag{F.4}$$

With such a value for $m_{\frac{r}{\sqrt{\theta}}}$, we obtain that the covering number admits the following upper bound:

$$\mathcal{N}_\infty\left(\partial B(r), \frac{r}{\sqrt{\theta}}\right) \leq \left(\underbrace{8\widetilde{\rho_f}d_Y\sqrt{\theta}C_m^{\frac{s}{d_X}}}_{\doteq C_L}\left(\frac{1}{r}\right)^{\frac{4s-d_X}{2s-d_X}} + 1\right)^{d_Y C_m\left(\frac{1}{r}\right)^{\frac{2d_X}{2s-d_X}}}. \tag{F.5}$$

**Hypercontractivity parameter.** Again, using the result in Theorem D.3 and using the expression for $m_{\frac{r}{\sqrt{\theta}}}$ in (F.4), we obtain that

$$C(r) \overset{\text{(D.4)}}{\propto} \left(\frac{\mu_\lambda(\Omega)}{32} + 8\mu_\lambda(\Omega)\left(\frac{\mu_\lambda(\Omega)}{8} + 2\right)^2 C_m^2\left(\frac{1}{r}\right)^{\frac{4d_X}{2s-d_X}}\right)$$

$$\longrightarrow C(r) = C_h\left(\frac{1}{r}\right)^{\frac{4d_X}{2s-d_X}} \text{ for some sufficiently large constant } C_h. \tag{F.6}$$

The lower-isometry probability bound is then obtained by plugging (F.5) and (F.6) in the claim of Theorem F.2. $\qquad\square$

## G MARTINGALE OFFSET COMPLEXITY BOUNDS

In this section we focus on some useful results concerning the martingale offset complexity presented in (4.2) and that will play a prominent role in the main results of Sections 4 and 5, being an upper-bound on the empirical excess risk. We start by proving the inequality leading to the definition of the martingale offset complexity, building upon Liang et al. (2015). Next, we report its bounds in probability and in expectation obtained by the chaining arguments of (Ziemann, 2022, Theorem 4.2.2, Theorem 3.2.1). The proofs of the latter, given in Section G.2 and Section G.3 respectively, are given in full detail to keep track of all of the constants involved.

### G.1 BEHIND THE SCENES OF THE DEFINITION

The first result is an ancillary inequality derived by extending Liang et al. (2015) to the regularized case (see also (Ziemann et al., 2022, Lemma 1)). The following lemma is the basis yielding the definition of martingale offset complexity.

**Lemma G.1.** *Let $\hat{f}$ be the solution of the regularized empirical risk minimization problem (3.3). Then, it holds that*

$$\frac{1}{T}\sum_{t=0}^{T-1}\left\|\hat{f}(X_t) - f_\star(X_t)\right\|_2^2 \leq \frac{1}{T}\sum_{t=0}^{T-1} 4\left\langle W_t, \hat{f}(X_t) - f_\star(X_t)\right\rangle_2 - \left\|\hat{f}(X_t) - f_\star(X_t)\right\|_2^2.$$

*Proof.* We start by showing the following facts:

- *Fact 1:* for any $f$ and the measurement model $Y_t = f_\star(X_t) + W_t$ in (3.1), it holds that

$$\|f(X_t) - f_\star(X_t)\|_2^2 = \|Y_t - f(X_t)\|_2^2 - \|Y_t - f_\star(X_t)\|_2^2 + 2\langle W_t, f(X_t) - f_\star(X_t)\rangle_2;$$

- *Fact 2:* from the construction of (3.3), we have that

$$\frac{1}{T} \sum_{t=0}^{T-1} \left\| Y_t - \hat{f}(X_t) \right\|_2^2 + \lambda_T \Psi(\hat{f}) \le \frac{1}{T} \sum_{t=0}^{T-1} \|Y_t - f_\star(X_t)\|_2^2 + \lambda_T \Psi(f_\star). \qquad \text{(G.1)}$$

Additionally, since $\Psi(f_\star) \le \Psi(\hat{f})$, we have

$$\frac{1}{T} \sum_{t=0}^{T-1} \left\| Y_t - \hat{f}(X_t) \right\|_2^2 - \|Y_t - f_\star(X_t)\|_2^2 \le 0. \qquad \text{(G.2)}$$

*Fact 2* follows immediately from optimality of $\hat{f}$. To see why *Fact 1* holds:

$$\|f(X_t) - f_\star(X_t) \pm Y_t\|_2^2 = \|Y_t - f(X_t)\|_2^2 + \|Y_t - f_\star(X_t)\|_2^2 - 2 \langle Y_t - f(X_t), Y_t - f_\star(X_t) \rangle_2 .$$

Considering the last addendum on the right-hand side, adding and subtracting $f_\star(X_t)$ in $(Y_t - f(X_t))$ and using the definition of $W_t$ for the other term, we obtain

$$\|f(X_t) - f_\star(X_t)\|_2^2 = \|Y_t - f(X_t)\|_2^2 + \|Y_t - f_\star(X_t)\|_2^2$$
$$- 2 \|Y_t - f_\star(X_t)\|_2^2 - 2 \langle W_t, f_\star(X_t) - f(X_t) \rangle_2 ,$$

thus proving the claim in *Fact 1*.

We are now ready to prove Theorem G.1. We start by applying *Fact 1* to the estimate $\hat{f}$ of (3.3) and multiplying everything by 2. Rearranging the terms, we then obtain

$$\frac{1}{T} \sum_{t=0}^{T-1} \left\| \hat{f}(X_t) - f_\star(X_t) \right\|_2^2 = \frac{1}{T} \sum_{t=0}^{T-1} 2 \left[ \overbrace{\left\| Y_t - \hat{f}(X_t) \right\|_2^2 - \|Y_t - f_\star(X_t)\|_2^2}^{(\natural)} \right]$$
$$+ 4 \left\langle W_t, \hat{f}(X_t) - f_\star(X_t) \right\rangle_2 - \left\| \hat{f}(X_t) - f_\star(X_t) \right\|_2^2 .$$

The conclusion follows by applying *Fact 2* to $(\natural)$. In particular, the claim is obtained by deploying (G.2). If on the other hand one would use (G.1), we would obtain

$$\frac{1}{T} \sum_{t=0}^{T-1} \left\| \hat{f}(X_t) - f_\star(X_t) \right\|_2^2$$
$$\le \frac{1}{T} \sum_{t=0}^{T-1} 4 \left\langle W_t, \hat{f}(X_t) - f_\star(X_t) \right\rangle_2 - \left\| \hat{f}(X_t) - f_\star(X_t) \right\|_2^2 + 2\lambda_T \left( \Psi(f_\star) - \Psi(\hat{f}) \right) \qquad \text{(G.3)}$$
$$\le \frac{1}{T} \sum_{t=0}^{T-1} 4 \left\langle W_t, \hat{f}(X_t) - f_\star(X_t) \right\rangle_2 - \left\| \hat{f}(X_t) - f_\star(X_t) \right\|_2^2 + 2\lambda_T \Psi(f_\star). \qquad \text{(G.4)}$$

$\square$

## G.2 BOUND IN PROBABILITY

We now provide the high-probability bound for the martingale offset complexity of a given hypothesis space $\mathcal{H}$. This will then be deployed in the high-probability bound for the excess risk in Theorem 4.1, and its analysis will be key to determine the desired rate.

**Theorem G.2** (Ziemann (2022), Theorem 4.2.2)**.** *Let Assumptions 1 to 3 and 6 hold, and let $\mathcal{H} \subset \mathcal{H}^s(\Omega^T; \mathbb{P}_X; \mathbb{R}^{d_Y})$ be a convex hypothesis space satisfying Assumption 5. Let $u, v, w \ge 0$. Then,*

*with probability* $1 - 4\exp\{-u^2/2\} - \exp\{-v/2\}$ *the martingale offset complexity satisfies*

$$\mathbf{M}_T[\mathcal{H}] \leq \inf_{\gamma > 0} \left\{ 8\gamma(u+1)\sqrt{\frac{\sigma_W^2}{T}} + 8\int_0^\gamma \sqrt{\frac{\sigma_W^2 \log \mathcal{N}_\infty(\mathcal{H}, \varepsilon)}{T}} d\varepsilon \right.$$

$$\left. + 32\sigma_W^2 \frac{(v + \log \mathcal{N}_\infty(\mathcal{H}, \gamma))}{T} + 4\gamma^2 \right\}.$$

*Proof.* The core of the proof consists in a *chaining* argument (Talagrand, 2005), i.e., in finding a suitable finite cover of $\mathcal{H}$ and deploying it to derive the desired bounds. We start by defining the terms of the chaining.

**Chaining set-up.** Let $F_k$ denote the cover of $\mathcal{H}$ with radius $\epsilon_k = \frac{1}{2^k}$; given two arbitrary positive scalars $\delta < \gamma$, the values of $k$ belong to an interval of integers $[\underline{K}, \overline{K}]$ such that

$$\frac{1}{2^{\overline{K}+1}} \leq \delta \leq \frac{1}{2^{\overline{K}}} \leq \frac{1}{2^{\underline{K}+1}} \leq \gamma. \tag{G.5}$$

For an arbitrary $f \in \mathcal{H}$, denote with $\pi_k(f)$ the center of the ball in the cover $F_k$ that contains $f$ – i.e., the function such that $\|f - \pi_k(f)\|_{\mathscr{L}^\infty(\Omega^T; \mathbb{R}^{d_Y})} \leq \epsilon_k$.

Let us write the martingale offset complexity $\mathbf{M}_T[\mathcal{H}]$ using the following notation:

$$\mathbf{M}_T[\mathcal{H}] = \sup_{f \in \mathcal{H}} \frac{1}{T} \underbrace{\left[ \underbrace{\sum_{t=0}^{T-1} 4\langle W_t, f(X_t)\rangle_2}_{\doteq M_T(f)} - \underbrace{\sum_{t=0}^{T-1} \|f(X_t)\|_2^2}_{\doteq S_T(f)} \right]}_{\doteq N_T(f)}. \tag{G.6}$$

Now, let us exploit the sequence of coverings to write $M_T(f)$ as a telescopic sum:

$$M_T(f) = M_T(f) \pm M_T(\pi_{\overline{K}}(f)) \pm \cdots \pm M_T(\pi_{\underline{K}}(f))$$
$$= [M_T(f) - M_T(\pi_{\overline{K}}(f))] + D_T(f) + M_T(\pi_{\underline{K}}(f)),$$

where we set $D_T(f) \doteq \sum_{k=\underline{K}+1}^{\overline{K}} M_T(\pi_k(f)) - M_T(\pi_{k-1}(f))$. We can now focus on using such a rewriting of $M_T(f)$ in the expression of $N_T(f)$ of Equation (G.6). With adding and subtracting $S_T(\pi_{\underline{K}}(f))/2$, it reads as

$$N_T(f) = M_T(f) - S_T(f)$$

$$= [M_T(f) - M_T(\pi_{\overline{K}}(f))] + D_T(f) + \left[ M_T(\pi_{\underline{K}}(f)) - \frac{S_T(\pi_{\underline{K}}(f))}{2} \right]$$

$$+ \left[ \frac{S_T(\pi_{\underline{K}}(f))}{2} - S_T(f) \right]$$

$$\leq [M_T(f) - M_T(\pi_{\overline{K}}(f))] + D_T(f) + \left[ M_T(\pi_{\underline{K}}(f)) - \frac{S_T(\pi_{\underline{K}}(f))}{2} \right] + T\epsilon_{\underline{K}}^2,$$

where the inequality acting on the last term is obtained through (D.1). Thus, overall, to find a bound for $\mathbf{M}_T[\mathcal{H}] = \sup_{f \in \mathcal{H}} N_T(f)/T$, we are interested in

$$\sup_{f \in \mathcal{H}} N_T(f) \leq \underbrace{\sup_{\substack{f,g \in \mathcal{H} \\ \|f-g\|_{\mathscr{L}^\infty(\Omega^T;\mathbb{R}^{d_Y})} \leq 2^{-\overline{K}}}} [M_T(f) - M_T(g)]}_{\doteq(\mathbf{N.A})} + \underbrace{\sup_{f \in \mathcal{H}} D_T(f)}_{\doteq(\mathbf{N.B})}$$

$$+ \underbrace{\sup_{f \in F_{\underline{K}}} \left[ M_T(f) - \frac{S_T(f)}{2} \right]}_{\doteq(\mathbf{N.C})} + T \underbrace{\left(\frac{1}{2^{\underline{K}}}\right)^2}_{\stackrel{(G.5)}{\leq} 4\gamma^2}. \tag{G.7}$$

The proof proceeds with the following steps. For each (**N.x**) with **x** = **A, B, C**, we derive high-probability bounds of the form

$$
\mathbb{P}\left( \frac{1}{T}(\mathbf{N.x}) > \frac{1}{T}\boxed{\text{value}(\mathbf{x})} \right) \leq \boxed{\text{probability bound}(\mathbf{x})} ;
$$

then, by deploying the union bound, we combine those results and obtain an upper-bound for $\mathbf{M}_T[\mathcal{H}]$ that depends on the parameters $\delta$ and $\gamma$ introduced in the chaining set-up (G.5). Finally, by leveraging the condition on the Sobolev order in Assumption 2, we show that we can let $\delta \to 0$ and $w \to +\infty$ to obtain the final claim.

**Bound for (N.A).** Defining $\mathscr{F}_{\overline{K}} \doteq \{ f = f^{\mathfrak{a}} - f^{\mathfrak{b}}; \ f^{\mathfrak{a}}, f^{\mathfrak{b}} \in \mathcal{H} \mid \|f\|_{\mathscr{L}^{\infty}(\Omega^T; \mathbb{R}^{d_Y})} \leq 2^{-\overline{K}} \}$ and by linearity of $M_T(\cdot)$, we are interested in

$$
\mathbb{P}\left( \sup_{f \in \mathscr{F}_{\overline{K}}} M_T(f) > w \right) \leq e^{-\xi w} \mathbb{E}\left[ \exp\left\{ \xi \sup_{f \in \mathscr{F}_{\overline{K}}} \sum_{t=0}^{T-1} 4 \langle W_t, f(X_t) \rangle_2 \right\} \right], \tag{G.8}
$$

where the inequality follows by applying a Chernoff bound with $\xi > 0$. Now, by deploying monotonicity of the exponential function, we can work on finding upper-bounds for the term in curly brackets in (G.8). Specifically, we consider

$$
\xi \sup_{f \in \mathscr{F}_{\overline{K}}} \sum_{t=0}^{T-1} 4 \langle W_t, f(X_t) \rangle_2 \leq \xi \sup_{f \in \mathscr{F}_{\overline{K}}} 4 \sqrt{\sum_{t=0}^{T-1} \|W_t\|_2^2} \sqrt{\sum_{t=0}^{T-1} \|f(X_t)\|_2^2}
$$

$$
\leq \xi \frac{4\sqrt{T}}{2^{\overline{K}}} \sqrt{\sum_{t=0}^{T-1} \|W_t\|_2^2}
$$

$$
\leq \frac{1}{2} \left( \frac{4\sqrt{T}\xi}{2^{\overline{K}}} \right)^2 \sum_{t=0}^{T-1} \|W_t\|_2^2 + \frac{1}{2} \quad \text{by Young's inequality.} \tag{G.9}
$$

Now, by plugging (G.9) into (G.8), we obtain

$$
\mathbb{P}\left( \sup_{f \in \mathscr{F}_{\overline{K}}} M_T(f) > w \right) \leq e^{-\xi w + 1/2} \mathbb{E}\left[ \exp\left\{ \left( \frac{4\sqrt{T}\xi}{\sqrt{2} \cdot 2^{\overline{K}}} \right)^2 \sum_{t=0}^{T-1} \|W_t\|_2^2 \right\} \right]
$$

$$
\overset{(Lemma\ A.4)}{\leq} \exp\left\{ -\xi w + \frac{1}{2} + \xi^2 \left( \frac{4 T d_Y \sigma_W^2}{\sqrt{2} \cdot 2^{\overline{K}}} \right)^2 \right\}, \tag{G.10}
$$

provided that $\xi < \frac{\sqrt{2} \cdot 2^{\overline{K}}}{4 T d_Y \sigma_W^2}$ and using the law of total expectation on the sum of $\|W_t\|_2^2$. In view of obtaining a bound in terms of $w$ and not $w^2$, we can choose at our convenience $\xi$ such that the quadratic term in (G.10) becomes equal to 1/2. To this aim, setting $\xi = \frac{2^{\overline{K}}}{4 T d_Y \sigma_W^2}$ (note that it satisfies the constraint of Theorem A.4) and deploying the definition of $\delta$ in (G.5), we obtain

$$
\mathbb{P}\left( \sup_{f \in \mathscr{F}_{\overline{K}}} M_T(f) > w \right) \leq \exp\left\{ 1 - \frac{w}{4 \delta T d_Y \sigma_W^2} \right\}.
$$

By substituting $w \leftrightarrow w \cdot 4\delta T d_Y \sigma_W^2$ and dividing by $T$, we finally obtain

$$
\mathbb{P}\left( \sup_{f \in \mathscr{F}_{\overline{K}}} \frac{M_T(f)}{T} > 4 w \delta d_Y \sigma_W^2 \right) \leq \exp\{-w + 1\}. \tag{G.11}
$$

**Bound for (N.B).** We start by introducing the short-hand notation for the function space $\widetilde{F}_k \doteq \left\{ f = f^{\mathfrak{a}} - f^{\mathfrak{b}}; \ f^{\mathfrak{a}} \in F_k, f^{\mathfrak{b}} \in F_{k-1} \mid \|f\|_{\mathscr{L}^{\infty}(\Omega^T; \mathbb{R}^{d_Y})} \leq 2^{-k} \right\}$ for all $k = \underline{K} + 1, \dots, \overline{K}$. Additionally, by linearity of $M_T(\cdot)$, we also have that

$$\sup_{f \in \mathcal{H}} D_T(f) \leq \sum_{k=\underline{K}+1}^{\overline{K}} \sup_{f \in \mathcal{H}} M_T \left( \pi_k(f) - \pi_{k-1}(f) \right) = \sum_{k=\underline{K}+1}^{\overline{K}} \max_{f \in \widetilde{F}_k} M_T(f).$$

We proceed by first studying the single addendum $\max_{f \in \widetilde{F}_k} M_T(f)$, and then apply a union bound to reach the desired claim for (N.B).

Letting $u_k > 0$, by deploying a Chernoff bound, we get

$$\mathbb{P}\left( \max_{f \in \widetilde{F}_k} M_T(f) > u_k \right) \leq \min_{\xi} e^{-\xi u_k} \mathbb{E}\left[ \exp\left\{ \xi \max_{f \in \widetilde{F}_k} M_T(f) \right\} \right]$$

$$\leq \min_{\xi} e^{-\xi u_k} \mathbb{E}\left[ \sum_{f \in \widetilde{F}_k} \exp\left\{ \xi M_T(f) \right\} \right]. \tag{G.12}$$

We now upper-bound (G.12) by iteratively applying the law of total expectation: specifically, we have that

$$\mathbb{E}\left[ \sum_{f \in \widetilde{F}_k} \exp\left\{ \xi M_T(f) \right\} \right] = \sum_{f \in \widetilde{F}_k} \mathbb{E}\left[ \mathbb{E}\left[ \exp\left\{ \xi \sum_{t=0}^{T-1} 4 \langle W_t, f(X_t) \rangle_2 \right\} \Big| \mathcal{X}_{T-2} \right] \right]$$

$$= \sum_{f \in \widetilde{F}_k} \mathbb{E}\left[ \exp\left\{ \xi \sum_{t=0}^{T-2} 4 \langle W_t, f(X_t) \rangle_2 \right\} \right] \mathbb{E}\left[ \exp\left\{ \xi 4 \langle W_{T-1}, f(X_{T-1}) \rangle_2 \right\} \Big| \mathcal{X}_{T-2} \right]$$

$$\overset{(3.2)}{\leq} \sum_{f \in \widetilde{F}_k} \mathbb{E}\left[ \exp\left\{ \xi \sum_{t=0}^{T-2} 4 \langle W_t, f(X_t) \rangle_2 \right\} \right] \exp\left\{ \frac{8\xi^2 \sigma_W^2}{2^{2k}} \right\}$$

$$\leq \vdots \quad \text{(i.e., repeating the argument with the next filtrations } \mathcal{X}_{T-3}, \dots, \mathcal{X}_0)$$

$$\leq |\widetilde{F}_k| \exp\left\{ \frac{8T\xi^2 \sigma_W^2}{2^{2k}} \right\}, \tag{G.13}$$

where $|\widetilde{F}_k|$ is the cardinality of $\widetilde{F}_k = F_k \times F_{k-1}$. Now, after noting that $|\widetilde{F}_k| \leq \left( \mathcal{N}_{\infty} \left( \mathcal{H}, 2^{-k} \right) \right)^2$, we can plug the bound of (G.13) into (G.12) and obtain, by minimizing over $\xi$ (yielding $\xi = 2^{2k} u_k / (16 T \sigma_W^2)$),

$$\mathbb{P}\left( \max_{f \in \widetilde{F}_k} M_T(f) > u_k \right) \leq \left( \mathcal{N}_{\infty} \left( \mathcal{H}, \frac{1}{2^k} \right) \right)^2 \exp\left\{ -\frac{2^{2k} u_k^2}{32 T \sigma_W^2} \right\}$$

Additionally, by substituting $u_k \leftrightarrow u_k + 2^{-k} \sqrt{64 T \sigma_W^2 \log \mathcal{N}_{\infty} \left( \mathcal{H}, \frac{1}{2^k} \right)} > 0$, we can remove the dependence on the covering number from the probability bound. Applying the union bound over all $k = \underline{K} + 1, \cdots, \overline{K}$, we obtain that the bound for (N.B) can be written as

$$\mathbb{P}\left( \sup_{f \in \mathcal{H}} D_T(f) > \sum_{k=\underline{K}+1}^{\overline{K}} u_k + 2^{-k} \sqrt{64 T \sigma_W^2 \log \mathcal{N}_{\infty} \left( \mathcal{H}, \frac{1}{2^k} \right)} \right)$$

$$\leq \sum_{k=\underline{K}+1}^{\overline{K}} \exp\left\{ -\frac{2^{2k} u_k^2}{32 \sigma_W^2 T} \right\}. \tag{G.14}$$

We now want the right-hand side of (G.14) to depend on a single $u \in \mathbb{R}$, in order to obtain an upper-bound that reads, informally, as $\boxed{constant} \times \exp\left\{ -u^2/2 \right\}$. To do so, we operate on the terms $u_k$,

$k = \underline{K} + 1, \cdots, \overline{K}$ and set them to $u_k = 2^{2-k}\sqrt{\sigma_W^2 T}\sqrt{u^2 - \log 2^{-k+\underline{K}+1}}$: thanks to this choice, the right-hand side of (G.14) becomes

$$
\sum_{k=\underline{K}+1}^{\overline{K}} \exp\left\{ -\frac{2^{2k}}{32T\sigma_W^2} \cdot \left(2^{2-k}\sqrt{\sigma_W^2 T}\sqrt{u^2 - \log 2^{-k+\underline{K}+1}}\right)^2 \right\}
$$

$$
= \sum_{k=\underline{K}+1}^{\overline{K}} \exp\left\{ \frac{-u^2}{2} \right\} (\sqrt{2})^{-k+\underline{K}+1}
$$

$$
\leq \exp\left\{ \frac{-u^2}{2} \right\} \sum_{k=0}^{\infty} \left(\frac{1}{\sqrt{2}}\right)^k = (2+\sqrt{2})\exp\left\{ \frac{-u^2}{2} \right\} \tag{G.15}
$$

as desired. Now we can analyze such a choice for $u_k$ in the left-hand side of (G.14), which becomes

$$
\mathbb{P}\left( \sup_{f \in \mathcal{H}} D_T(f) > \overbrace{\sum_{k=\underline{K}+1}^{\overline{K}} 2^{2-k}\sqrt{\sigma_W^2 T(u^2 - \log 2^{-k+\underline{K}+1})}}^{\doteq s_1} \right.
$$

$$
\left. + \underbrace{\sum_{k=\underline{K}+1}^{\overline{K}} 2^{-k}\sqrt{64T\sigma_W^2 \log \mathcal{N}_\infty\left(\mathcal{H}, 2^{-k}\right)}}_{\doteq s_2} \right). \tag{G.16}
$$

We now want to find upper bounds for $s_1$ and $s_2$ and remove the sum over $k$. Regarding $s_1$, we have

$$
s_1 \leq \sum_{k=\underline{K}+1}^{\overline{K}} 2^{2-k}\sqrt{\sigma_W^2 T u^2} + \sum_{k=\underline{K}+1}^{\overline{K}} 2^{2-k}\sqrt{\sigma_W^2 T}\sqrt{\log\left(2^{k-\underline{K}-1}\right)}
$$

$$
\leq 4 \cdot 2^{-\underline{K}-1}\sqrt{\sigma_W^2 T u^2} \sum_{k=0}^{\infty} 2^{-k} + 4 \cdot 2^{-\underline{K}-1}\sqrt{\sigma_W^2 T \log 2} \underbrace{\sum_{k=0}^{\infty} 2^{-k}\sqrt{k}}_{=\mathrm{Li}_{-1/2}(1/2)}
$$

$$
\overset{\text{(G.5)}}{\leq} 8\gamma\sqrt{\sigma_W^2 T}(u+1), \tag{G.17}
$$

because the polylogarithmic function satisfies $\mathrm{Li}_{1/2}(1/2) \approx 1.35$. Now, going to $s_2$, noting that $2^{-k} = (2^{-k+1} - 2^{-k})$ and by deploying a truncated Dudley's entropy integral (Wainwright, 2019, Theorem 5.22), we have

$$
s_2 \leq \sum_{k=\underline{K}+1}^{\overline{K}} \left(\frac{1}{2^{k-1}} - \frac{1}{2^k}\right)\sqrt{64T\sigma_W^2 \log \mathcal{N}_\infty\left(\mathcal{H}, 2^{-k}\right)}
$$

$$
\leq \int_{2^{-\overline{K}}}^{2^{-\underline{K}-1}} \sqrt{64T\sigma_W^2 \log \mathcal{N}_\infty\left(\mathcal{H}, \varepsilon\right)}d\varepsilon
$$

$$
\overset{\text{(G.5)}}{\leq} \int_{\delta}^{\gamma} \sqrt{64T\sigma_W^2 \log \mathcal{N}_\infty\left(\mathcal{H}, \varepsilon\right)}d\varepsilon. \tag{G.18}
$$

Thus, plugging (G.17,G.18) in (G.16), dividing by $T$ and using the bound in (G.15), we obtain the desired bound for (**N.B**), which reads as

$$
\boxed{\mathbb{P}\left( \sup_{f \in \mathcal{H}} \frac{D_T(f)}{T} > \frac{8(u+1)\gamma\sqrt{\sigma_W^2}}{\sqrt{T}} + \frac{8}{\sqrt{T}}\int_{\delta}^{\gamma} \sqrt{\sigma_W^2 \log \mathcal{N}_\infty\left(\mathcal{H}, \varepsilon\right)}d\varepsilon \right) \leq 4\exp\left\{ -\frac{u^2}{2} \right\}.}
$$

$$
\tag{G.19}
$$

**Bound for (`N.C`).** Applying again a Chernoff bound along the lines of the manipulations for (`N.B`) in (G.12), we consider

$$
\mathbb{P}\left( \max_{f \in F_K} \sum_{t=0}^{T-1} 4 \langle W_t, f(X_t) \rangle_2 - \frac{1}{2} \|f(X_t)\|_2^2 > v \right)
$$

$$
\leq \min_{\xi} e^{-\xi v} \sum_{f \in F_K} \mathbb{E}\left[ \exp\left\{ \xi\left( \sum_{t=0}^{T-1} 4 \langle W_t, f(X_t) \rangle_2 - \frac{1}{2} \|f(X_t)\|_2^2 \right) \right\} \right]
$$

$$
\leq \min_{\xi} e^{-\xi v} \sum_{f \in F_K} \mathbb{E}\left[ \mathbb{E}\left[ \exp\left\{ \xi\left( \sum_{t=0}^{T-1} 4 \langle W_t, f(X_t) \rangle_2 - \frac{1}{2} \|f(X_t)\|_2^2 \right) \right\} \Big| \mathcal{X}_{T-2} \right] \right]
$$

$$
\leq \min_{\xi} e^{-\xi v} \left( \xi \sum_{f \in F_K} \mathbb{E}\left[ \exp\left( \left\{ \sum_{t=0}^{T-2} 4 \langle W_t, f(X_t) \rangle_2 - \frac{1}{2} \|f(X_t)\|_2^2 \right\} \right) \right] \right)
$$

$$
\times \mathbb{E}\left[ \exp\left\{ 4\xi \langle W_{T-1}, f(X_{T-1}) \rangle_2 - \xi\frac{1}{2} \|f(X_{T-1})\|_2^2 \right\} | \mathcal{X}_{T-2} \right]. \tag{G.20}
$$

We now focus on last expected value in (G.20) and discuss its upper bound. Specifically, we have

$$
\exp\left\{ -\xi\frac{1}{2} \|f(X_{T-1})\|_2^2 \right\} \mathbb{E}\left[ \exp\left\{ 4\xi \langle W_{T-1}, f(X_{T-1}) \rangle_2 \right\} \right]
$$

$$
\overset{(3.2)}{\leq} \exp\left\{ \|f(X_{T-1})\|_2^2 \left( -\frac{\xi}{2} + 8\xi^2 \sigma_W^2 \right) \right\} \leq 1
$$

by setting $\xi = (32\sigma_W^2)^{-1}$. By this choice of $\xi$, applying the law of total expectation iteratively over $t$ in (G.20) and using the definition of $\gamma$ in (G.5), we obtain

$$
\mathbb{P}\left( \max_{f \in F_K} \sum_{t=0}^{T-1} 4 \langle W_t, f(X_t) \rangle_2 - \frac{1}{2} \|f(X_t)\|_2^2 > v \right) \leq \mathcal{N}_\infty \left( \mathcal{H}, \gamma \right) \exp\left\{ -\frac{v}{32\sigma_W^2} \right\};
$$

finally, substituting $v \leftrightarrow 32\sigma_W^2(v/2 + \log \mathcal{N}_\infty \left( \mathcal{H}, \gamma \right))$ and dividing by $T$, we obtain the bound for (`N.C`):

$$
\boxed{\mathbb{P}\left( \sup_{f \in F_K} \frac{M_T(f)}{T} - \frac{1}{2T} S_T(f) > \frac{32\sigma_W^2}{T} \left( \frac{v}{2} + \log \mathcal{N}_\infty \left( \mathcal{H}, \gamma \right) \right) \right) \leq \exp\left\{ -\frac{v}{2} \right\}.} \tag{G.21}
$$

**Obtaining the final bound.** We can now combine these results and derive the high-probability bound for the martingale offset complexity $\mathbf{M}_T [\mathcal{H}] = \sup_{f \in \mathcal{H}} N_T(f)/T$. Leveraging the decomposition of $N_T(f)$ according to (G.7), we combine the bounds on (`N.A`), (`N.B`) and (`N.C`) in (G.11), (G.19) and (G.21) using the union bound and obtain that

$$
\mathbb{P}\Bigg( \mathbf{M}_T [\mathcal{H}] > 4w\delta d_Y \sigma_W^2 + 8\sqrt{\frac{\sigma_W^2}{T}} \left( (u+1)\gamma + \int_\delta^\gamma \sqrt{\log \mathcal{N}_\infty \left( \mathcal{H}, \varepsilon \right)} d\varepsilon \right)
$$

$$
+ \frac{32\sigma_W^2}{T} \left( \frac{v}{2} + \log \mathcal{N}_\infty \left( \mathcal{H}, \gamma \right) \right) + 4\gamma^2 \Bigg)
$$

$$
\leq \exp\left\{ -w + 1 \right\} + 4\exp\left\{ -\frac{u^2}{2} \right\} + \exp\left\{ -\frac{v}{2} \right\},
$$

and the expression for the lower bound of the martingale offset complexity is to be maximized with respect to $\gamma$ and $\delta < \gamma$.

We now claim that we can set $\delta = 0$ and simplify the bound. Setting $\delta = 0$ is possible only if the integral in the term (`N.B`) converges. Under our assumptions, we have that (see Section C.2)

$$
\int_\delta^\gamma \sqrt{\log \mathcal{N}_\infty \left( \mathcal{H}, \varepsilon \right)} d\varepsilon \propto \int_\delta^\gamma \left( \frac{1}{\varepsilon} \right)^{\frac{d_X}{2s}} d\varepsilon = \frac{\varepsilon^{1 - d_X/2s}}{1 - d_X/2s}, \tag{G.22}
$$

and the value is finite for $\delta \to 0$ if and only if $d_X/2s < 1$, which is guaranteed by Assumption 2. Therefore, we can set $\delta = 0$, and since the term associated to $(\mathtt{N.A})$ becomes 0, we can also let $w \to \infty$ and increase the final probability level in the claim of the theorem.

$\square$

## G.3 BOUND IN EXPECTATION

Along the lines of the result in probability of the previous subsection, we now present the bound in expectation for the martingale offset complexity.

**Theorem G.3** (Ziemann (2022), Theorem 3.2.1). *Let Assumptions 1 to 3 and 6 hold, and let $\mathcal{H} \subset \mathscr{H}^s(\Omega^T; \mathbb{P}_X; \mathbb{R}^{d_Y})$ be a convex hypothesis space satisfying Assumption 5. Then, the martingale offset complexity satisfies*

$$
\mathbb{E}\left[\mathbf{M}_T\left[\mathcal{H}\right]\right] \leq \inf_{\gamma > 0} \left\{ 8 \int_0^\gamma \sqrt{\frac{\sigma_W^2 \log \mathcal{N}_\infty\left(\mathcal{H}, \varepsilon\right)}{T}} d\varepsilon + \frac{32\sigma_W^2 \log \mathcal{N}_\infty\left(\mathcal{H}, \gamma\right)}{T} + 4\gamma^2 \right\}.
$$

*Proof.* The result is again obtained by chaining, using the construction leading to (G.7). Using the definitions of $N_T(f)$, $M_T(f)$ and $S_T(f)$ in (G.6), as well as the ones for the chaining resolutions $\delta, \gamma$ in (G.5), we are looking at

$$
\mathbb{E}\left[\sup_{f \in \mathcal{H}} \frac{1}{T} N_T(f)\right] \leq \frac{1}{T}\mathbb{E}\left[\underbrace{\sup_{\substack{f,g \in \mathcal{H} \\ \|f-g\|_{\mathscr{L}^\infty(\Omega^T;\mathbb{R}^{d_Y})} \leq 2^{-\overline{K}}}} M_T(f) - M_T(g)}_{(\mathtt{N.A})}\right] + \frac{1}{T}\mathbb{E}\left[\underbrace{\sup_{f \in \mathcal{H}} D_T(f)}_{(\mathtt{N.B})}\right]
$$

$$
+ \frac{1}{T}\mathbb{E}\left[\underbrace{\sup_{f \in F_{\underline{K}}} M_T(f) - \frac{S_T(f)}{2}}_{(\mathtt{N.C})}\right] + \underbrace{\left(\frac{1}{2^{\underline{K}}}\right)^2}_{\substack{(G.5) \\ \leq 4\gamma^2}}. \tag{G.23}
$$

We now proceed with deriving the bounds for the expected values of the terms $(\mathtt{N.A})$, $(\mathtt{N.B})$ and $(\mathtt{N.C})$; the final claim is obtained by summing all of the contributions together. Finally, we will discuss the fact that we are allowed to let $\delta = 0$ and simplify the bound.

**Bound for $(\mathtt{N.A})$.** Define $\mathscr{F}_{\overline{K}} \doteq \{f = f^{\mathfrak{a}} - f^{\mathfrak{b}};\ f^{\mathfrak{a}}, f^{\mathfrak{b}} \in \mathcal{H}\ |\ \|f\|_{\mathscr{L}^\infty(\Omega^T;\mathbb{R}^{d_Y})} \leq 2^{-\overline{K}}\}$. By linearity of $M_T(\cdot)$, we are looking at

$$
\mathbb{E}\left[\sup_{f \in \mathscr{F}_{\overline{K}}} \sum_{t=0}^{T-1} 4 \langle W_t, f(X_t) \rangle_2\right] \leq \mathbb{E}\left[\sup_{f \in \mathscr{F}_{\overline{K}}} \sum_{t=0}^{T-1} 4 \|W_t\|_2 \|f(X_t)\|_2\right]
$$

$$
\leq \frac{4}{2^{\overline{K}}} \sum_{t=0}^{T-1} \mathbb{E}\left[\|W_t\|_2\right] \overset{Lemma\ A.5}{\leq} \frac{12T}{2^{\overline{K}}} \sqrt{d_Y \sigma_W^2} \overset{(G.5)}{\leq} 24T\delta\sqrt{d_Y \sigma_W^2}.
$$

Dividing by $T$, we obtain the first term in the bound (G.23).

**Bound for $(\mathtt{N.B})$.** We start by defining the auxiliary search space $\widetilde{F}_k \doteq \left\{f = f^{\mathfrak{a}} - f^{\mathfrak{b}};\ f^{\mathfrak{a}} \in F_k, f^{\mathfrak{b}} \in F_{k-1}\ |\ \|f\|_{\mathscr{L}^\infty(\Omega^T;\mathbb{R}^{d_Y})} \leq 2^{-k}\right\}$ for all $k = \underline{K} + 1, \ldots, \overline{K}$. Next, using the definition of $D_T(f)$ and exploiting its linearity, we consider

$$
\mathbb{E}\left[\sup_{f \in \mathcal{H}} \sum_{k=\underline{K}+1}^{\overline{K}} M_T\left(\pi_k(f) - \pi_{k-1}(f)\right)\right] \leq \sum_{k=\underline{K}+1}^{\overline{K}} \mathbb{E}\left[\sup_{f \in \widetilde{F}_k} M_T(f)\right]. \tag{G.24}
$$

To upper-bound the right-hand side of (G.24), we focus on its addenda and proceed with the following argument. Noting that $\widetilde{F}_k$ is a finite-dimensional class, let us consider, for some $\xi > 0$,

$$
\exp\left\{\xi\mathbb{E}\left[\max_{f\in\widetilde{F}_k} M_T(f)\right]\right\} \leq \mathbb{E}\left[\exp\left\{\xi\max_{f\in\widetilde{F}_k} M_T(f)\right\}\right] \quad \text{by Jensen's inequality,}
$$

$$
= \mathbb{E}\left[\max_{f\in\widetilde{F}_k}\exp\left\{\xi M_T(f)\right\}\right] \quad \text{by monotonicity,}
$$

$$
\leq \sum_{f\in\widetilde{F}_k}\mathbb{E}\left[\exp\left\{\xi M_T(f)\right\}\right]
$$

$$
= \sum_{f\in\widetilde{F}_k}\mathbb{E}\left[\mathbb{E}\left[\exp\left\{\xi\sum_{t=0}^{T-1}4\left\langle W_t,\,f(X_t)\right\rangle_2\right\}\bigg|\mathcal{X}_{T-2}\right]\right]
$$

$$
\overset{(3.2)}{\leq}\mathbb{E}\left[\exp\left\{\xi\sum_{t=0}^{T-2}4\left\langle W_t,\,f(X_t)\right\rangle_2\right\}\right]\exp\left\{\frac{8\xi^2\sigma_W^2}{2^{2k}}\right\}
$$

$$
\leq\vdots \quad \text{(i.e., repeating the argument with the subsequent filtrations)}
$$

$$
\leq\left(\mathcal{N}_\infty\left(\mathcal{H},2^{-k}\right)\right)^2\exp\left\{\frac{8T\xi^2\sigma_W^2}{2^{2k}}\right\}, \tag{G.25}
$$

noting that the cardinality of $\widetilde{F}_k = F_k \times F_{k-1}$ is upper-bounded by $\left(\mathcal{N}_\infty\left(\mathcal{H},2^{-k}\right)\right)^2$. Now, taking logarithms of both sides of the whole inequality (G.25), we obtain

$$
\mathbb{E}\left[\max_{f\in\widetilde{F}_k} M_T(f)\right] \leq \frac{2}{\xi}\log\mathcal{N}_\infty\left(\mathcal{H},2^{-k}\right) + \frac{8T\xi\sigma_W^2}{2^{2k}}
$$

$$
\rightarrow\mathbb{E}\left[\max_{f\in\widetilde{F}_k} M_T(f)\right] \leq 2^{-k}\sqrt{64T\sigma_W^2\log\mathcal{N}_\infty\left(\mathcal{H},2^{-k}\right)} \tag{G.26}
$$

after minimizing with respect to $\xi$.

We can now go back to (G.24). Plugging (G.26), we obtain

$$
\mathbb{E}\left[\sup_{f\in\mathcal{H}}\sum_{k=\underline{K}+1}^{\overline{K}} M_T\left(\pi_k(f)-\pi_{k-1}(f)\right)\right] \leq \sum_{k=\underline{K}+1}^{\overline{K}}\frac{1}{2^k}\sqrt{64T\sigma_W^2\log\mathcal{N}_\infty\left(\mathcal{H},2^{-k}\right)}
$$

$$
= \sum_{k=\underline{K}+1}^{\overline{K}}\left(\frac{1}{2^{k-1}}-\frac{1}{2^k}\right)\sqrt{64T\sigma_W^2\log\mathcal{N}_\infty\left(\mathcal{H},2^{-k}\right)}
$$

$$
\leq \sum_{k=\underline{K}+1}^{\overline{K}}\int_{2^{-\overline{K}}}^{2^{-\underline{K}-1}}\sqrt{64T\sigma_W^2\log\mathcal{N}_\infty\left(\mathcal{H},\varepsilon\right)}d\varepsilon
$$

$$
\overset{(G.5)}{\leq} 8\int_\delta^\gamma\sqrt{T\sigma_W^2\log\mathcal{N}_\infty\left(\mathcal{H},\varepsilon\right)}d\varepsilon
$$

having used in the second inequality a truncated Dudley entropy integral (Wainwright, 2019, Theorem 5.22). Finally, the second term in (G.11) is obtained by divigind the last inequality by $T$.

**Bound for (N.C).** We are now working to find the upper bound for the expected value of the martingale offset complexity of a finite class of functions, $\mathbb{E}\left[\mathbf{M}_T\left[F_{\overline{K}}\right]\right]$, where $F_{\overline{K}}$ is the $2^{-\overline{K}}$-cover of the hypothesis space $\mathcal{H}$. Similarly to what has been done for (N.B), we start by noticing that, for any $\xi > 0$,

$$
\exp\left\{\xi\mathbb{E}\left[\max_{f\in F_{\overline{K}}} M_T(f)-\frac{1}{2}S_T(f)\right]\right\}
$$

$$\leq \mathbb{E}\left[\max_{f \in F_K} \exp\left\{\xi\left(M_T(f) - \frac{1}{2}S_T(f)\right)\right\}\right] \quad \text{(Jensen's inequality)}$$

$$\leq \sum_{f \in F_K} \mathbb{E}\left[\exp\left\{\xi\left(M_T(f) - \frac{1}{2}S_T(f)\right)\right\}\right]$$

$$= \sum_{f \in F_K} \mathbb{E}\left[\mathbb{E}\left[\exp\left\{\xi\left(\sum_{t=0}^{T-1} 4\xi \langle W_t, f(X_t)\rangle_2 - \frac{\xi}{2}\|f(X_t)\|_2^2\right)\right\}\Big| \mathcal{X}_{T-2}\right]\right] \quad \text{(total expectation)}$$

$$= \sum_{f \in F_K} \mathbb{E}\left[\exp\left\{\sum_{t=0}^{T-1} 4\langle W_t, f(X_t)\rangle_2 - \frac{1}{2}\|f(X_t)\|_2^2\right\}\right]$$

$$\cdot \mathbb{E}\left[4\langle W_{T-1}, f(X_{T-1})\rangle_2 - \frac{1}{2}\|f(X_{T-1})\|_2^2 \Big| \mathcal{X}_{T-2}\right]$$

$$\overset{(3.2)}{\leq} \sum_{f \in F_K} \mathbb{E}\left[\exp\left\{\sum_{t=0}^{T-1} 4\langle W_t, f(X_t)\rangle_2 - \frac{1}{2}\|f(X_t)\|_2^2\right\}\right] \underbrace{\exp\left\{\|f(X_{T-1})\|_2^2\left(-\frac{\xi}{2} + 8\xi^2\sigma_W^2\right)\right\}}_{\leq 1 \text{ by letting } \xi = (32\sigma_W^2)^{-1}}$$

$$\leq \quad \vdots \quad \text{(iterating over the subsequent filtrations)}$$

$$\leq \mathcal{N}_\infty\left(\mathcal{H}, \frac{1}{2^K}\right). \tag{G.27}$$

Now, by taking the logarithm on both sides of (G.27) and using the value $\xi = (32\sigma_W^2)^{-1}$ found above, we obtain

$$\mathbb{E}\left[\max_{f \in F_K} M_T(f) - \frac{1}{2}S_T(f)\right] \leq 32\sigma_W^2 \log \mathcal{N}_\infty\left(\mathcal{H}, \frac{1}{2^K}\right) \overset{(G.5)}{\leq} 32\sigma_W^2 \log \mathcal{N}_\infty\left(\mathcal{H}, \gamma\right),$$

and the bound for the third term in (G.23) is obtained by dividing the terms above by $T$.

**Wrapping up.** Putting the bounds for all the terms (`N.A`), (`N.B`) and (`N.C`) together, we obtain

$$\boxed{\mathbb{E}\left[\mathbf{M}_T\left[\mathcal{H}\right]\right] \leq 24\delta\sqrt{d_Y\sigma_W^2} + \int_\delta^\gamma \sqrt{\frac{64\sigma_W^2 \log \mathcal{N}_\infty\left(\mathcal{H}, \varepsilon\right)}{T}}d\varepsilon + \frac{32\sigma_W^2 \log \mathcal{N}_\infty\left(\mathcal{H}, \varepsilon\right)}{T} + 4\gamma^2.}$$

Following the reasoning at the end of the proof for Theorem G.2, we have that in the scenarios of our interest the integral does not diverge at $\delta = 0$. For this reason, in the final claim we will make use of $\delta = 0$ and simplify the bound. $\qquad\square$

## H PROOFS OF THE EXCESS RISK BOUNDS IN SECTION 4

This section provides the proof of Theorems 4.1 and 4.2. As discussed in Section 4.1, the results are derived leveraging the lower isometry event (F.3) and the bound on its probability presented in Section F. Moreover, we make use of the regularizer's properties elucidated in Section E, and of $(C(r), 2)$-hypercontractivity proved in Section D.2. Ultimately, we obtain that our *complexity-dependent* bounds on the excess risk feature three main ingredients: the complexity of the hypothesis class, captured by the martingale offset complexity; the critical radius $r$ identifying the set $B(r)$ and determining its size (thus, the covering number of its boundary, see Theorem F.2); and the ground-truth regularizer $\Psi(f_\star)$. These results bring together the small-ball method with learning with dependent data, and are the starting point for the derivation of our convergence rate results presented in Section 5 and proved in Section I.

### H.1 PROOF OF THEOREM 4.1 (RESULT IN PROBABILITY)

**Theorem 4.1.** Let Assumptions 1 to 3, 5 and 6 hold. Consider a parameter $\theta > 8$, and let $\hat{f}$ be the solution of the estimation problem (3.3) with $\lambda_T > 0$, and let the radius $\rho$ defining the effective

hypothesis class $\mathscr{F}^\rho$ be such that $\rho \geq 10\Psi(f_\star)$. Then, on the event

$$\mathcal{A}_r^\complement \cap \left\{ \lambda_T \geq \frac{40}{3\rho} \mathbf{M}_T \left[ \mathscr{F}^\rho \right] \right\}$$

we have that

$$\left\| \hat{f} - f_\star \right\|_{\mathscr{L}^2(\Omega^T, \mathbb{P}_X; \mathbb{R}^{d_Y})}^2 \leq \theta \mathbf{M}_T \left[ \mathscr{F}^\rho \right] + 2\lambda_T \Psi(f_\star) + r^2.$$

*Proof.* We start by noting that $\mathcal{A}_r^\complement$ can happen in the following situations:

(i) $\left\| \hat{f} - f_\star \right\|_{\mathscr{L}^2(\Omega^T, \mathbb{P}_X; \mathbb{R}^{d_Y})}^2 \leq r^2$;

(ii) $\hat{f} - f_\star$ is in $\mathscr{F}^\rho \setminus B(r)$, but it happens that $\|\hat{f} - f_\star\|_{\mathscr{L}^2}^2 \leq \frac{\theta}{T} \sum_{t=0}^{T-1} \left\| \hat{f}(X_t) - f_\star(X_t) \right\|_2^2$ (see (4.1));

(iii) $\hat{f}$ is outside $\mathscr{F}^\rho$.

The key idea of this Theorem is to prove that scenario (iii) cannot occur with our choice of the regularization parameter $\lambda_T$. We will now analyze each situation separately.

**Case (i).** This is the simple situation in which we are already in the $\mathscr{L}^2$-ball with radius $r$, $B(r)$, leading to $\|f - f_\star\|_{\mathscr{L}^2(\Omega^T, \mathbb{P}_X; \mathbb{R}^{d_Y})}^2 \leq r^2$.

**Case (ii).** On this event, we have

$$\left\| \hat{f} - f_\star \right\|_{\mathscr{L}^2(\Omega^T, \mathbb{P}_X; \mathbb{R}^{d_Y})}^2 \leq \frac{\theta}{T} \sum_{t=0}^{T-1} \left\| \hat{f}(X_T) - f_\star(X_t) \right\|_2^2$$

$$\overset{(G.4)}{\leq} \frac{\theta}{T} \sup_{g \in \mathscr{F}_\star^\rho} \sum_{t=0}^{T-1} \left[ 4 \langle W_t, g(X_t) \rangle_2 - \|g(X_t)\|_2^2 \right] + 2\lambda_T \Psi(f_\star)$$

$$\overset{(4.2)}{\leq} \theta \mathbf{M}_T \left[ \mathscr{F}^\rho \right] + 2\lambda_T \Psi(f_\star).$$

**Case (iii).** By Theorem D.1, the hypothesis space is convex, and the regularizer $\Psi(\cdot)$ is continuous: therefore, there exists $R > 1$ and $h \in \partial \mathscr{F}^\rho$ such that $\hat{f} = f_\star + R(h - f_\star)$. Additionally, by Definition E.1(b), we have that

$$\Psi(h) \geq \frac{1}{2} \Psi(h - f_\star) - \Psi(f_\star) \Rightarrow \Psi(h) - \Psi(f_\star) \geq \frac{3\rho}{10} \tag{H.1}$$

by virtue of our choice $\Psi(f_\star - h) = \rho$ and by the assumption $\Psi(f_\star) \leq \frac{\rho}{10}$. We can use this in our construction and consider

$$\frac{1}{T} \sum_{t=0}^{T-1} \left\| \hat{f}(X_t) - f_\star(X_t) \right\|_2^2$$

$$\overset{(G.3)}{\leq} \frac{1}{T} \sum_{t=0}^{T-1} 4 \left\langle W_t, \hat{f}(X_t) - f_\star(X_t) \right\rangle_2 - \left\| \hat{f}(X_t) - f_\star(X_t) \right\|_2^2 + 2\lambda_T \left( \Psi(f_\star) - \Psi(\hat{f}) \right)$$

$$\overset{Theorem\ E.1}{\leq} \frac{1}{T} \sum_{t=0}^{T-1} 4R \langle W_t, h(X_t) - f_\star(X_t) \rangle_2 - R^2 \|h(X_t) - f_\star(X_t)\|_2^2 - \frac{R\lambda_T}{4} \left( \Psi(h) - \Psi(f_\star) \right)$$

$$\leq R \left[ \frac{1}{T} \sum_{t=0}^{T-1} 4 \langle W_t, h(X_t) - f_\star(X_t) \rangle_2 - \|h(X_t) - f_\star(X_t)\|_2^2 - \frac{\lambda_T}{4} \left( \Psi(h) - \Psi(f_\star) \right) \right]$$

$$\overset{(H.1)}{\leq} R\left[\frac{1}{T}\sum_{t=0}^{T-1} 4\left\langle W_t,\, h(X_t) - f_\star(X_t)\right\rangle_2 - \|h(X_t) - f_\star(X_t)\|_2^2 - \frac{3\rho\lambda_T}{40}\right].$$

However, by taking $\lambda > 40\mathbf{M}_T\left[\mathscr{F}^\rho\right]/(3\rho)$, the term in the square brackets becomes negative, leading to an absurd statement.

In light of the analysis for cases (i)-(iii), it results that only cases (i) and (ii) are of interest under the assumptions of Theorem 4.1. Therefore, it holds that

$$\left\|\hat{f} - f_\star\right\|_{\mathscr{L}^2(\Omega^T,\mathbb{P}_X;\mathbb{R}^{d_Y})}^2 \leq \min\left\{\theta\mathbf{M}_T\left[\mathscr{F}^\rho\right] + 2\lambda_T\Psi(f_\star),\, r^2\right\}$$

$$\leq \theta\mathbf{M}_T\left[\mathscr{F}^\rho\right] + 2\lambda_T\Psi(f_\star) + r^2,$$

as we wanted to prove. □

## H.2 PROOF OF THEOREM 4.2 (RESULT IN EXPECTATION)

**Theorem 4.2.** Let Assumptions 1 to 3, 5 and 6 hold. Having the set $\partial B(r)$ that is $(C(r), 2)$-hypercontractive, let $\mathscr{F}_r$ be a $r/\sqrt{\theta}$-cover in the infinity norm of $\partial B(r)$, and let $\mathcal{N}_\infty\left(\partial B(r), \frac{r}{\sqrt{\theta}}\right)$ be the associated covering number. Consider the regularized empirical risk minimization problem in (3.3) with regularization parameter satisfying $\lambda_T \geq \frac{40}{3\rho}\mathbb{E}_W\left[\mathbf{M}_T\left[\mathscr{F}^\rho\right]\right]$, where $\rho \geq 10\Psi(f_\star)$. Then, letting $B$ be the positive constant such that $\|f\|_{\mathscr{L}^\infty(\Omega^T;\mathbb{R}^{d_Y})} \leq B$ for all $f \in \mathscr{F}$, we have

$$\mathbb{E}\left[\left\|\hat{f} - f_\star\right\|_{\mathscr{L}^2(\Omega^T,\mathbb{P}_X;\mathbb{R}^{d_Y})}^2\right] \leq 4B^2\mathcal{N}_\infty\left(\partial B(r), \frac{r}{\sqrt{\theta}}\right)\exp\left\{-\frac{8T}{\theta^2 C_r S}\right\}$$

$$+ \theta\mathbb{E}\left[\mathbf{M}_T\left[\mathscr{F}^\rho\right]\right] + \lambda_T\Psi(f_\star) + r^2.$$

*Proof.* First, we observe that $\mathscr{F}_r$ is $(C(r), 2)$-hypercontractive as shown in Theorem D.3, and $B$-boundedness of $\mathscr{F}$ (thus, also of $\mathscr{F}^\rho \subset \mathscr{F}$) follows from Theorem D.1.

We now use the lower isometry event $\mathcal{A}_r$ in (F.3) to decompose the expected value as

$$\mathbb{E}\left[\left\|\hat{f} - f_\star\right\|_{\mathscr{L}^2(\Omega^T,\mathbb{P}_X;\mathbb{R}^{d_Y})}^2\right] = \mathbb{E}\left[\left\|\left(\hat{f} - f_\star\right)\chi_{\mathcal{A}_r}\right\|_{\mathscr{L}^2(\Omega^T,\mathbb{P}_X;\mathbb{R}^{d_Y})}^2\right]$$

$$+ \mathbb{E}\left[\left\|\left(\hat{f} - f_\star\right)\chi_{\mathcal{A}_r^\complement}\right\|_{\mathscr{L}^2(\Omega^T,\mathbb{P}_X;\mathbb{R}^{d_Y})}^2\right], \qquad \text{(H.2)}$$

where $\chi_\mathfrak{A}$ is the indicator function of the event $\mathfrak{A}$, such that it is equal to 1 if $\mathfrak{A}$ is true, and 0 otherwise. To obtain the desired bound, we proceed by analyzing the two addenda separately.

**First scenario ($\mathcal{A}_r$ is true).** In the lower isometry event, we can write

$$\mathbb{E}\left[\left\|\left(\hat{f} - f_\star\right)\chi_{\mathcal{A}_r}\right\|_{\mathscr{L}^2(\Omega^T,\mathbb{P}_X;\mathbb{R}^{d_Y})}^2\right] \leq \left\|\hat{f} - f_\star\right\|_{\mathscr{L}^\infty(\Omega^T;\mathbb{R}^{d_Y})}^2 \mathbb{P}_X(\mathcal{A}_r).$$

Then, we can bound the norm on the right-hand side of such an expression by $(2B)^2$, being

$$\sup_x\left\|\hat{f}(x) - f_\star(x)\right\|_{\mathscr{L}^\infty(\Omega^T;\mathbb{R}^{d_Y})} \leq \sup_x\left(\left\|\hat{f}(x)\right\|_{\mathscr{L}^\infty(\Omega^T;\mathbb{R}^{d_Y})} + \|f_\star(x)\|_{\mathscr{L}^\infty(\Omega^T;\mathbb{R}^{d_Y})}\right) \leq 2B,$$

and the bound for $\mathbb{P}_X(\mathcal{A}_r)$ follows by Theorem F.2. Ultimately, we obtain the bound for the first addendum in (H.2) as

$$\mathbb{E}\left[\left\|\left(\hat{f} - f_\star\right)\chi_{\mathcal{A}_r}\right\|_{\mathscr{L}^2(\Omega^T,\mathbb{P}_X;\mathbb{R}^{d_Y})}^2\right] \leq 4B^2\mathcal{N}_\infty\left(\partial B(r), \frac{r}{\sqrt{\theta}}\right)\exp\left\{-\frac{8T}{\theta^2 C_r S}\right\}. \qquad \text{(H.3)}$$

**Second scenario ($\mathcal{A}_r$ is false).** This case is treated as in the high-probability bound of Theorem 4.1. Again, we express the cases leading to the realization of $\mathcal{A}_r^\complement$ as (i) $\hat{f} \in B(r)$; (ii) $\hat{f} \in \mathscr{F}^\rho \setminus B(r)$, but it happens that $\|\hat{f} - f_\star\|_{\mathscr{L}^2(\Omega^T,\mathbb{P}_X;\mathbb{R}^{d_Y})}^2 \leq \theta\frac{1}{T}\sum_{t=0}^{T-1}\|\hat{f}(X_t) - f_\star(X_t)\|_2^2$; (iii) $\hat{f} \in \mathscr{F} \setminus \mathscr{F}^\rho$. Along the lines of Theorem 4.1, we find an upper bound for the second addendum in (H.2) by showing that, with our choice of $\lambda_T$, case (iii) does not happen.

CASE (I)   When $\hat{f} \in B(r)$, by definition we have that $\mathbb{E}\left[\left\|\hat{f} - f_\star\right\|^2_{\mathscr{L}^2(\Omega^T, \mathbb{P}_X; \mathbb{R}^{d_Y})}\right] \leq r^2$.

CASE (II)   Following the steps in the proof of Theorem 4.1, in this high-probability scenario we have that

$$\mathbb{E}\left[\left\|\hat{f} - f_\star\right\|^2_{\mathscr{L}^2(\Omega^T, \mathbb{P}_X; \mathbb{R}^{d_Y})}\right] \leq \theta \mathbb{E}\left[\mathbf{M}_T\left[\mathscr{F}^\rho\right]\right] + 2\lambda_T \Psi(f_\star).$$

CASE (III)   The argument in the corresponding part of the proof of Theorem 4.1 carries out also when considering the expected value, leading to an absurd conclusion as soon as $\lambda_T \geq \frac{40\mathbb{E}[\mathbf{M}_T[\mathscr{F}^\rho]]}{3\rho}$.

Therefore, overall, the term for the case in which $\mathcal{A}_r$ is false (i.e., the second addendum in (H.2)) is upper-bounded by

$$\mathbb{E}\left[\left\|\left(\hat{f} - f_\star\right)\chi_{\mathcal{A}_r^\complement}\right\|^2_{\mathscr{L}^2(\Omega^T, \mathbb{P}_X; \mathbb{R}^{d_Y})}\right] \leq r^2 + \theta \mathbb{E}\left[\mathbf{M}_T\left[\mathscr{F}^\rho\right]\right] + 2\lambda_T \Psi(f_\star); \tag{H.4}$$

thus, the claim follows by upper-bounding (H.2) by the sum of (H.3) and (H.4). □

# I   PROOFS OF CONVERGENCE RATE RESULTS IN SECTION 5

We now present the proofs of Theorems 5.1 and 5.2. These results build upon Theorems 4.1 and 4.2 and rely on specifying the martingale offset complexity bounds (Section G) and the covering number of the boundary of $B(r)$ (Section D.2). By setting the squared critical radius $r^2$ be dominated by the martingale offset complexity term, we obtain the desired complexity-dependent bounds for the excess risk.

## I.1   PROOF OF THEOREM 5.1 (RESULT IN PROBABILITY)

**Theorem 5.1.** Let Assumptions 1 to 6 hold, and let $\hat{f}$ be the solution of (3.3). Fix a probability of failure $\delta \in (0, 1)$, and assume the regularization parameter $\lambda_T$ satisfies

$$\lambda_T \geq \frac{4}{3T^d}\left[\frac{C_I \sigma_W^{1+d}}{\Psi(f_\star)^{1-\frac{d'}{4}}} + \frac{(C_{II} + C_{IV})\sigma_W^{2d}}{\Psi(f_\star)^{1-\frac{d'}{2}}} + \frac{C_{III}\sigma_W^2 \log(1/\delta)}{\Psi(f_\star)}\right],$$

where $d = 2s/2s+d_X$, $d' = 2d_X/2s+d_X$, and $C_I, C_{II}, C_{III}$ and $C_{IV}$ are constants depending only on $s, d_X, d_Y$ and $\sqrt{\log(1/\delta)}$. If the number of samples $T$ satisfies

$$T \geq \frac{\theta^2 C_h S}{8}\left[C_M \left(\frac{1}{r}\right)^{\frac{6d_X}{2s-d_X}} \log\left(1 + C_L \left(\frac{1}{r}\right)^{\frac{4s-d_X}{2s-d_X}}\right) + \left(\frac{1}{r}\right)^{\frac{4d_X}{2s-d_X}} \log(1/\delta)\right]$$

for $r^2 = \lambda_T \Psi(f_\star) + \sigma_W^2/T$ and $C_h, C_M, C_L$ being uniform constants depending on $\rho_f, \overline{\kappa}, \theta, s, d_X$ and $\Omega$, then, with probability at least $1 - 6\delta$, the excess risk enjoys the following convergence rate:

$$\left\|\hat{f} - f_\star\right\|^2_{\mathscr{L}^2(\Omega^T, \mathbb{P}_X; \mathbb{R}^{d_Y})} \leq C_{\texttt{slow}} \frac{\max\left\{\Psi(f_\star)^{d'/4}, \Psi(f_\star)^{d'/2}\right\}}{T^d} + C_{\texttt{fast}} \frac{\sigma_W^2 \log(1/\delta)}{T},$$

where $C_{\texttt{slow}}$ is a constant that depends on $s, d_X, d_Y, \sigma_W^2, \sqrt{\log(1/\delta)}$, and $C_{\texttt{fast}}$ is a constant that depends on $s, d_X, d_Y$.

*Proof.* The starting point is the bound in probability on the excess risk of Theorem 4.1 given in Equation (4.3). As one of its main ingredients is the bound on the martingale offset complexity, we start the proof by characterizing such a bound reported in Theorem G.2 for the effective hypothesis space $\mathscr{F}^\rho$. A key role is also played by the covering number of $\mathscr{F}^\rho$, which is derived in Theorem C.4. Next, we choose the parameters $\rho, \lambda_T$ and $r^2$ according to the requirements of Theorem 4.1, and this leads to the desired excess risk bound. The proof is concluded by characterizing the lower isometry event probability, which leads to the specification of the burn-in time stated in the claim.

**Martingale offset complexity bound.** We start by determining the bound for $\mathbf{M}_T\left[\mathscr{F}^\rho\right]$ entering (4.3) using the general result of Theorem G.2. By setting $u = \sqrt{2\log(1/\delta)}$ and $v = 2\log(1/\delta)$, we have that, with probability at least $1 - 5\delta$,

$$\mathbf{M}_T\left[\mathscr{F}^\rho\right] \leq \inf_{\gamma > 0}\left\{8\gamma\sqrt{\frac{\sigma_W^2}{T}}(1 + \sqrt{2\log(1/\delta)}) + 8\sqrt{\frac{\sigma_W^2}{T}}\int_0^\gamma \sqrt{\mathcal{N}_\infty\left(\mathscr{F}^\rho, \varepsilon\right)}d\varepsilon\right.$$
$$\left. + \frac{64\sigma_W^2\log(1/\delta)}{T} + \frac{32\sigma_W^2}{T}\mathcal{N}_\infty\left(\mathscr{F}^\rho, \gamma\right) + 4\gamma^2\right\}.$$

By using the covering number result in Theorem C.4 and noting that, according to (G.22),

$$\int_0^\gamma \left(\frac{1}{\varepsilon}\right)^{\frac{d_X}{2s}}d\varepsilon = \frac{\gamma^{1 - d_X/2s}}{1 - d_X/2s}, \tag{I.1}$$

the bound on the martingale offset complexity can be re-written as

$$\mathbf{M}_T\left[\mathscr{F}^\rho\right] \leq \inf_{\gamma > 0}\left\{8\gamma\sqrt{\frac{\sigma_W^2}{T}}(1 + \sqrt{2\log(1/\delta)}) + 8\sqrt{\frac{\sigma_W^2}{T}}\frac{\sqrt{C_c}d_Y^{\frac{2s+d_X}{4s}}}{1 - \frac{d_X}{2s}}(\sqrt{\rho})^{\frac{d_X}{2s}}\gamma^{1 - d_X/2s}\right.$$
$$\left. + \frac{64\sigma_W^2\log(1/\delta)}{T} + \frac{32\sigma_W^2}{T}C_c d_Y^{\frac{2s+d_X}{2s}}\left(\frac{\sqrt{\rho}}{\gamma}\right)^{\frac{d_X}{s}} + 4\gamma^2\right\}. \tag{I.2}$$

By following the reasoning presented in Liang et al. (2015) (see also Yang & Barron (1999)), minimization over $\gamma$ is obtained by equating the last two terms in (I.2), which yields

$$\gamma = \left(8C_c d_Y^{\frac{2s+d_X}{2s}}\right)^{\frac{s}{2s+d_X}}\left(\frac{\sigma_W^2}{T}\right)^{\frac{s}{2s+d_X}}(\sqrt{\rho})^{\frac{d_X}{2s+d_X}}.$$

Plugging in such a value for $\gamma$ in (I.2), and recalling the definitions $d \doteq {}^{2s}/_{2s+d_X}$ and $d' = {}^{2d_X}/_{2s+d_X}$, we obtain

$$\mathbf{M}_T\left[\mathscr{F}^\rho\right] \leq C_I\frac{\sigma_W^{1+d}}{T^{\frac{1+d}{2}}}(\sqrt{\rho})^{\frac{d'}{2}} + C_{II}\frac{(\sigma_W^2)^d}{T^d}(\sqrt{\rho})^{d'} + C_{III}\frac{\sigma_W^2\log(1/\delta)}{T} + C_{IV}\frac{(\sigma_W^2)^d}{T^d}(\sqrt{\rho})^{d'},$$

$$\text{where } \begin{cases} C_I & \doteq 8(1 + \sqrt{2\log(1/\delta)})\left(8C_c d_Y^{1/d}\right)^{d/2} \\ C_{II} & \doteq \frac{2s}{2s-d_X}8\sqrt{C_c d_Y^{1/d}}\left(8C_c d_Y^{1/d}\right)^{\frac{2s-d_X}{2(2s+d_X)}} \\ C_{III} & \doteq 64 \\ C_{IV} & \doteq 8\left(8C_c d_Y^{1/d}\right)^d \end{cases} \tag{I.3}$$

**Choice of the parameters $\rho$, $\lambda_T$ and $r$.** According to Theorem 4.1, we set the radius of the effective hypothesis class $\mathscr{F}^\rho$ to satisfy $\rho = 10\Psi(f_\star)$; similarly, the regularization parameter is chosen as $\lambda_T = \frac{40}{3\rho}\mathbf{M}_T\left[\mathscr{F}^\rho\right]$. Regarding the radius of the $\mathscr{L}^2$-ball $B(r)$, we conveniently set it as $r^2 = \lambda_T\Psi(f_\star) + \frac{\sigma_W^2\log(1/\delta)}{T}$. Thanks to these choices, the excess risk bound in (4.3) reads as

$$\left\|\hat{f} - f_\star\right\|_{\mathscr{L}^2(\Omega^T, \mathbb{P}_X; \mathbb{R}^{d_Y})}^2 \leq (\theta + 4)\left(C_I\frac{\sigma_W^{1+d}}{T^{\frac{1+d}{2}}}(\sqrt{\rho})^{\frac{d'}{2}} + C_{II}\frac{(\sigma_W^2)^d}{T^d}(\sqrt{\rho})^{d'}\right.$$
$$\left. + C_{III}\frac{\sigma_W^2\log(1/\delta)}{T} + C_{IV}\frac{(\sigma_W^2)^d}{T^d}(\sqrt{\rho})^{d'}\right) + \frac{\sigma_W^2\log(1/\delta)}{T}.$$

Now, noting that $(1+d)/2 > d$, we obtain the desired claim, namely that

$$\left\|\hat{f} - f_\star\right\|_{\mathscr{L}^2(\Omega^T, \mathbb{P}_X; \mathbb{R}^{d_Y})}^2 \leq C_{\texttt{slow}}\frac{\max\left\{\Psi(f_\star)^{d'/4}, \Psi(f_\star)^{d'/2}\right\}}{T^d} + C_{\texttt{fast}}\frac{\sigma_W^2\log(1/\delta)}{T},$$

$$\text{where } \begin{cases} C_{\texttt{slow}} & \doteq (\theta + 4)\left(C_I 10^{d'/4}\sigma_W^{1+d} + C_{II}10^{d'/2}\sigma_W^{2d} + C_{IV}\sigma_W^{2d}10^{d'/2}\right) \\ C_{\texttt{fast}} & \doteq (\theta + 4)C_{III} + 1. \end{cases}$$

**Characterization of the burn-in time.** We conclude the proof by setting the probability of the lower isometry event $\mathcal{A}_r$ equal to $\delta$, so that the overall claim can hold with the desired probability $1 - 5\delta - \delta$.

By Theorem F.3, we have the following bound for the probability of the lower isometry event:

$$
\mathbb{P}_X(\mathcal{A}_r) \leq \left( C_L \left( \frac{1}{r} \right)^{\frac{4s-d_X}{2s-d_X}} + 1 \right)^{d_Y C_m \left( \frac{1}{r} \right)^{\frac{2d_X}{2s-d_X}}} \exp \left\{ -\frac{8Tr^{\frac{4d_X}{2s-d_X}}}{\theta^2 C_h S} \right\} \overset{!}{\leq} \delta.
$$

Taking logarithms on both sides of the last inequality, letting $C_M \doteq C_m d_Y$, we obtain that $T$ has to satisfy the condition

$$
T \geq \frac{\theta^2 C_h S}{8} \left[ C_M \left( \frac{1}{r} \right)^{\frac{6d_X}{2s-d_X}} \log \left( 1 + C_L \left( \frac{1}{r} \right)^{\frac{4s-d_X}{2s-d_X}} \right) + \left( \frac{1}{r} \right)^{\frac{4d_X}{2s-d_X}} \log(1/\delta) \right].
$$

The effective condition is obtained by substituting $r^2 = \lambda_T \Psi(f_\star) + \sigma_W^2/T$. $\qquad\square$

## I.2 PROOF OF THEOREM 5.2 (RESULT IN EXPECTATION)

**Theorem 5.2.** Let Assumptions 1 to 6 hold, and let $\hat{f}$ be the solution of (3.3) with regularization parameter $\lambda_T$ satisfying

$$
\lambda_T \geq \frac{4(C_I + C_{II})(\sigma_W^2)^d}{3T\Psi(f_\star)^{1-\frac{d'}{2}}},
$$

where $d = {}^{2s}/_{2s+d_X}$ is the Sobolev minimax rate, $d' = {}^{2d_X}/_{2s+d_X}$, and $C_I$ and $C_{II}$ are constants depending only on $s, d_X$ and $d_Y$. If the amount of samples $T$ satisfies

$$
T \geq \frac{\theta^2 C_h S}{8} \left( \frac{1}{r} \right)^{\frac{4d_X}{2s-d_X}} \left[ C_M \left( \frac{1}{r} \right)^{\frac{2d_X}{2s-d_X}} \log \left( 4B^2 \left( 1 + C_L \left( \frac{1}{r} \right)^{\frac{4s-d_X}{2s-d_X}} \right) \right) + \log \left( \frac{\sigma_W^2}{T} \right) \right],
$$

where $B$ is such that $\|f\|_{\mathscr{L}^\infty(\Omega^T;\mathbb{R}^{d_Y})} \leq B$ for all $f \in \mathscr{F}$ and $C_M, C_h, C_L$ are constants depending on $\rho_f, \overline{\kappa}, \theta, s, d_X$ and $\Omega$, then the excess risk enjoys the following convergence rate:

$$
\mathbb{E} \left[ \left\| \hat{f} - f_\star \right\|^2_{\mathscr{L}^2(\Omega^T, \mathbb{P}_X; \mathbb{R}^{d_Y})} \right] \leq C_{\text{slow}} \frac{\Psi(f_\star)^{d'/2}}{T^d} + C_{\text{fast}} \frac{\sigma_W^2}{T},
$$

where $C_{\text{slow}}$ and $C_{\text{fast}}$ are constants that depend on $s, d_X, d_Y$ and $\sigma_W^2$.

*Proof.* Similarly to the proof of Theorem 5.1, we start by characterizing the bound on the expected value of the martingale offset complexity of $\mathscr{F}^\rho$. Next, by choosing the parameters $\rho$, $\lambda_T$ and $r$ according to the requirements of Theorem 4.2, we arrive to the desired claim on the bound. Finally, we discuss the burn-in time by characterizing the lower-isometry event probability.

**Martingale offset complexity bound.** As stated in Theorem G.3, we have that

$$
\mathbb{E}\left[\mathbf{M}_T\left[\mathscr{F}^\rho\right]\right] \leq \inf_{\gamma>0} 8\sqrt{\frac{\sigma_W^2}{T}} \int_0^\gamma \sqrt{\log \mathcal{N}_\infty \left(\mathscr{F}^\rho, \varepsilon\right)} d\varepsilon + \frac{32\sigma_W^2 \log \mathcal{N}_\infty \left(\mathscr{F}^\rho, \gamma\right)}{T} + 4\gamma^2.
$$

Leveraging C.4 to characterize the metric entropy of $\mathscr{F}^\rho$ and leveraging (I.1), such a bound can be re-written as

$$
\mathbb{E}\left[\mathbf{M}_T\left[\mathscr{F}^\rho\right]\right] \leq \inf_{\gamma>0} 8\sqrt{\frac{\sigma_W^2}{T}} \frac{\sqrt{C_c d_Y^{\frac{2s+d_X}{2s}}}}{1 - d_X/2s} (\sqrt{\rho})^{\frac{d_X}{2s}} \gamma^{1-\frac{d_X}{2s}} + \frac{32\sigma_W^2}{T} (C_c d_Y^{\frac{2s+d_X}{2s}}) \left( \frac{\sqrt{\rho}}{\gamma} \right)^{\frac{d_X}{s}} + 4\gamma^2.
$$

As done in Liang et al. (2015); Yang & Barron (1999), we minimize the right-hand side by balancing the last two addenda, which leads to

$$
\gamma = \left( 8C_c d_Y^{\frac{2s+d_X}{2s}} \right)^{\frac{s}{2s+d_X}} \left( \frac{\sigma_W^2}{T} \right)^{\frac{s}{2s+d_X}} (\sqrt{\rho})^{\frac{d_X}{2s+d_X}}.
$$

By substituting such a value of $\gamma$ in the martingale offset complexity bound, we obtain that

$$\mathbb{E}\left[\mathbf{M}_T\left[\mathscr{F}^\rho\right]\right] \leq (C_I + C_{II})\left(\frac{\sigma_W^2}{T}\right)^d (\sqrt{\rho})^{d'},$$

where we recall that $d = {}^{2s}/_{2s+d_X}$ and $d' = {}^{2d_X}/_{2s+d_X}$, and the constants $C_I$ and $C_I I$ are equal to

$$\begin{cases} C_I &\doteq \frac{8\sqrt{C_c d_Y^d}}{1-d_X/2s}\left(8C_c d_Y^{\frac{1}{d}}\right)^{\frac{2s-d_X}{2(2s+d_X)}} \\ C_{II} &\doteq 8\left(8C_c d_Y^{\frac{1}{d}}\right)^d \end{cases}$$

**Choosing parameters $\rho$, $\lambda_T$ and $r$.** We proceed by following the requirements of Theorem 4.2, setting $\rho = 10\Psi(f_\star)$ and $\lambda_T = \frac{40}{3\rho}\mathbb{E}\left[\mathbf{M}_T\left[\mathscr{F}^\rho\right]\right]$. Furthermore, setting $r^2 = 2\lambda_T \Psi(f_\star) + \frac{\sigma_W^2}{T}$, we obtain that the desired bound reads as

$$\mathbb{E}\left[\left\|\hat{f} - f_\star\right\|^2_{\mathscr{L}^2(\Omega^T, \mathbb{P}_X; \mathbb{R}^{d_Y})}\right] \leq 4B^2 \mathcal{N}_\infty\left(\partial B(r), \frac{r}{\sqrt{\theta}}\right)\exp\left\{-\frac{8T}{\theta^2 C(r)S}\right\}$$

$$+ (\theta + 4)10^{\frac{d'}{2}}(C_I + C_{II})\left(\frac{\sigma_W^2}{T}\right)^d \Psi(f_\star)^{\frac{d'}{2}} + \frac{\sigma_W^2}{T}. \quad \text{(I.4)}$$

**Characterizing the burn-in.** We conclude the proof by imposing that the first term on the right-hand side of (I.4) is upper-bounded by $\frac{\sigma_W^2}{T}$, i.e.,

$$4B^2 \mathcal{N}_\infty\left(\partial B(r), \frac{r}{\sqrt{\theta}}\right)\exp\left\{-\frac{8T}{\theta^2 C(r)S}\right\} \leq \frac{\sigma_W^2}{T}.$$

Leveraging Theorem F.3, we deploy the values for the covering number and the hypercontractivity parameter and obtain that the number of samples $T$ has to satisfy

$$T \geq \frac{\theta^2 C_h S}{8}\left(\frac{1}{r}\right)^{\frac{4d_X}{2s-d_X}}\left[C_M\left(\frac{1}{r}\right)^{\frac{2d_X}{2s-d_X}}\log\left(4B^2\left(1 + C_L\left(\frac{1}{r}\right)^{\frac{4s-d_X}{2s-d_X}}\right)\right) + \log\left(\frac{\sigma_W^2}{T}\right)\right].$$

The effective condition for the burn-in is obtained by letting $r^2 = 2\lambda_T \Psi(f_\star) + \sigma_W^2/T$. Ultimately, with such a choice of $T$, we obtain that

$$\mathbb{E}\left[\left\|\hat{f} - f_\star\right\|^2_{\mathscr{L}^2(\Omega^T, \mathbb{P}_X; \mathbb{R}^{d_Y})}\right] \leq C_{\texttt{slow}}\frac{\Psi(f_\star)^{\frac{d'}{2}}}{T^d} + C_{\texttt{fast}}\frac{\sigma_W^2}{T},$$

where the constants read as

$$\begin{cases} C_{\texttt{slow}} &\doteq (\theta + 4)10^{\frac{d'}{2}}(C_I + C_{II})(\sigma_W^2)^d \\ C_{\texttt{fast}} &\doteq 2. \end{cases}$$

$\square$

## J    RESULTS FOR THE CASE WITHOUT PHYSICS-INFORMED REGULARIZATION

We now derive the bounds in the situation in which there is no physics-informed regularization (i.e., $\lambda_T = 0$ in (3.3)). The obtained bounds will be the benchmark against which we compare Theorems 5.1 and 5.2, showing the impact of knowledge alignment in the excess risk bounds to obtain faster convergence.

We start by recalling that, when physics-informed regularization is absent, the empirical risk minimization problem (3.3) reads as

$$\hat{f}' = \arg\min_{f \in \mathscr{F}} \frac{1}{T}\sum_{t=0}^{T-1}\|Y_t - f(X_t)\|_2^2, \quad \text{(J.1)}$$

and the lower isometry event takes this expression:

$$\mathcal{A}'_r \doteq \sup_{f \in \mathscr{F}_\star \backslash B(r)}\left\{\frac{1}{T}\sum_{t=0}^{T-1}\|f(X_t)\|_2^2 - \frac{1}{\theta}\|f\|^2_{\mathscr{L}^2(\Omega^T, \mathbb{P}_X; \mathbb{R}^{d_Y})} \leq 0\right\}. \quad \text{(J.2)}$$

## J.1 COROLLARY OF THEOREM 5.1 (RESULT IN PROBABILITY)

**Corollary J.1.** *Let Assumptions 1 to 3, 5 and 6 hold, and let $\hat{f}'$ be the solution of the estimation problem* (J.1). *Setting $\theta > 8$, if*

$$T \geq \frac{\theta^2 C_h S}{8} \left[ C_M \left( \frac{1}{r} \right)^{\frac{6d_X}{2s-d_X}} \log \left( 1 + C_L \left( \frac{1}{r} \right)^{\frac{4s-d_X}{2s-d_X}} \right) + \left( \frac{1}{r} \right)^{\frac{4d_X}{2s-d_X}} \log(1/\delta) \right],$$

*where $C_h, C_M, C_L$ are uniform constants, then the excess risk satisfies*

$$\left\| \hat{f}' - f_\star \right\|_{\mathscr{L}^2(\Omega^T, \mathbb{P}_X; \mathbb{R}^{d_Y})}^2 \leq C'_{slow} \frac{1}{T^d} + C'_{fast} \frac{\sigma_W^2}{T},$$

*with probability at least $1 - 6\delta$, where $C'_{slow}$ is a constant that depends only on $s, d_X, d_Y, \rho_f, \sigma_W^2, \theta$ and $\sqrt{\log(1/\delta)}$, and $C'_{fast}$ is a constant that depends only on $\theta$.*

*Proof.* We first adapt Theorem 4.1 to the non-regularized case, and then derive the bounds along the lines of Theorem 5.1.

**Expression for the bound.** By adapting Theorem 4.1 to the lower-isometry event (J.2), we have that the event $(\mathcal{A}'_r)^\complement$ happens in two scenarios:

(i) $\hat{f}' \in B(r) \implies \left\| \hat{f}' - f_\star \right\|_{\mathscr{L}^2(\Omega^T, \mathbb{P}_X; \mathbb{R}^{d_Y})}^2 \leq r^2$;

(ii) $\hat{f} - f_\star$ belong to $\mathscr{F} \setminus B(r)$, but it holds that

$$\left\| \hat{f}' - f_\star \right\|_{\mathscr{L}^2(\Omega^T, \mathbb{P}_X; \mathbb{R}^{d_Y})}^2 \leq \frac{\theta}{T} \sum_{t=0}^{T-1} \left\| \hat{f}'(X_t) - f_\star(X_t) \right\|_2^2 \leq \theta \mathbf{M}_T[\mathscr{F}].$$

Therefore, on $(\mathcal{A}'_r)^\complement$, we have that

$$\left\| \hat{f}' - f_\star \right\|_{\mathscr{L}^2(\Omega^T, \mathbb{P}_X; \mathbb{R}^{d_Y})}^2 \leq \theta \mathbf{M}_T[\mathscr{F}] + r^2. \tag{J.3}$$

**Bounding the martingale offset complexity.** We can now rely on Theorem G.2 to bound the martingale offset complexity: setting $u = \sqrt{2\log(1/\delta)}$ and $v = 2\log(1/\delta)$, and using Theorem C.3 to characterize the metric entropy of the hypothesis space $\mathscr{F}$, we have that

$$\mathbf{M}_T[\mathscr{F}] \leq \inf_{\gamma > 0} 8 \sqrt{\frac{\sigma_W^2}{T}} (\sqrt{2\log(1/\delta)} + 1) + 8 \sqrt{\frac{\sigma_W^2}{T}} \frac{\sqrt{C'_c d_Y^{\frac{2s+d_X}{2s}}}}{1 - d_X/2s} \rho_f^{\frac{d_X}{2s}} \gamma^{\frac{2s-d_X}{2s}}$$

$$+ \frac{64\sigma_W^2 \log(1/\delta)}{T} + \frac{32\sigma_W^2}{T} C'_c d_Y^{\frac{2s+d_X}{2s}} \left( \frac{\rho_f}{\gamma} \right)^{\frac{d_X}{s}} + 4\gamma^2. \tag{J.4}$$

As done in Theorem 5.1, we balance the last two addenda to get

$$\gamma = \left( 8 C'_c d_Y^{\frac{2s+d_X}{2s}} \right)^{\frac{s}{2s+d_X}} \rho_f^{\frac{d_X}{2s+d_X}} \left( \frac{\sigma_W^2}{T} \right)^{\frac{s}{2s+d_X}}.$$

Substituting into (J.4) and recalling the definition of the Sobolev minimax rate $d = 2s/(2s + d_X)$, we obtain

$$\mathbf{M}_T[\mathscr{F}] \leq C'_I \left( \frac{\sigma_W^2}{T} \right)^{\frac{d+1}{2}} + C'_{II} \left( \frac{\sigma_W^2}{T} \right)^d + C'_{III} \log(1/\delta) \frac{\sigma_W^2}{T} + C'_{IV} \left( \frac{\sigma_W^2}{T} \right)^d,$$

$$
\text{where} \quad
\begin{cases}
C'_I & \doteq 8 \left( 8 C'_c d_Y^{\frac{2s+d_X}{2s}} \right)^{\frac{s}{2s+d_X}} \left( 1 + \sqrt{2 \log(1/\delta)} \right) \rho_f^{\frac{d_X}{2s+d_X}} \\[2ex]
C'_{II} & \doteq 8 \frac{\sqrt{C'_c d_Y^{\frac{2s+d_X}{2s}}}}{1 - d_X/2s} \left( 8 C'_c d_Y^{\frac{2s+d_X}{2s}} \right)^{\frac{2s-d_X}{2(2s+d_X)}} \rho_f^{\frac{2d_X}{2s+d_X}} \\[2ex]
C'_{III} & \doteq 64 \\[1ex]
C'_{IV} & \doteq 8 \left( 8 C'_c d_Y^{\frac{2s+d_X}{2s}} \right)^{\frac{2s}{2s+d_X}} \rho_f^{\frac{2d_X}{2s+d_X}}.
\end{cases}
$$

**Final expression for the bound.** Going back to (J.3), noting that $T^{-(d+1)/2} < T^{-d}$ and setting $r^2 \leq \sigma_W^2/T$, we obtain the bound for the excess risk

$$
\left\| \hat{f}' - f_\star \right\|^2_{\mathscr{L}^2(\Omega^T, \mathbb{P}_X; \mathbb{R}^{d_Y})} \leq \frac{C'_{\texttt{slow}}}{T^d} + \frac{C'_{\texttt{fast}} \sigma_W^2 \log(1/\delta)}{T}
$$

where $C'_{\texttt{slow}} = \theta(C'_I + C'_{II} + C'_{IV}) \max\{\sigma_W^{1+d}, \sigma_W^{2d}\}$ and $C'_{\texttt{fast}} = 1 + (\theta) C'_{III}$.

**Characterization of the burn-in.** Similarly to the derivation in Theorem 5.1, the burn-in condition consists in having the amount $T$ satisfying

$$
T \geq \frac{\theta^2 C_h S}{8} \left[ C_M \left( \frac{1}{r} \right)^{\frac{6 d_X}{2s - d_X}} \log \left( 1 + C_L \left( \frac{1}{r} \right)^{\frac{4s - d_X}{2s - d_X}} \right) + \left( \frac{1}{r} \right)^{\frac{4 d_X}{2s - d_X}} \log(1/\delta) \right]
$$

where $r^2 \leq \sigma_W^2/T$. $\qquad\square$

## J.2 COROLLARY OF THEOREM 5.2 (RESULT IN EXPECTATION)

**Corollary J.2.** *Let Assumptions 1 to 3, 5 and 6 hold, and let $\hat{f}'$ be the solution of the estimation problem* (J.1). *Let $B$ the infinity-norm bound of functions in $\mathscr{F}$ and $\theta > 8$. If the number of samples $T$ satisfies*

$$
T \geq \frac{\theta^2 C_h S}{8} \left( \frac{1}{r} \right)^{\frac{4 d_X}{2s - d_X}} \left[ C_M \left( \frac{1}{r} \right)^{\frac{2 d_X}{2s - d_X}} \log \left( 4 B^2 \left( 1 + C_L \left( \frac{1}{r} \right)^{\frac{4s - d_X}{2s - d_X}} \right) \right) + \log \left( \frac{\sigma_W^2}{T} \right) \right]
$$

*with $r \leq \sqrt{\sigma_W^2/T}$ and uniform constants $C_h, C_L, C_M$, then we have that*

$$
\mathbb{E} \left[ \left\| \hat{f}' - f_\star \right\|^2_{\mathscr{L}^2(\Omega^T, \mathbb{P}_X; \mathbb{R}^{d_Y})} \right] \leq \frac{C'_{\texttt{slow}}}{T^d} + C'_{\texttt{fast}} \frac{\sigma_W^2}{T}
$$

*Proof.* Similarly to the previous Corollary, we first adapt Theorem 4.2 to the non-regularized case and then derive the bounds following Theorem 5.2.

**Expression for the bound.** Considering the lower-isometry event (J.2), we decompose the expected value as follows:

$$
\mathbb{E} \left[ \left\| \hat{f}' - f_\star \right\|^2_{\mathscr{L}^2(\Omega^T, \mathbb{P}_X; \mathbb{R}^{d_Y})} \right] = \overbrace{\mathbb{E} \left[ \left\| \left( \hat{f}' - f_\star \right) \chi_{\mathcal{A}'_r} \right\|^2_{\mathscr{L}^2(\Omega^T, \mathbb{P}_X; \mathbb{R}^{d_Y})} \right]}^{(I)}
$$
$$
+ \underbrace{\mathbb{E} \left[ \left\| \left( \hat{f}' - f_\star \right) \chi_{(\mathcal{A}'_r)^{\complement}} \right\|^2_{\mathscr{L}^2(\Omega^T, \mathbb{P}_X; \mathbb{R}^{d_Y})} \right]}_{(II)} \qquad (\text{J.5})
$$

along the lines of the proof of Theorem 4.2. Looking at the two addenda separately:

(I) when $\mathcal{A}'_r$ is true, we have that

$$
\mathbb{E} \left[ \left\| \left( \hat{f} - f_\star \right) \chi_{\mathcal{A}'_r} \right\|^2_{\mathscr{L}^2(\Omega^T, \mathbb{P}_X; \mathbb{R}^{d_Y})} \right] \leq \left\| \hat{f}' - f_\star \right\|^2_{\mathscr{L}^\infty(\Omega^T; \mathbb{R}^{d_Y})} \mathbb{P}_X(\mathcal{A}'_r)
$$

$$\overset{Theorem\ F.2}{\leq} 4B^2 \mathcal{N}_\infty \left( \partial B(r), \frac{r}{\sqrt{\theta}} \right) \exp \left\{ -\frac{8T}{\theta^2 C(r)S} \right\}$$

by bounding the worst-case distance between $\hat{f}'$ and $f_\star$.

(II) when $\mathcal{A}'_r$ is false, there are two scenarios possible: (i) $\hat{f}' \in B(r)$; or (ii) $\hat{f} \in \mathscr{F} \setminus B(r)$, but it happens that $\|\hat{f}' - f_\star\|^2_{\mathscr{L}^2} \leq \frac{\theta}{T} \sum_{t=0}^{T-1} \|\hat{f}'(X_t) - f_\star(X_t)\|^2_2$. Looking at these two cases:

  (i) $\|\hat{f}' - f_\star\|^2_{\mathscr{L}^2(\Omega^T, \mathbb{P}_X; \mathbb{R}^{d_Y})} \leq r^2$ by definition;

  (ii) $\|\hat{f}' - f_\star\|^2_{\mathscr{L}^2(\Omega^T, \mathbb{P}_X; \mathbb{R}^{d_Y})} \leq \theta \mathbf{M}_T [\mathscr{F}]$.

Bringing all these terms together, the bound in (J.5) reads as

$$\mathbb{E}\left[ \left\| \hat{f}' - f_\star \right\|^2_{\mathscr{L}^2(\Omega^T, \mathbb{P}_X; \mathbb{R}^{d_Y})} \right] \leq 4B^2 \mathcal{N}_\infty \left( \partial B(r), \frac{r}{\sqrt{\theta}} \right) \exp \left\{ -\frac{8T}{\theta^2 C(r)S} \right\} \\ + \theta \mathbb{E}\left[ \mathbf{M}_T [\mathscr{F}] \right] + r^2. \tag{J.6}$$

**Bounding the martingale offset complexity.** We can proceed by upper-bounding the martingale offset complexity term by deploying Theorem G.3. Specifically, we have that

$$\mathbb{E}\left[ \mathbf{M}_T [\mathscr{F}] \right] \leq \inf_{\gamma > 0} 8 \sqrt{\frac{\sigma_W^2}{T}} \int_0^\gamma \sqrt{\log \mathcal{N}_\infty (\mathscr{F}, \varepsilon)} d\varepsilon + \frac{32\sigma_W^2 \log \mathcal{N}_\infty (\mathscr{F}, \gamma)}{T} + 4\gamma^2.$$

We deploy Theorem C.3 to provide the expression for the metric entropy of $\mathscr{F}$ (see also the proof of Theorem 5.2) and obtain

$$\mathbb{E}\left[ \mathbf{M}_T [\mathscr{F}] \right] \leq \inf_{\gamma > 0} 8 \sqrt{\frac{\sigma_W^2}{T}} \frac{\sqrt{C'_c d_Y^{\frac{2s+d_X}{2s}}}}{1 - d_X/2s} \rho_f^{\frac{d_X}{2s}} \gamma^{\frac{2s-d_X}{2s}} + \frac{32\sigma_W^2}{T} C'_c d_Y^{\frac{2s+d_X}{2s}} \left( \frac{\rho_f}{\gamma} \right)^{\frac{d_X}{s}} + 4\gamma^2$$

Balancing the last two terms of the right-hand side, we obtain

$$\gamma = \left( 8C'_c d_Y^{\frac{2s+d_X}{2s}} \rho_f^{\frac{d_X}{s}} \right)^{\frac{s}{2s+d_X}} \left( \frac{\sigma_W^2}{T} \right)^{\frac{s}{2s+d_X}},$$

which we substitute back in the expression of the martingale offset complexity to get (recalling that $d = 2s/(2s + d_X)$ is the Sobolev minimax exponent)

$$\mathbb{E}\left[ \mathbf{M}_T [\mathscr{F}] \right] \leq \frac{C'_I}{T^d} + \frac{C'_{II}}{T^d},$$

$$\text{where} \quad \begin{cases} C'_I & \doteq 8 \frac{\sqrt{C'_c d_Y^{\frac{2s+d_X}{2s}}}}{1 - d_X/2s} \left( 8C'_c d_Y^{\frac{2s+d_X}{2s}} \rho_f^{\frac{d_X}{s}} \right)^{\frac{s}{2s+d_X}} \rho_f^{\frac{d_X}{s} + \frac{d_X}{2s+d_X}} \\ C'_{II} & \doteq 8 \left( 8C'_c d_Y^{\frac{2s+d_X}{2s}} \rho_f^{\frac{d_X}{s}} \right)^{\frac{2s}{2s+d_X}}. \end{cases} \tag{J.7}$$

**Final expression for the bound.** We can now go back to the excess risk bound (J.6). We let $r^2 \leq \sigma_W^2/T$, substitute (J.7) in the expected value for the martingale offset complexity, and let the first addendum in (J.6) be upper-bounded by $\frac{\sigma_W^2}{T}$. Ultimately, this yields

$$\mathbb{E}\left[ \left\| \hat{f}' - f_\star \right\|^2_{\mathscr{L}^2(\Omega^T, \mathbb{P}_X; \mathbb{R}^{d_Y})} \right] \leq \frac{C'_{\text{slow}}}{T^d} + C'_{\text{fast}} \frac{\sigma_W^2}{T},$$

$$\text{where} \quad \begin{cases} C'_{\text{slow}} & \doteq \theta(C'_I + C'_{II})(\sigma_W^2)^d \\ C'_{\text{fast}} & \doteq 2. \end{cases}$$

**Characterization of the burn-in.** The bound has been obtained by imposing that the first addendum in (J.6) satisfies

$$4B^2 \mathcal{N}_\infty \left( \partial B(r), \frac{r}{\sqrt{\theta}} \right) \exp \left\{ -\frac{8T}{\theta^2 C(r) S} \right\} \leq \frac{\sigma_W^2}{T}.$$

Leveraging Theorem F.3, we deploy the values for the covering number and the hypercontractivity parameter and obtain that the number of samples $T$ has to satisfy

$$T \geq \frac{\theta^2 C_h S}{8} \left( \frac{1}{r} \right)^{\frac{4d_X}{2s-d_X}} \left[ C_M \left( \frac{1}{r} \right)^{\frac{2d_X}{2s-d_X}} \log \left( 4B^2 \left( 1 + C_L \left( \frac{1}{r} \right)^{\frac{4s-d_X}{2s-d_X}} \right) \right) + \log \left( \frac{\sigma_W^2}{T} \right) \right]$$

The effective condition for the burn-in is obtained by letting $r^2 = \sigma_W^2 / T$. $\qquad \square$

## K    Addendum to Numerical Tests

In the following, we provide in Section K.1 the full details of the numerical test presented in Section 6, also providing more comments on the burn-in, the choice of the regularization parameter $\lambda_T$, and the impact of misaligned priors. Additionally, we present in Section K.2 an additional numerical example, in which the prior is given by the Poisson equation.

### K.1    Unicycle experiment

**Model set-up.** We consider a nonlinear dynamical system that describes the dynamics of a unicycle robot. The ground-truth model is given by

$$\dot{x}_1(t) = \nu(t) \cos \vartheta(t),$$
$$\dot{x}_2(t) = \nu(t) \sin \vartheta(t)$$
$$\dot{\vartheta}(t) = \omega(t),$$

where $(x_1, x_2) \in \mathbb{R}^2$ is the position of the robot on the plane, $\vartheta \in [0, \pi/2]$ is the orientation angle, and $(\nu, \omega)$ are the translational and angular velocities, respectively.
We simulate the continuous dynamics using a forward Euler discretization with step size $dt = 0.05$, yielding the discrete-time dynamical system

$$x_{1,t+1} = x_{1,t} + \nu_t \cos(\vartheta_t) dt,$$
$$x_{2,t+1} = x_{2,t} + \nu_t \sin(\vartheta_t) dt,$$
$$\vartheta_{t+1} = \vartheta_t + \omega_t dt.$$

We generate training pairs $\{(s_t, u_t), s_{t+1}\}$ where $s_t = (x_{1,t}, x_{2,t}, \vartheta_t)$ and $u_t = (\nu_t, \omega_t)$, corrupted by an additive Gaussian noise on $s_{t+1}$ with variance $\sigma_W^2 = 1$.

**Physics-informed regularization.** The unicycle kinematics enforce that the velocity has no lateral component. This constraint takes the form

$$q(s_t, u_t) = (x_{2,t+1} - x_{2,t}) \cos \vartheta_t - (x_{1,t+1} - x_{1,t}) \sin \vartheta_t = 0.$$

To promote models consistent with the physics, we penalize the squared $\mathscr{L}^2$-norm of this residual over a compact domain of states and inputs, i.e.,

$$\|q\|_{\mathscr{L}^2(\Omega)}^2 = \int_\Omega q(\xi)^2 d\xi, \text{ with } \xi = (s, u).$$

We approximate the above integral with Monte Carlo sampling from the input domain $\Omega$. For each mini-batch, we evaluate the residuals under the adopted predictor $g_\theta(s_t, u_t)$ (which we specify below) and compute

$$\widehat{\|q\|}_{\mathscr{L}^2(\Omega)}^2 = \frac{\mu_\lambda(\Omega)}{N} \sum_{i=1}^N q(\xi_i)^2, \text{ with } \xi_i \text{ uniformly sampled from } \Omega.$$

The total loss combines the data mean squared error and the physics-informed penalty according to (3.3). This ensures the learned predictor both fits noisy data and respects the no-slip constraint.

**Adopted estimator.** We use a feedforward neural network $g_\theta \colon (s_t, u_t) \in \mathbb{R}^5 \to \hat{s}_{t+1} \in \mathbb{R}^3$ to approximate the discrete-time dynamics. The architecture is a two-hidden-layer multilayer perceptron (MLP) with ReLU activation function and 64 inner nodes.

We train the estimator using the Adam optimizer with learning rate $0.5 \times 10^{-3}$ and batch size $N = 200$. We vary the effective number of training samples $T$ over the range $T \in [300, 10^6]$, and compute the average learning rate after the burn-in.

**Experiment regimes.** We compare three learning regimes:

1. **Without knowledge**: $\lambda_T = 0$.

2. **With knowledge**: the correct non-slip operator is enforced,
$$q(s_t, u_t) = (x_{2,t+1} - x_{2,t}) \cos \vartheta_t - (x_{1,t+1} - x_{1,t}) \sin \vartheta_t.$$

3. **Misaligned knowledge**: an incorrect operator with two sources of error is enforced. First, we use a multiplicative distortion of the physics, given by
$$q_{\mathrm{mis}}(s_t, u_t) = (x_{2,t+1} - x_{2,t}) \cos \vartheta_t - \beta (x_{1,t+1} - x_{1,t}) \sin \vartheta_t.$$
for some $\beta \neq 1$. Second, we use an angle perturbation, where the physics operator is evaluated using a perturbed angle $\vartheta_{\mathrm{mis},t} = \vartheta_t + \delta\vartheta$, with $\delta\vartheta \neq 0$. We obtain $\|\mathscr{D}(f_\star)\|_{\mathscr{L}^2} = 0.15$ with $\delta\vartheta = 0.005$ and $\beta = 1.2$, and $\|\mathscr{D}(f_\star)\|_{\mathscr{L}^2} = 0.50$ with $\delta\vartheta = 0.5$ and $\beta = 0.5$.

**Results.** The top plot of panel of Figure 3 complements the results displayed in Figure 2 on page 10, presenting the log-log plot of the empirical excess risk (estimation error) as a function of the number of samples $T$. While Figure 2 displayed models trained with and without physics-informed regularization, we now illustrate the effect of incorrect physical priors on the learning rate. Each curve is obtained again by averaging over 20 independent random realizations of the training data, with solid lines indicating the mean estimation error and shaded regions denoting 95% confidence intervals. We can observe that, when the degree of misalignment is mild (for instance, when $\|\mathscr{D}(f_\star)\|_{\mathscr{L}^2} = 0.15$), the estimator still substantially benefits from incorporating physics: the decay rate remains faster than in the case without knowledge. In contrast, when the misalignment grows larger (e.g., $\|\mathscr{D}(f_\star)\|_{\mathscr{L}^2} = 0.5$), the advantage of the prior diminishes, and the observed rate approaches the slower Sobolev minimax behavior of the purely data-driven estimator. We summarize the empirical rates for all the tests in Table 2.

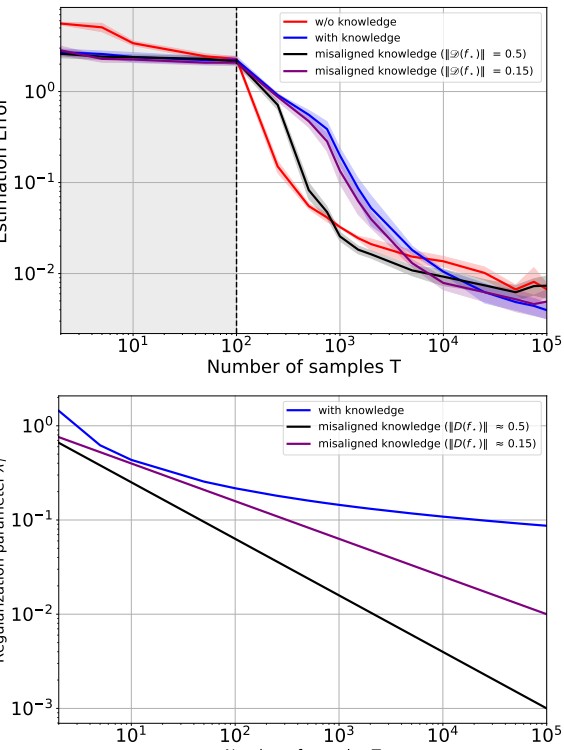

The bottom panel of Figure 3 reports the behavior of the regularization parameter $\lambda_T$ as a function of the sample size $T$. Consistent with our theory, as the prior becomes increasingly unreliable, $\lambda_T$ must decrease more aggressively: this prevents the learner from enforcing an inaccurate physical bias too strongly, striking an appropriate balance between data fit and physics regularization.

Figure 3: Log-log plots of the empirical excess risk (estimation error) for the unicycle experiment (top panel), shown as a function of the sample size $T$. The gray shaded region on the left highlights the estimated burn-in period, after which the learning rate becomes apparent. The bottom panel displays the behavior of the regularization parameter $\lambda_T$ as $T$ increases.

Table 2: Comparison of average learning rates for the unicycle experiment.

| Experiment | No-Knowledge | Full-Knowledge $\|\mathscr{D}(f_\star)\|_{\mathscr{L}^2} = 0$ | Misaligned Knowledge $\|\mathscr{D}(f_\star)\|_{\mathscr{L}^2} = 0.15$ | Misaligned Knowledge $\|\mathscr{D}(f_\star)\|_{\mathscr{L}^2} = 0.5$ |
|---|---|---|---|---|
| Unicycle | $T^{-0.646}$ | $T^{-0.993}$ | $T^{-0.972}$ | $T^{-0.700}$ |

### K.2 Poisson Equation Experiment

We complement the unicycle experiment with an additional numerical study based on a two–dimensional Poisson equation. This example illustrates the same qualitative behavior observed in the unicycle setting: physics-informed regularization accelerates convergence even when the data are dependent.

**Model setup.** Consider the Poisson equation on the unit square $\Omega = [0,1]^2$:

$$-\Delta f_\star(x,y) = g(x,y), \text{ for } (x,y) \in \Omega,$$

with homogeneous Dirichlet boundary conditions. We choose a ground-truth solution

$$f_\star(x,y) = \sin(\pi x)\sin(\pi y),$$

for which the forcing term is analytically given by

$$g(x,y) = 2\pi^2 \sin(\pi x)\sin(\pi y).$$

The physical operator considered in the regularization term is the Laplace operator $\mathscr{D}(f) = -\Delta f$, which satisfies Assumption 4.

**Sequential data generation.** To illustrate the dependent-data setting, we generate inputs sequentially. Starting from an initial point $(x_0, y_0)$ drawn uniformly from $\Omega$, we propagate the trajectory according to the stochastic Markovian dynamics

$$(x_{t+1}, y_{t+1}) = (x_t, y_t) + \varepsilon_t, \;\; \varepsilon_t \sim \mathcal{N}(0, \sigma_{\text{step}}^2 \mathbb{I}_2)$$

with $\sigma_{\text{step}}^2 = 0.05$ and $\mathbb{I}_2$ is the $2 \times 2$ identity matrix. To make the generated data satisfy Assumption 1, we project them onto $\Omega$ by clipping each coordinate.
At each time step we observe the noisy evaluation

$$Z_t = f_\star(X_t, Y_t) + W_t,$$

where $W_t$ is an additive Gaussian noise with variance $\sigma_W^2 = 1$.

**Adopted estimator.** We approximate $f_\star$ with a multilayer perceptron with two hidden layers of width 64 and ReLU activations. We train the estimator using the Adam optimizer with learning rate $0.5 \times 10^{-3}$. We vary the number of training samples $T$ over the range $T \in [10, 5 \times 10^4]$. The approximated rate is computed after the burn-in time.

**Experiment regimes.** We compare three learning regimes:

1. **Without knowledge**: $\lambda_T = 0$.

2. **With knowledge**: use the correct operator $\mathscr{D}(f) = -\Delta f$.

3. **Misaligned knowledge**: enforce the incorrect physics prior

$$-\Delta f = \beta g, \;\; \beta \neq 1,$$

leading to the regularizer

$$\|\mathscr{D}(f_\star)\|_{\mathscr{L}^2} = |1 - \beta|\pi^2,$$

where we use $\beta = 0.93$ to obtain $\|\mathscr{D}(f_\star)\|_{\mathscr{L}^2} = 0.7$ and $\beta = 0.8$ for $\|\mathscr{D}(f_\star)\|_{\mathscr{L}^2} = 1.97$.

**Results.** Figure 4 reports the log–log plot of the empirical excess risk as a function of the number of samples $T$ for the Poisson equation experiment, comparing the learning curves obtained with the three experiment regimes presented above. As in the unicycle experiment, each curve is computed by averaging over 20 independent realizations of the training data, with solid lines showing the empirical mean and shaded regions representing 95% confidence intervals. Consistently with the theoretical predictions in Section 5, the estimator that incorporates the correct PDE structure exhibits a faster decay of the estimation error than the purely data-driven baseline.

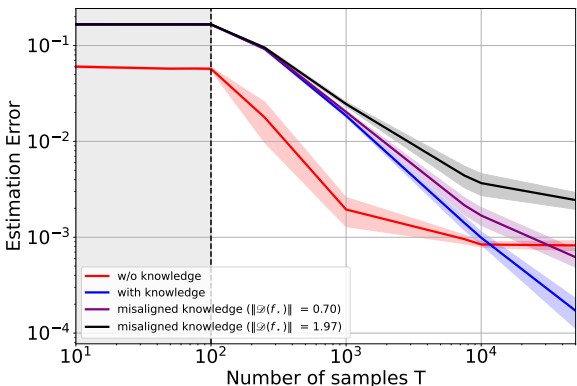

Figure 4: Log–log plot of the empirical excess risk (estimation error) as a function of the sample size $T$ for the Poisson equation experiment. The gray shaded region on the left highlights the estimated burn-in period, after which the learning rate becomes apparent.

Beyond the aligned-knowledge setting, Figure 4 also highlights the effect of misaligned physical priors on the convergence behavior. When the degree of misspecification is moderate, for instance when $\|\mathscr{D}(f_\star)\|_{\mathscr{L}^2} = 0.70$, the learning rate remains significantly improved relative to the regime without knowledge, demonstrating robustness of physics-informed regularization to small prior errors. On the other hand, as the misalignment increases further, e.g., when $\|\mathscr{D}(f_\star)\|_{\mathscr{L}^2} = 1.97$, the benefits of incorporating the PDE prior gradually diminish: the empirical decay rate slows down and approaches the behavior of the estimator trained without physics. This transition mirrors the trends predicted by our theory, where the rate interpolates between the fast i.i.d.-like regime under knowledge alignment and the slower Sobolev minimax rate as the prior becomes increasingly incorrect. For a summary of the obtained empirical average learning rates obtained in the studied scenarios, we defer to Table 3.

Table 3: Comparison of Learning Rates for the Poisson Equation Experiment.

| Experiment | No-Knowledge | Full-Knowledge $\|\mathscr{D}(f_\star)\|_{\mathscr{L}^2} = 0$ | Misaligned Knowledge $\|\mathscr{D}(f_\star)\|_{\mathscr{L}^2} = 0.70$ | Misaligned Knowledge $\|\mathscr{D}(f_\star)\|_{\mathscr{L}^2} = 1.95$ |
|---|---|---|---|---|
| Poisson | $T^{-0.741}$ | $T^{-1.105}$ | $T^{-0.935}$ | $T^{-0.733}$ |