# OpenReview forum: "Physics-informed learning under mixing: How physical knowledge speeds up learning"
_ICLR.cc/2026/Conference — ICLR 2026 Poster_

### Official Review · Reviewer_7BGW · 2025-11-01

**Soundness:** 3
**Presentation:** 2
**Contribution:** 3
**Rating:** 6
**Confidence:** 2

**Summary:**

This manuscript presents a theoretical analysis of physics informed machine learning with data are dependent. The work is an attempt to address the open challenge why incorporating physics prior can benefit data-drive learning.

**Strengths:**

1. This paper addresses the open challenge: the theoretical soundness of physics informed ML, beyond empirical evidence and intuitive understanding that the prior physics knowledge is conductive to learning.
2. Theoretical analysis appears to be rigorous (but I didn't check all the proofs).

**Weaknesses:**

1. The organization and exposition of the manuscript can be improved. Without loss of theoretical rigor, it would be great if the intuitive explanations can be provided for deep learning practitioners the meaning of the theoretical results in practice.
2. Related to the first question, intuitive understanding of the importance of conditions related to key variables such as $T$, as in Theorem 5.2 and $\lambda_T$ as in theorem 5.1, would greatly benefit the reader.
3. It's understandable this is a theoretical paper, but the only numerical experiment does add the weight to the paper. A well thought-off experiments to demonstrate the conditions related to $T$ and $\lambda_T$ would also greatly benefit the readers, see the concern above.

**Questions:**

1. Minor formatting issue: all equations should be numbered.

---

> ### Author Response · Authors · 2025-11-20
>
> **On giving more intuitive explanations and strengthening the connection to deep learning.** In the revised version of the paper, which can be found in the updated Supplementary material folder, we included additional references ([1-3]) to better contextualize our work and connect it more with physics-informed deep learning. To further enforce such a connection, for instance, we stressed in the Conclusions that \
> "*The analysis is set in the general framework of statistical learning, and encompasses a wide range of approaches such as kernel-based methods and (deep) neural networks with physics-informed loss functions*". \
> We refer to the revised version of the paper for further added explanations. \
> **On the meaning of the conditions on $T$ and $\lambda_T$.** The conditions on the regularizer $\lambda_T$ and on the sample dimension $T$ presented in Theorems 5.1 and 5.2 emerge naturally from the construction carried out in the paper, and are similar in structure to the ones presented in related works in statistical learning. Specifically, the condition on $T$ is the burn-in time condition, a phase after which the asymptotic behavior becomes observable. We commented more on this in the beginning of Section 5 in the revised paper, where we write:\
> "We finally provide our main results in terms of convergence rates for the excess risk $\Vert \hat{f} - f_{\star}\Vert^2_{\mathscr{L}^2}$, whose detailed proofs are deferred to Section I. Notably, Theorems 4.1 and 4.2 not only lend the expression for the bounds, but also provide sufficient conditions on the regularization parameter $\lambda_T$ and on the minimal sample size $T$ for the bounds to hold." \
> **On the numerical experiment.** We agree that, even if the numerical test is just for illustration purposes, it should cover more aspects of the theoretical results derived in the paper. For this reason, we expanded the numerical example on the unicycle model, which now highlights the burn-in phase, showcases the behavior of the regularizer as the data-set size increases, and displays the impact of mis-aligned prior information. Note also that we provide an additional example featuring the Poisson equation, an elliptic PDE that is common in many physics-informed neural network applications. However, we emphasize that these numerical tests are just for illustration purposes of our theoretical and algorithm-agnostic results, not for thorough empirical validation. \
> **On numbering the equations.** We adopted the policy of numbering only the equations we refer to in the text, which is one of the customary formatting choices. This allows us to avoid clutter in long proofs with multi-line equations; moreover, it enables us not to break long expressions, such as the burn-in time requirements in Theorems 5.1 and 5.2, over multiple lines, thus enhancing readability.
>
> **References:** \
> [1] M. Raissi et al. Physics-informed neural networks: A deep learning framework for solving forward and inverse problems involving nonlinear partial differential equations. Journal of Computational Physics, 2019 \
> [2] S. Cuomo et al. Scientific Machine Learning Through Physics–Informed Neural Networks: Where we are and What’s Next. Journal of Scientific Computing, 2022 \
> [3] Z. Hao et al. Physics-Informed Machine Learning: A Survey on Problems, Methods and Applications. arXiv preprint 2211.08064, 2023.

---

### Official Review · Reviewer_6PJM · 2025-11-01

**Soundness:** 2
**Presentation:** 3
**Contribution:** 3
**Rating:** 4
**Confidence:** 3

**Summary:**

This paper develops a rigorous statistical-learning theory for physics-informed learning under dependent (mixing) data. It studies regularized empirical risk minimization where the regularizer encodes known physical laws through a linear differential operator, and the data arise from a stochastic dynamical system $X_{t+1}=f_{\star}(X_t)+W_t$. Using tools from Sobolev-space analysis, the small-ball method, and martingale offset complexity, the authors prove complexity-dependent excess-risk bounds showing that when the physical prior is aligned with the ground-truth dynamics (i.e., PDE residual of $f_{\star}$ is nearly zero), the learning rate acclerates from the traditional Sobolev minimax rate $O(T^{-d})$ to the fast i.i.d optimal rate $O(1/T)$, even when samples are correlated. A simple unicycle-dynamics experiment empirically confirms the predicted speed-up, demonstrating that properly aligned physics-based regularization can provably improve sample efficiency in learning dynamical systems.

**Strengths:**

The paper addresses an important topic at the intersection of physics-informed learning and statistical learning under dependence. The authors aim to provide a theoretical foundation for when incorporating physical structure can mitigate the challenges of temporally correlated data. In this context, the paper offers several notable positive aspects:

1. The central idea of using elliptic differential operators to encode physical priors and recover i.i.d.-like learning rates (under suitable alignment) despite sample correlation is novel and insightful.

2. The theoretical analysis is mathematically sound. The assumptions are clearly stated, and the main theorems follow logically from the lemmas and proof techniques provided. The paper carefully extends existing theory to accommodate a Sobolev-based physics regularizer in the presence of Markovian data, which is nontrivial. While I did not look into the proofs in detail and some proof components rely on established techniques, the overall combination is coherent and technically competent.

3. The results contribute to a better theoretical understanding of physics-informed learning, particularly in settings where data dependence is unavoidable, such as system identification and scientific modeling. Showing that incorporating correct physical structure can mitigate the adverse effects of temporal correlation is a useful insight, although demonstrated under idealized assumptions.

**Weaknesses:**

While the paper makes a meaningful theoretical contribution and is clearly written, several limitations temper its overall impact and practical relevance. Most of these relate to the idealized nature of the assumptions and the gap between the theory and empirical applicability. The following points highlight areas where the work could be strengthened or where additional clarification or experimentation would improve the contribution:


1. The fast-rate $O(1 / T)$ convergence is achieved only under an idealized knowledge alignment condition, namely when $\|\|D\left(f_{\star}\right)\|\|_{L^2} \approx 0$. In practice, the physical operator $D$ is rarely known with such precision, and even modest mismatch can revert the rate to the slower $O\left(T^{-d}\right)$ regime. The paper does not provide a quantitative robustness analysis that would clarify how sensitive the rates are to partial or imperfect alignment, which limits the applicability of the theoretical claims in realistic settings.

2. The theoretical guarantees depend on $\lambda_T$ choices tied to latent problem quantities (e.g., $\Psi(f_{\star}), \sigma_W^2$ ). While the authors note that cross-validation could be used in principle, the paper offers limited practical guidance or empirical validation for tuning $\lambda_T$ in realistic settings.

3. The single toy experiment (unicycle) is supportive but narrow (one setting, small MLP, no real-world data/baselines), so the robustness of the phase transition across architectures/noise regimes remains unclear. Even in this controlled synthetic setting, implementing the physics-informed regularization term $\Psi(f)= \|\|D(f)\|\|_{L^2}^2$ may require evaluating higher-order derivatives and Sobolev norms, which can be computationally demanding in higher dimensions. In addition, the paper does not discuss how discretization, numerical differentiation, or instability in PDE solvers or neural approximators would affect the performance or validity of the theoretical bounds, leaving a gap between the continuous theory and practical implementation.

4. The framework relies on $D$ being a known, linear elliptic operator. Many modern scientific machine learning applications involve unknown or nonlinear physics, or operators that must be learned jointly with the model (e.g., operator learning, neural PDE surrogates, and PINNs). As the current analysis does not extend to such settings, it is unclear how the insights would generalize to applications where the governing equations are only partially known or inherently nonlinear.

**Questions:**

In relation to the weaknesses stated above, please see my questions/comments below:

1. The paper presents a two-term bound combining the $T^{-d}$ and $T^{-1}$ rates, but it is unclear how to interpret the intermediate regime. Could the authors clarify whether the transition between the two regimes is smooth or abrupt, and provide any threshold conditions under which the fast term becomes dominant? In addition, it'd be great if the authors could shed some light on the following: how robust this behavior is to imperfect knowledge alignment? For example, if $\|\|D\left(f_{\star}\right)\|\|_{L^2}$ is small but nonzero, does the convergence rate degrade gradually (and remain faster than $T^{-d}$ ) or does it collapse sharply to the slower regime?

3. Since $\Psi(f)=\|\|D(f)\|\|_{L^2}^2$ may involve higher-order derivatives, do the authors foresee computational or numerical challenges when scaling beyond low-dimensional synthetic problems? Any guidance on discretization or numerical stability when implementing this regularizer in practice would be helpful.

3. The analysis assumes that $D$ is a linear and elliptic operator, which can be restrictive. Could the authors comment on whether any part of the analysis may extend to mildly nonlinear or non-elliptic operators, or if linear ellipticity is fundamentally required for the proof techniques used?

4. The paper imposes $s \geq 2 d_X$ to ensure the burn-in term vanishes. Is this threshold believed to be intrinsic to the problem, or could the two-phase rate behavior persist under weaker smoothness assumptions (e.g., $s>d_X$ or $s>3 d_X / 2$ ) with possibly different constants?

5. Could the authors provide any preliminary numerical results or intuition on how large $T$ must be for the asymptotic behavior to manifest in practice? Even a brief discussion of the finite-sample regime would help readers assess when the theoretical rates become observable.

---

> ### Author Response · Authors · 2025-11-20
>
> **On the impact of imperfect knowledge alignment.** While the focus of our work is to assess the impact of *aligned* prior knowledge on the learning rate, it is very interesting to understand the overall behavior of the bound when the prior is mis-specified. In general, one can notice that the bounds presented in Section 5 are affine functions of $\Psi(f_{\star})^{\beta}$ for some $\beta < 1$, and one can find some requirements for the slow Sobolev-type term to have minor impact on the overall rate even in presence of knowledge misalignment. For instance, a necessary condition for a mis-specified prior to still benefit the overall rate is that $C_{\texttt{fast}}\sigma_W^2 \log(1/\delta) \gg C_{\texttt{slow}} \Psi(f_{\star})^{d'/2}$. Yet, exact conditions and general robustness analyses are heavily problem-dependent, rely on user-defined tolerance thresholds, and pertain to empirical studies that are out of the scope of this paper. However, to bring the issue of prior misalignment to attention, we expanded our numerical tests to display its effect, also showcasing that the transition to the slow rate does not happen abruptly. Please refer to Section K in the revised version of the paper in the updated Supplementary material, where we compute the empirical learning rates for different values of $\Vert \mathscr{D}(f_{\star})\Vert$ in both the unicycle example and the newly-added illustration based on the Poisson PDE. \
> **On the choice of $\lambda_T$.** The condition on the regularization parameter $\lambda_T$ presented in our results naturally stems from the theoretical and model-agnostic construction: as such, similarly to other works in the statistical learning literature, it depends on unknown quantities, such as $\Psi(f_{\star})$. Since our analysis is carried out for general nonparametric learning and nonlinear regression, practical algorithm-dependent issues such as hyper-parameter estimation are deferred to more empirical studies such as those reviewed in [1]. \
> **On numerical issues in practical implementations.** Even if of great interest, aspects such as discretization, numerical differentiation and solver stability depend on the adopted algorithm and its implementation. All of these issues do not impact the theoretical analysis carried out in our paper, and their investigation requires an extensive empirical testing that is out of the scope of this work. \
> **On the operator being known, linear and elliptic.** Dealing with unknown physics or unspecified operators is an important problem, but it departs from the premises of our work, which aims to quantify the impact of given (and aligned) physical knowledge in nonparametric learning of nonlinear dynamics in presence of dependent data. If the given operator is nonlinear, our current analysis cannot be directly applied, and this is a very interesting future research question. Regarding the assumption on the operator being elliptic, as specified in the Official comment, we were able to relax it to encompass also other scenarios of interest, such as second-order parabolic and hyperbolic PDEs, broadening the applicability of our analysis. \
> **On the requirement $s \geq 2d_{\mathrm{X}}$.** This is a very interesting point. As correctly pointed out, this stricter requirement on the smoothness parameter $s$ is imposed in our set-up to ensure that the burn-in is well-defined. In general, one can intuitively explain that a larger smoothness degree is needed to cope with dependent data; however, as posted also in the Official comment, we do not exclude that different analysis strategies could lead to a relaxation of the assumption on $s$, possibly at the price of strengthening some other requirement. \
> **On the finite-sample regime.** Our theoretical analysis captures the behavior of general nonparametric learning after the burn-in time: thus, it cannot characterize the finite-sample regime, which depends heavily on the algorithm and its implementation. Also, similarly to what happens with the regularization parameter $\lambda_T$, in practice one can only approximate the value of the burn-in, because its expression depends on unknown quantities such as $\Psi(f_{\star})$. However, we expanded the numerical illustrations to bring this aspect into discussion. Specifically, for both unicycle and Poisson equation examples, we now display the estimated burn-in time after which the predicted behavior of the learning rate becomes observable.
>
> **References:** \
> [1] R. Rai and S. Sahu. Driven by Data or Derived Through Physics? A Review of Hybrid Physics Guided Machine Learning Techniques With Cyber-Physical System (CPS) Focus. IEEE Access, 2020.

---

### Official Review · Reviewer_3SEU · 2025-11-01

**Soundness:** 4
**Presentation:** 3
**Contribution:** 3
**Rating:** 8
**Confidence:** 3

**Summary:**

The paper derives complexity-dependent bounds that formally characterize the impact of aligned prior domain knowledge on learning rates in physics-informed machine learning settings. This is approached by focusing on bounds of the excess risk in regularized ERM under dependent data generated from non-linear dynamical systems. Results under the combined assumptions of physics-informed regularization and non-IID data have yet to be shown in existing literature, and the derived bounds reveal new insights into the relationship between statistical learning theory and physics-informed ML.

**Strengths:**

- The paper is well-organized and has a presentation that's easy to follow from start to finish. I appreciated the clear groups in Section 3 and the visual distinction for all stated assumptions.
- The authors address an important theoretical challenge in the physics-informed ML space, with a fundamental connection to all PIML architectures and practitioners looking to tackle scientific problems. Characterizing the impact of well-aligned domain knowledge takes a clear step toward helping researchers and practitioners alike when it comes to justifying additional time spent refining assumptions and calibrating inductive biases.
- Relaxing the IID assumptions from prior work (e.g., Doumèche et al. 2024) appears to be an important adjustment that broadens the reach of the resulting bounds. In most PIML settings, one is working with heavily dependent sequential data, so this seems a prudent step toward establishing useful bounds that mirror the real world.

**Weaknesses:**

- Assumption 5 is quite strong, limiting the applicability of the results in many real-world physics-informed modeling scenarios. For instance, unless additional constraints are applied, popular methods like physics-informed neural networks would presumably violate this assumption, and in these cases it's unclear how one should think about the applicability of the results. I understand details on approximations were stated as out of scope, but it would be nice to include some discussion/analysis of the likely ways practitioners may violate assumptions in practice and what elements of the original bound still apply (if any).
- Similar to the above point, Assumption 4 also excludes common practical scenarios that leverage non-linear PDE priors. Discussion on what remains of the bound in these cases would be helpful for position the paper's analysis in a broader context.
- This is somewhat tangential to the core theoretical aims of the paper, but it would be instructive to include a more holistic case study. In particular, the impact of using misaligned/incomplete prior physics knowledge and how this empirically threads between the "with knowledge" and "without knowledge" scenarios. The sample sizes at which various levels of knowledge alignment outperform others, if only for short ranges of $T$, would also be helpful to characterize insofar as they relate to the paper's analysis of sample size.

**Questions:**

- Are there any promising avenues for achieving the stronger burn-in requirement while relaxing $s\ge 2d_X$ toward the standard $s\ge d_X/2$, or are there clear reasons to believe this is a necessary tradeoff? Can factors/assumptions be loosened in its place while maintaining the original bound, giving another route to the same result?
- In the numerical experiment and Figure 2, does the reported slope fit account for burn-in (if even needed in this case)? It would be instructive to see how this shows up here, and/or when burn-in exceeds realistically attainable sample sizes in practice.

---

> ### Author Response · Authors · 2025-11-20
>
> **On Assumption 5.** As detailed in the Official comment, this assumption allows us to focus on the data-dependent, stochastic component of the excess risk. In the revised version of the paper, which can be found in the updated Supplementary material, we provide more references from the statistical learning literature that quantify the bias error. Specifically, the revised comment below Assumption 5 now reads as follows: \
>  ``*In case such an assumption is violated, the excess risk bound would feature an additional term quantifying the error introduced by the mis-specification of the hypothesis space $\mathscr{F}$. Such a bias error is entirely data-independent and deterministic: therefore, it is decoupled from the analysis performed in this paper. In general, tools to bound the bias error rely on interpolation theory [1], and some results can be found, e.g., in [2,3].*"\
> **On linearity of PDEs.** We agree that treating nonlinear PDEs would be of great interest -- however, the present analysis does not extend to such a scenario in a straightforward manner, and we leave this topic to further investigations. Furthermore, we point out that we were able to generalize Assumption 4, making it possible to consider also second-order hyperbolic and parabolic PDEs: please refer to the Official comment for more details. \
> **On more holistic case study.** We expanded the numerical tests, showing (i) how misaligned priors could still positively impact the learning rate, (ii) the estimated burn-in period, and (iii) the expected behavior of the regularization parameter $\lambda_T$ -- please see the updated Supplementary material. Note also that we include an additional example involving the Poisson PDE.
> However, an extensive empirical study is out of the scope of this paper and is left to future investigations. \
> **On relaxing $s>2 d_{\mathrm{X}}$.** This is a very interesting point. For now, imposing such a condition on $s$ stems from our construction -- however, as presented in the Official comment, we could expect that other analysis procedures could lead to having the more standard $s \geq d_{\mathrm{X}}/2$ at the price of restricting other assumptions. \
> **On displaying the burn-in.** In the previous version of the paper, the numerical tests were not displaying the burn-in. We now extended the discussion on the numerical illustrations, where we highlight the estimated burn-in phase, which takes approximately 100 samples in both examples.
>
> **References:**\
> [1] A. Lunardi. Interpolation theory. Publications of the Scuola Normale Superiore di Pisa, 2018.\
> [2] F. Cucker and S. Smale. On the mathematical foundations of learning. Bulletin of the American Mathematical Society, 2002.\
> [3] F. Cucker and D.-X. Zhou. Learning theory: An approximation theory viewpoint. Cambridge University press, 2007.

---

### Official Review · Reviewer_tMrX · 2025-11-08

**Soundness:** 3
**Presentation:** 2
**Contribution:** 3
**Rating:** 6
**Confidence:** 3

**Summary:**

This paper presents a theoretical analysis of physics-informed learning in the presence of dependent data. The authors analyze empirical risk minimization with a physics-informed regularizer that encodes known physical priors in the form of elliptic PDE constraints. Using the small-ball method and martingale offset complexity, they derive complexity-dependent excess risk bounds both in probability and expectation. The main result shows that when the physical prior regularizer is aligned with the true dynamics, the convergence rate improves from the slow Sobolev minimax rate to the optimal i.i.d. rate, even with dependent samples. The theoretical framework is supported by a unicycle dynamics experiment illustrating the empirical benefit of incorporating physics-informed regularization.

**Strengths:**

1. The paper rigorously extends the small-ball method and offset complexity analysis to dependent data settings with physics-informed regularization, which is technically novel.

2. The paper establishes a clear theoretical connection between physical priors (encoded as elliptic PDE constraints) and improved learning rates, addressing a long-standing gap in understanding the benefits of physics-informed models.

3. The proofs and appendices appear detailed, well-grounded in functional analysis, Sobolev space theory, and dependent process theory.

4. The  experiment, though simple, can demonstrate the theoretical prediction that physical priors accelerate convergence.

**Weaknesses:**

1. The analysis assumes elliptic linear PDE operators; it is unclear whether the results generalize to non-elliptic, nonlinear, or mixed-type operators often seen in physics-informed neural networks (PINNs). The gap or difficulty is not well addressed.

2. The optimization error's influnene is not discussed in the theoretical part and the numerical example. Note that the physics-informed regularizer will increase the stiffness of the Hessian matrix and increase difficulties for the optimization in PINNs, which is not aligned with the main result (convergence rate improves with physics-inforemed regularizer added)

3. Too many assumptions may not hold for complex real-world systems, a more interpretable or verifiable condition would strengthen the practical relevance.

4. Only one low-dimensional example is provided. Additional tests on nonlinear PDE systems or stochastic dynamical systems would reinforce the claims.

**Questions:**

1. see in the weakness

2. The phsyics-informed regularizer is limited to linear elliptic PDEs. However, the data is dependent, which is always occured in dynamical systems. Is this two conflicting? The numerical example is an ODE, not elliptic PDEs. How the convergence rate behaves for common PINNs problems such as Poisson equation or Darcy flow problem? How can the proposed theory be connected to the optimization landscape of neural networks trained with PINNs?

3. Does the empirical rate in the numerical example persist when neural architectures differ from MLPs, or with stochastic training?

4. Can the presentation be more friendly to the general ICLR audience without additional intuition or graphical explanation of the key proof mechanisms? The current form is mathematically dense and more suitable to journals like JMLR, not general top conferences.

---

> ### Author Response · Authors · 2025-11-20
>
> **On linear and elliptic PDEs.** As presented in the Official comment, we were able to relax Assumption 4 to account also for linear, second-order hyperbolic and parabolic PDEs. The extension to nonlinear PDEs is not immediate and will be subject of future research. \
> **On optimization errors.** We agree that numerical aspects are key in practical implementations. However, their analysis heavily depends on the specific algorithm and problem at hand: thus, such an investigation is out of the scope of our paper, which
> is focused on the theoretical analysis of any supervised learning algorithm that leverages prior (aligned) physical information. \
> **On standing assumptions.** The assumptions made in the paper are quite standard, general and verifiable.
> Specifically, Assumption 1 encompasses a wide range of probability distributions encountered in practice, and the existence of their density is a minimal requirement. For Assumption 3, having noises described by martingale difference sequences is the most typical set-up in learning with dependent data, and assuming that their distribution is sub-Gaussian is very standard and general. We defer to the Official comment and to the revised paper for the discussion on Assumptions 2, 4, 5. In light of this, our framework is capable of capturing many complex scenarios of practical relevance, and relies on standard and general assumptions --  even more now that Assumption 4 has been relaxed.\
> **On the numerical test.** Even if low-dimensional, the unicycle experiment reflects a realistic robotics application and captures the set-up investigated in the paper, being the involved ODE a particular case of PDEs; also, since measurements are noisy, the example describes a stochastic system. We did not consider nonlinear PDEs because our theoretical framework does not seamlessly apply to them. However, we added in the revised paper (see updated Supplementary material) an example involving a Poisson equation, as presented below. \
> **On elliptic PDEs and dependent data coexisting.** There is no conflict between the two: the effective hypothesis space $\mathscr{F}^{\rho}$, depending on the differential operator $\mathscr{D}$, is contained in the ball of the weighted, vector-valued Sobolev space constructed in Appendix B, which accounts for trajectories of dependent data. \
> **Application to Poisson equation and Darcy flow.** Being these great examples of linear, elliptic PDEs, our analysis holds for them, and the obtained convergence rate holds regardless of the adopted learning algorithm. We performed an additional numerical test on the Poisson equation, which we report in the revised Section K, showcasing a similar empirical behavior compared to the unicycle experiment. \
> **Connection with optimization landscape of PINNs.** Our physics-informed regularizer is closely related to the PDE-residual penalties typically used in PINNs, where several works have shown that such soft physics constraints reshape the nonconvex optimization landscape [1,2]. Our contribution is complementary: instead of analyzing optimization trajectories of
> PINN models, we study how the same type of soft physics penalty reduces the effective hypothesis class, thereby reducing statistical complexity under dependent data. This contraction eliminates large physically-implausible regions of the landscape, which explains the transition from the slow Sobolev rate to the fast $\mathcal{O}(T^{-1})$ rate under knowledge alignment. This interpretation parallels the role of weight decay as a soft inductive bias in deep learning and large-scale models [3-5], where penalties reshape the geometry of the loss surface by shrinking the search space. Our analysis thus complements the PINN landscape literature by providing a rationale for why aligned physics constraints can improve sample efficiency. \
> **On the role of the architecture in the empirical rate.** Empirical rates might change when using different learning algorithms. However, our theoretical results are algorithm-agnostic: thus, if the learning approach makes use of correct prior
> information, we expect to observe the same relative behavior displayed in our numerical illustrations. \
> **On presentation.** Without compromising the mathematical rigor that is considered essential for our results, we expanded several parts with further explanations and added references on empirical studies in PIML/PINNs: please see the revised paper in the updated Supplementary material.
>
> **References:**\
> [1] M. Raissi et al.  Physics-Informed Neural Networks. JCP, 2019\
> [2] A. Krishnapriyan et al. Characterizing Possible Failure Modes in Physics-Informed Neural Networks. NeurIPS, 2021\
> [3] A. Hoerl and R. Kennard. Ridge Regression: Biased Estimation for Nonorthogonal Problems. Technometrics, 1970\
> [4] C. Zhang et al. Understanding Deep Learning Requires Rethinking Generalization. ICLR, 2017\
> [5] AG Wilson. Deep Learning is Not So Mysterious or Different. ICML, 2025.

---

> > ### Comment · Reviewer_tMrX · 2025-11-24
> >
> > Thanks for the respond. What I mean in Q2 is that, for ODEs, it is common to choose dependent data; but for PDEs, we often choose collocation points randomly in PINN's loss, these points are not dependent, why should you study the dependent-case? In line 2990 of the revised paper, how to keep the generated data in the bounded domain $\Omega$ if you apply the Markovian dynamics sequential data?

---

> > > ### Author Response · Authors · 2025-11-25
> > >
> > > **On studying the dependent-data case for PDEs.** Thank you for following up on this point. The set-up considered in our paper captures scenarios in which data are sequentially generated according to $X_{t+1} = f_{\star}(X_t) + W_t$ (see (3.1) in our paper). On top of this, we assume that $f_{\star}$ satisfies a partial differential equation given by the linear (elliptic, hyperbolic or parabolic) operator $\mathscr{D}$ entering our physics-informed regularizer. These two aspects are not conflicting, as seen for instance in time-stepping schemes in parabolic PDEs and in MCMC-type samplers in the spatial domain -- all scenarios that are encompassed by our setting. Furthermore, we would like to emphasize that the i.i.d set-up can be regarded as a particular case of our construction: indeed, as remarked in Section A.2, the $S$-persistence assumption with $S=1$ captures the scenario of independent data which, as correctly pointed out, is common in many PINNs with PDEs.
> > >
> > > **Clarification on the experiment involving the Poisson equation.** Thank you for spotting this missing piece of information. In the newly-added experiment in Appendix K.2, whenever the point in the trajectory falls outside the domain $\Omega$, we project it back onto it by clipping each coordinate: this allows us to generate data satisfying Assumption 1. We will rephrase the paragraph titled "Sequential data generation" accordingly as follows: \
> > > "To illustrate the dependent-data setting, we generate inputs sequentially. Starting from an initial point $(x_0,y_0)$ drawn uniformly from $\Omega$, we propagate
> > > the trajectory according to the stochastic Markovian dynamics \
> > > $(x_{t+1}, y_{t+1}) = (x_t, y_t) + \varepsilon_t, \quad
> > > \varepsilon_t \sim \mathcal{N}(0, \sigma^2_s \mathbb{I}_2),$ \
> > > with $\sigma^2_s = 0.05$ and $\mathbb{I}_2$ is the $2\times 2$ identity matrix. To make the generated data satisfy Assumption 1, we  project them onto $\Omega$ by clipping each coordinate."

---

> > > > ### Comment · Reviewer_tMrX · 2025-11-28
> > > > **Thank you**
> > > >
> > > > Thank you for the detailed response. I tend to keep my positive score.

---

### Author Response · Authors · 2025-11-20

First of all, we would like to thank the Reviewers for the careful reading of our paper and for their insightful comments.

Motivated by the interest and success of physics-informed machine learning algorithms, our paper aims at theoretically proving their underpinning conjecture, namely, that aligned prior knowledge speeds up learning. Set in the context of statistical learning, our construction provides the first convergence rates for physics-informed learning under dependent data, deriving results that are entirely algorithm-agnostic and hold for any nonparametric approach.\
Many of the Reviewers' questions regarded practical aspects such as implementation, neural network architecture and optimization errors.
We appreciate that these are issues of tremendous interest in machine learning practice, but answering them would require an extensive empirical study that is out of the scope of this paper, which does not focus on any specific algorithm. Nevertheless, we made a more explicit connection to practical implementations, enriching the bibliography on physics-informed machine learning and deep networks. Also, following the suggestion of Reviewers tMrX and 7BGW, we expanded the explanation of our results to make our work more accessible -- please find the proposed revised version in the Supplementary material, where we uploaded the new version with modifications in blue-printed fonts.

Along the same lines, we would like to point out that the numerical illustration presented in the paper is just to display a sample behavior of what is predicted by the theory: the empirical excess risk decreases faster if injected with correct physical knowledge. Its goal is not to claim superiority of one algorithm compared to a baseline. However, we modified its presentation, featuring more comments on practical issues like burn-in quantification, finite-sample behavior, selection of $\lambda_T$ and impact of non-aligned prior knowledge (Reviewers 3SEU, 6PJM and 7BGW).

Furthermore, we appreciated the Reviewers' comments on the standing assumptions in our paper:
- Reviewers tMrX, 3SEU, 6PJM asked about relaxing Assumption 4 on having the differential operator to be linear and elliptic. While unfortunately our analysis does not carry over seamlessly to nonlinear operators (which will be a compelling future research direction), we were able to relax the requirement on the PDE to be elliptic. Indeed, such a condition lends us the norm inequality (C.2), which we use only in Proposition C.4 to compute the covering-number bound for the effective hypothesis space $\mathscr{F}^{\rho}$. However, elliptic regularity is only a sufficient, not necessary condition for inequalities like (C.2) to hold: similar results can be obtained (i) for second-order parabolic operators, via $\mathscr{L}^2$-regularity in anisotropic Sobolev spaces for $\mathscr{D} = \partial_t - \mathscr{A}$, with $\mathscr{A}$ being the spatial differential operator ([1], Ch.7.1.3); and (ii) for second-order hyperbolic operators, relying on energy estimates ([1], Ch.7.2.3). Thanks to this, we were able to extend the applicability of our results to other priors of interest, such as those given by the heat and wave equations.
- Reviewers tMrX and 3SEU consider Assumption 5 (on having the unknown function to be estimated to be contained in the hypothesis space) very restrictive. In general, such an assumption is quite common in the learning theory literature and allows us to focus on the stochastic component of the excess risk, deriving the results presented in our paper. If Assumption 5 were violated, the excess risk bounds would feature an additional deterministic term given by the choice of the hypothesis space. Bounds for such a bias have been derived in the statistical learning literature leveraging, e.g., interpolation theory [2] -- however, being entirely data-independent, these bounds are essentially decoupled by the ones derived in our paper, and our analysis would not advance their theory. Nevertheless, we expanded the comment on Assumption 5 based on these considerations.
- Reviewers 3SEU and 6PJM ask about different avenues to relax Assumption 2 on the Sobolev space order $s$, which is indeed a very interesting matter. At the moment, the stricter assumption on $s$ is needed to have a well-defined burn-in time, which in turn emerges from the whole construction of martingale offset complexity bounds. Even if intuitively one can explain that a larger smoothness degree is needed to cope with dependent data, we do not exclude that different set-ups could lead to relaxing the assumption on $s$, possibly introducing some restriction elsewhere.

Please refer to the individual comments to Reviewers for further detailed responses.

**References:**\
[1] L. Evans. Partial differential equations. Vol.19. American mathematical society, 2022.\
[2] F. Cucker and D.-X Zhou. Learning theory: An approximation theory viewpoint. Cambridge University press, 2007.

---

### Meta-Review · Area_Chair_qtw9 · 2026-01-07

**Summary:**

Three out of four reviewers were positive. The remaining one was somewhat negative. This reviewer's main concern was that the results seem to rely on idealized assumptions, and there might be a gap between the theory and practice. Since the discussion was terminated, it is unclear whether this concern was addressed or not. However, this reviewer also acknowledged the theoretical contribution to a certain extent. Rejecting this paper solely based on this concern would be undesirable.

**Reviewer Concerns:**

While the paper makes a meaningful theoretical contribution, its practical impact is limited by the highly idealized assumptions and the gap between theory and real-world applicability. The analysis assumes known linear elliptic PDE operators, leaving it unclear whether the results extend to more practical problems. Some assumptions are very strong and can be violated in practical PINN settings. Overall, there might be a gap between the theory and practice.

To address the concerns, the authors explained that while extending to nonlinear equations is difficult, their analysis can be extended to parabolic and hyperbolic linear partial differential equations. They also explained that the assumptions are justified to a certain extent.

Unfortunately, only a part of the discussion remains. In the discussion, one reviewer responded that he/she would maintain the positive score. It remains unclear whether the concerns raised by the negative reviewer have been addressed or not.

**Reviewer Scores:**

The scores were 8-6-6-4. I suppose there has been no change in the positive scores, and the negative score of 4 could be raised. So the expected average score is slightly above 6, which is around the typical acceptance threshold.

---

### Decision · Program_Chairs · 2026-01-26

Accept (Poster)